# Universal Learning of Nonlinear Dynamics

## Abstract

We study the fundamental problem of learning a marginally stable unknown nonlinear dynamical system. We describe an algorithm for this problem, based on the technique of spectral filtering, which learns a mapping from past observations to the next based on a spectral representation of the system. Using techniques from online convex optimization, we prove vanishing prediction error for any nonlinear dynamical system that has finitely many marginally stable modes, with rates governed by a novel quantitative control-theoretic notion of learnability. The main technical component of our method is a new spectral filtering algorithm for linear dynamical systems, which incorporates past observations and applies to general noisy and marginally stable systems. This significantly generalizes the original spectral filtering algorithm to both asymmetric dynamics as well as incorporating noise correction, and is of independent interest.

## 1 Introduction

The problem of learning a dynamical system from observations is a cornerstone of scientific and engineering inquiry. Across disciplines ranging from econometrics and meteorology to robotics and neuroscience, a central task is to predict the future evolution of a system given only a sequence of its past measurements. Formally, given a sequence of observations $(y_1, ..., y_t)$ generated by an unknown, underlying dynamical system[1] of the form

$$x_{t+1} = f(x_t), \quad y_t = h(x_t), \tag{1.1}$$

the objective is to produce an accurate forecast $\hat{y}_{t+1}$. The challenge lies in the fact that the state $x_t$ is hidden and may be high-dimensional, and the dynamics function $f : \mathcal{X} \to \mathcal{X}$ and observation function $h : \mathcal{X} \to \mathcal{Y}$ are unknown.

Historically, two dominant paradigms have emerged to address this challenge. The first, rooted in control theory and system identification, aims to construct an explicit model of the underlying dynamics. This model-based approach seeks to learn the function $f$ or, more commonly, a linear representation of it. A powerful theoretical tool in this domain is the Koopman operator, which recasts the nonlinear dynamics of states into an infinite-dimensional, yet linear, evolution of observable functions. Data-driven methods such as Dynamic Mode Decomposition (DMD) attempt to find finite-dimensional approximations of this linear operator, yielding a model that can be used for prediction and control Brunton et al. (2021). While elegant, these methods often rely on strong assumptions to guarantee the convergence of their approximations and can be computationally demanding.

The second paradigm, driven by recent advances in machine learning, employs powerful, general-purpose black-box architectures to learn the mapping from past to future observations directly. Architectures such as Transformers Vaswani et al. (2017) and more recent state-space models like Mamba Gu & Dao (2023) and convolutional models like Hyena Poli et al. (2023) have demonstrated remarkable success in sequence modeling tasks. These methods are highly expressive and often achieve state-of-the-art performance, but their operation can be opaque, and they typically lack the formal performance guarantees characteristic of control-theoretic approaches, especially in the presence of adversarial disturbances or for systems operating at the edge of stability.

**Provable improper learning for dynamical systems.** This paper advocates for a third paradigm: improper learning for dynamical systems. The philosophy of this approach is that for the specific goal of prediction, it is not necessary to recover the true, and often intractably complex, underlying dynamics. Instead, one can design an efficient and provably correct algorithm by framing the learning problem as a competition. The algorithm does not attempt to learn a model from the same class as the true system (hence, "improper"); rather, it competes in hindsight against a class of idealized, well-behaved, and computationally tractable predictors.

---

[1]For ease of presentation, we keep the discussion deterministic in the main paper and discuss an extension to stochastic systems in Appendix J.

This reframing of the problem is a key conceptual leap. It shifts the focus from the difficult, nonconvex problem of system identification to the more manageable, often convex, problem of regret minimization against a carefully chosen comparator class. The central idea of this work is to leverage the rich mathematical structure of control theory not for designing a controller, but for defining this very comparator class. Instead of attempting to learn the true nonlinear system, we design an algorithm that is guaranteed to perform as well as the best possible high-dimensional linear observer system for the given observation sequence. This approach allows us to sidestep the complexities of nonlinear system identification while retaining strong theoretical guarantees. Furthermore, our analysis yields a natural quantitative description of the complexity of learning from observations with improper algorithms.

**Our contributions.** This work makes several contributions to the theory and practice of learning dynamical systems, bridging concepts from classical control theory and modern machine learning.

1. We introduce **Observation Spectral Filtering** (OSF), an efficient algorithm based on online convex optimization that is guaranteed to learn any observable nonlinear dynamical system with finitely many marginally stable modes. The algorithm is "improper" in that it does not explicitly estimate the system's hidden states or dynamics; instead, it directly constructs a mapping from past observations to future predictions.

2. An **LDS guarantee:** The core technical engine of OSF is a learning guarantee for linear dynamical systems (LDS). This method applies to asymmetric dynamics under adversarial noise with performance guarantees that are independent of the system's hidden dimension. This significantly generalizes prior spectral filtering methods, which were largely restricted to symmetric systems, and aligns our work with recent efforts to extend these powerful techniques to broader system classes.

3. **A control-theoretic analysis:** We provide a novel analytical framework that connects the learnability of a nonlinear system to the properties of its best possible high-dimensional linear observer. This connection is formalized through a "Luenberger program," an optimization problem whose solution quantifies the inherent difficulty of learning the system from its outputs. The optimal value of this program, a condition number we denote $Q_\star$, appears directly in our data-dependent learning bounds. This framework establishes a quantitative link between the control-theoretic concept of observability and the machine learning concept of learnability, and provides an elegant analytic toolkit with which to derive learning guarantees in nonlinear settings using linear methods. We discuss further implications in Section 6.

4. **A simple global linearization:** We introduce a simple construction to approximate any bounded, Lipschitz nonlinear system with a high-dimensional LDS. This technique, based on state-space discretization, provides a rigorous foundation for applying our linear systems algorithm to the nonlinear case. It trades an increase in dimensionality—to which our learning algorithm is immune—for guaranteed linearity and approximation accuracy, and it uses the improper nature of our analysis to circumvent the strong spectral assumptions and convergence difficulties associated with many contemporary data-driven Koopman operator methods.

Ultimately, this work presents a synthesis of ideas from two distinct fields. It repurposes tools from control theory (such as the Luenberger observer, pole placement, and global linearization) not for their traditional purpose of observer/controller design, but as analytical instruments to define a robust comparator class for a learning algorithm. By then applying the machinery of online convex optimization, we develop an algorithm whose performance guarantees are directly informed by the structural properties of the underlying prediction problem. This combination of a constructive global linearization and spectral filtering yields a universal learning algorithm for any observable nonlinear dynamical system with finitely many marginally stable modes, with regret scaling determined only by the system's robust observability constant $Q_\star$.

## 2 RELATED WORK

In this section, we survey the past and current methods for sequence prediction in dynamical systems, highlighting their relationships, strengths, and limitations. While it is impossible to provide an exhaustive account, we highlight the works most relevant to our study.

### 2.1 METHODS FOR LEARNING IN NONLINEAR SYSTEMS

Learning in nonlinear dynamical systems of the form (1.1), also called time series prediction in statistics, spans several fields of study. We can roughly divide the existing methods as follows:

**Statistical and online learning methods.**   The classical text of Box and Jenkins Box & Jenkins (1976) introduces ARMA (autoregressive moving average) statistical estimation techniques and their extensions, notably ARIMA. These methods are ubiquitous in applications and were extended to the adversarial online learning domain in Anava et al. (2013); Kuznetsov & Mohri (2018) via online convex optimization Hazan (2016). Later extensions include nonlinear autoregressive methods augmented with kernels, e.g. Cuturi & Doucet (2011). Also prominent in the statistical literature are nonparameteric time series methods such as Gaussian processes Rasmussen (2003); Murphy (2012).

**Deep learning methods.**   The most successful methods for learning dynamical systems are based on deep learning. Transformers Vaswani et al. (2017) are the most widely used architectures that form the basis for large language models. More recent deep architectures are based on convolutional models such as Hyena Poli et al. (2023) and spectral architectures such as FlashSTU Liu et al. (2024). Recurrent neural networks and state space models such as Mamba Gu & Dao (2023) exhibit faster inference and require lower memory, albeit being harder to train and potentially less expressive.

**Koopman operator methods.**   The most relevant technique for learning nonlinear dynamical systems is via the Koopman operator. This methodology relies on the mathematical fact that any (partially observed) dynamical system can be lifted to a linear representation in an infinite-dimensional space. The basic idea is to represent the evolution of the system by describing the evolution of all possible system observables via the Koopman operator, which becomes linear and allows for functional-analytic and spectral methods; see Mezic (2020) for a survey of the related theory. Detailed mathematical study of this operator can require complicated analysis and strong assumptions; a simple version of a related global linearization can be achieved via discretization, which we present in Section C.3.

Data-driven methods for Koopman operator learning are based on learning an approximate representation of the dynamics in high-dimensional linearly-evolving coordinates. Knowing the top $N$ eigenvalues and eigenfunctions of the Koopman operator allows one to approximately simulate, predict, and control the system using $N$-dimensional linear methods. These eigenfunctions may be learned from data via the use of SVD on the trajectories, a predetermined dictionary of reference functions, or a neural network, which inspires the classical DMD and eDMD algorithms. The linear coordinates are often learned jointly with the dynamics in this linear representation via regression. In short, this pipeline learns a lifting to a linear system on which system identification is performed. See the survey Brunton et al. (2021) for more details about this approach.

In contrast, our method is *improper*, i.e. it does not learn the mapping of the nonlinear dynamical system to higher dimensions explicitly. We rely on the spectral filtering method to automatically learn the best predictor that competes with a linear predictor with very high-dimensional hidden state.

## 2.2   METHODS FOR LEARNING IN LINEAR SYSTEMS

The simplest and most commonly studied dynamical systems in the sciences and engineering are linear. Given input vectors $u_1, \ldots, u_T \in \mathbb{R}^{d_{\text{in}}}$, the system generates a sequence of output vectors $y_1, \ldots y_T \in \mathbb{R}^{d_{\text{out}}}$ according to the following equations

$$x_{t+1} = Ax_t + Bu_t + w_t \tag{2.1}$$
$$y_t = Cx_t + Du_t + \xi_t,$$

where $x_0, \ldots, x_T \in \mathbb{R}^{d_h}$ is a sequence of hidden states, $(A, B, C, D)$ are matrices which parameterize the LDS, and $w_t, \xi_t$ are perturbation terms. We assume w.l.o.g. that $D, \xi_t = 0$, as these terms can be folded into the input of the previous iteration. The problem of *prediction* in such systems refers to predicting the next observation $y_t$ given all previous inputs $u_{1:t}$ and past observations $y_{1:t-1}$.

We survey the problem of prediction in linear dynamical systems (LDS) and how our results advance it in Appendix A. To summarize, linear systems become difficult due to either marginal stability (eigenvalues of $A$ near the unit circle) or complex system eigenvalues. Despite this, simple improper learning algorithms can learn linear systems well; the state of the art is spectral filtering, which competes against marginally-stable LDS's with real system eigenvalues.

## 3   SETTING

We consider nonlinear dynamical systems of the form (1.1), which generate a sequence $y_1, \ldots, y_T$ of observations. Our main results are proved as loss bounds in realizable settings, i.e. when the data is generated from a ground truth dynamical system. Since the derivations involved go through the language of online convex optimization, these can be converted to agnostic regret guarantees, which we do in Appendix G.

**Assumption 3.1** (Nonlinear data generator). *The signal $(y_t)_{t=1}^T$ is generated by a nonlinear dynamical system $(f, h, x_0)$ as per equations (1.1). We assume that:*

*(i) For all $t$, the states and signal are bounded by $\|x_t\|, \|y_t\| \leq R$.*

*(ii) The dynamics $f$ and observation $h$ are 1-Lipschitz.*

*(iii) The discretized linear lifting $(A', C')$ defined in Lemma C.5 is an observable LDS.*

## 3.1 Notation

We use $\mathbb{D}_r$ to denote the complex disk of radius $r$, i.e. $\mathbb{D}_r = \{\lambda \in \mathbb{C} : |\lambda| \leq r\}$. We use the notation $v_{a:b}$ to denote stacking the sequence $v_a, v_{a+1}, \ldots, v_{b-1}, v_b$. We allow indices to be negative for notational convenience, and quantities with negative indices are understood as zero. We define $\log_+(x) = 1 + \max(0, \log(x))$ for positive $x$. Unless otherwise stated, the norm $\|\cdot\|$ refers to the Euclidean $\ell_2$ norm for vectors and the corresponding operator/spectral norm for matrices. $\|\cdot\|_F$ denotes the Frobenius norm of a matrix. For $A \in \mathbb{R}^{d\times d}$ complex diagonalizable we let

$$\kappa_{\text{diag}}(A) = \|H\| \cdot \|H^{-1}\|$$

be the condition number of the best similarity transformation $H \in \mathbb{C}^{d\times d}$ for which $A = HDH^{-1}$ with $D$ diagonal, which is sometimes called the "spectral condition number" of $A$. If $A$ is not diagonalizable we set $\kappa_{\text{diag}}(A) \equiv +\infty$, and if $A$ is normal then $\kappa_{\text{diag}}(A) = 1$.

## 4 Algorithm and Main Result

Our algorithm leverages spectral filtering of past outputs to build a predictor. By filtering over past observations $y_{t-1}, y_{t-2}, \ldots$, the algorithm can handle nonlinear dynamics, asymmetric linear dynamics, and achieves robustness to adversarial process noise; this is implemented[2] in Algorithm 1. Our proof techniques also yield learning guarantees for other improper learning algorithms and in the agnostic setting, which we describe in Appendix G for completeness.

Henceforth, for simplicity of presentation we assume that the dimension of the observation is one, i.e. $d_{\mathcal{Y}} = d_{\text{obs}} = 1$; this is the hardest case to learn. The details that change when the observation space is larger are noted in Appendix H.

---

**Algorithm 1** Observation Spectral Filtering + Regression

1: **Input:** Horizon $T$, number of filters $h$, number of autoregressive components $m$, step sizes $\eta_t$, convex constraint set $\mathcal{K}$, prediction bound $R$.
2: Compute $\{(\sigma_j, \phi_j)\}_{j=1}^k$ the top eigenpairs of the $(T-1)$-dimensional Hankel matrix of Hazan et al. (2017).
3: Initialize $J_j^0, M_i^0 \in \mathbb{R}^{d_{out}\times d_{in}}$ and $P_j^0, N_i^0 \in \mathbb{R}^{d_{out}\times d_{out}}$ for $j \in \{1, \ldots, m\}$ and $i \in \{1, \ldots, h\}$. Let $\Theta^t = (J^t, M^t, P^t, N^t)$ for notation.
4: **for** $t = 0, \ldots, T-1$ **do**
5:     Compute

$$\hat{y}_t(\Theta^t) = \sum_{j=1}^{m-1} J_j^t u_{t-j} + \sum_{i=1}^h \sigma_i^{1/4} M_i^t \langle \phi_i, u_{t-2:t-T}\rangle$$

$$+ \sum_{j=1}^m P_j^t y_{t-j} + \sum_{i=1}^h \sigma_i^{1/4} N_i^t \langle \phi_j, y_{t-1:t-T-1}\rangle,$$

and if necessary project $\hat{y}_t(\Theta^t)$ to the ball of radius $R$ in $\mathcal{Y}$.
6:     Compute loss $\ell_t(\Theta^t) = \|\hat{y}_t(\Theta^t) - y_t\|$.
7:     Update $\Theta^{t+1} \leftarrow \Pi_{\mathcal{K}}[\Theta^t - \eta_t \nabla_\Theta \ell_t(\Theta^t)]$.     # can be replaced with other optimizer
8: **end for**

---

Theorem 4.1 states the learning guarantee this algorithm achieves for nonlinear dynamical systems. We will use the construction of the Luenberger program, defined precisely in Definition C.7, with an optimal value $Q_\star$ which depends on the dynamical system. We use the notation that $\widetilde{O}(\cdot)$ hides logarithmic dependence on $T$.

---

[2]To avoid restating many variations of the same algorithm, we state Algorithm 1 over both open-loop controls $u_t$ and observations $y_t$. It is understood that when applying our results to a nonlinear system such as (1.1), since there are no $u_t$'s we take $J^t, M^t \equiv 0$.

**Theorem 4.1** (Nonlinear). *Let $y_1, \ldots, y_T$ be the observations of a nonlinear dynamical system $(f, h)$ satisfying Assumption 3.1. Let $Q_\star = Q_\star \left( f, h, \frac{1}{T^{3/2}}, \left[ 0, 1 - \frac{1}{\sqrt{T}} \right] \cup \{1\} \cup \mathbb{D}_{1-\gamma} \right)$ be as specified in Definition C.7. If $\hat{y}_1, \ldots, \hat{y}_T$ are the predictions made by running Algorithm 1 with $J^t, M^t \equiv 0$ and $\mathcal{K} = R_D$ for $D = \Theta((m + h)Q_\star^2)$, then the following hold:*

(i) *If $h = \Theta(\log^2 T)$ and $m = \Theta\left( \frac{1}{\gamma} \log \left( \frac{T}{\gamma} \right) \right)$ and we use gradient descent with step sizes $\eta_t = \Theta\left( \frac{Q_\star^2}{R\sqrt{t} \log T} \right)$, then*

$$\sum_{t=1}^{T} \|\hat{y}_t - y_t\| \leq \widetilde{O}\left( Q_\star^2 \log(Q_\star) \cdot d_\mathcal{X} R m^2 \cdot \sqrt{T} \right).$$

(ii) *If we instead choose $h = \Theta(\log^2(Q_\star T))$ and $m = \Theta\left( \frac{1}{\gamma} \log \left( \frac{Q_\star T}{\gamma} \right) \right)$ and use the Vovk-Azoury-Warmuth forecaster as the optimizer for the square loss, then*

$$\sum_{t=1}^{T} \|\hat{y}_t - y_t\|^2 \leq \widetilde{O}\left( \log^2(Q_\star) \cdot d_\mathcal{X} R^2 m^2 \cdot \sqrt{T} \right).$$

**Remark 4.2.** *The bound is presented in two forms: in (i) the regret is $\mathrm{poly}(Q_\star)$ for the algorithm when run with $\mathrm{poly} \log(T)$ parameters, whereas in (ii) the regret is $\mathrm{poly} \log(Q_\star)$ at the cost of requiring $\mathrm{poly} \log(Q_\star T)$ parameters. Furthermore, observe that in (i) if $T = \mathrm{poly}\, Q_\star$ then $h = \mathrm{poly} \log(Q_\star T)$ anyway, and so (i) and (ii) describe similar situations. In either case, the algorithm succeeds using $\mathrm{poly} \log(Q_\star T)$ parameters.*

For some intuition, in Lemma F.2 we show that $Q_\star$ is upper bounded by $2^{\widetilde{O}(N)}$, where $N$ is the number of eigenvalues of (a truncation of) the Koopman operator outside $[0, 1 - \frac{1}{\sqrt{T}}] \cup \{1\} \cup \mathbb{D}_{1-\gamma}$. See Remark C.6 for more detail about the relationship with the Koopman operator. We can informally state this result as:

**Corollary 4.3** (Informal). *Let $N$ be the number of marginally-stable complex system eigenvalues of a LDS approximation to the original nonlinear system. Then, $\log(Q_\star) = \widetilde{O}(N)$ and so running Algorithm 1 with $\widetilde{O}(N^2)$ parameters achieves loss*

$$\sum_{t=1}^{T} \|\hat{y}_t - y_t\|^2 \leq \widetilde{O}\left( N^2 \sqrt{T} \right).$$

As we will see through our analysis, the quantity $Q_\star$ yields a natural measure of the complexity for the improper learning of nonlinear dynamical systems with linear predictors, and the notion of "desirable" eigenvalues allows us to specify the analysis to different classes of improper algorithms. In some cases $Q_\star$ can become large, as highlighted by the following examples:

- If $(f, h)$ is a permutation system (i.e. $y_t = v_{t \mod d}$ for a fixed sequence $v_1, \ldots, v_d$), then any linear approximation will have $d$-many complex eigenvalues on the unit circle, which implies complexity $Q_\star = 2^{\widetilde{\Theta}(d)}$ and therefore that Algorithm 1 can learn with a number of parameters polynomial in $N_\star = \Theta(d)$. Note the lower bound of $\Omega(d)$ parameters since any algorithm must somehow memorize the seqence $v_1, \ldots, v_d$.
- At our level of generality, some nonlinear systems are meant to be unlearnable. For example, if $(f, h)$ is constructed to simulate a pseudorandom number generator we can show via an exponential computational lower bound (Theorem E.2). Such systems cannot be efficiently predicted, and this behavior gets captured by the scaling of $Q_\star$.

The quantities $Q_\star$ and $N_\star$ appear to adapt to the hardness of problem; when they are small the algorithm can learn, and they are large in lower bound instances when the algorithm cannot learn. Importantly, there are natural conditions for low complexity, such as detailed balance conditions and phase bounds, which guarantee small $Q_\star$ and demonstrate that Algorithm 1 can efficiently learn many nonlinear systems. We discuss learnability of nonlinear systems and the implications of this complexity measure more in Section 6.

## 5 PROOF OVERVIEW

Our proof of Theorem 4.1 makes strong use of online convex optimization and improper learning/convex relaxation. There are three main steps: (1) linearization, (2) eigenvalue placement/observer systems, and (3) spectral filtering. We defer the proofs to Appendix C and sketch the argument here.

1. For a marginally-stable observable nonlinear system (i.e. $(f, h)$ satisfying Assumption 3.1), we construct a very high-dimensional LDS which approximates the output sequence of the original nonlinear system. This global linearization is achieved via discretized Markov chains or Koopman operators, see Lemma C.5. Via regret bounds, this reduces the original nonlinear sequence prediction task to competing against high-dimensional linear systems.

2. Given a candidate high-dimensional approximating LDS, we use the Luenberger observer construction and the linear algebraic theory of pole placement to construct an equivalent LDS with desirable spectral structure. The cost of doing so is precisely what is measured by $Q_\star$. In other words, $Q_\star$ is the complexity of the least complex high-dimensional linear system with desirable spectral structure which approximates $(f, h)$.

3. In the first two steps, we constructed a high-dimensional LDS with real eigenvalues whose outputs approximate the original system, with $Q_\star$ measuring its complexity. This is only an existence result: it would be unfeasible to try and directly learn this competitor LDS or to implement our construction. Instead, we make use of the framework of online convex optimization Hazan (2016) and regret guarantees like Theorem 1 of Hazan et al. (2017).

Our proof reduces the nonlinear case to the linear one, which it then reduces to improper learning over a spectrally-constrained class of LDS predictors with cost measured by $Q_\star$. Some adjustments to the original spectral filtering proofs are required, such as improving the dependence on initial conditions via a spectral gap and the addition of autoregressive components.

## 6    DISCUSSION

As a review of our nonlinear guarantee, we combined the Luenberger observer with a global linearization to construct a high-dimensional LDS with desirable spectral structure that competes against the original system, and showed that spectral filtering has a regret guarantee against this competitor. For a fixed nonlinear system, we defined the Luenberger program with value $Q_\star(\Sigma)$, which is the best possible condition number such that the constructed observer system has poles contained in $\Sigma$, and derived regret guarantees directly scaling with $Q_\star$.

**Interpretation of loss bounds.**    Suppressing dependence on $R$ and $d_\mathcal{X}$ for exposition, our sharpest analysis yields a regret guarantee of the form

$$\mathrm{Regret}_T(\mathcal{A}, \Pi) \leq \widetilde{O}\left(\mathrm{poly}\log(Q_\star) \cdot \sqrt{T}\right)$$

for $\Pi$ being a class of marginally stable bounded nonlinear predictors, where for $\mathcal{A}$ we run Algorithm 1 with $O(\mathrm{poly}\log(Q_\star, T))$ many parameters and the Vovk-Azoury-Warmuth forecaster as our optimizer (see Theorem G.6).

If $N_\star$ denotes the optimal number of undesirable eigenvalues, minimized over all high-dimensional approximating LDS's, then it can be learned using a linear method such as spectral filtering with $\mathrm{poly}(N_\star)$ parameters and $\mathrm{poly}(N_\star)$ samples. We arrive at the main takeaway of our learning results: if a nonlinear system has some high-dimensional approximating LDS with a certain spectral structure, then it can be efficiently learned by an improper algorithm suited for that structure. With the choice of Algorithm 1 as our improper learning method, we get the statement that "*the complexity of learning nonlinear signals via Algorithm 1 scales with the minmal number of marginally stable complex eigenvalues over all linear approximations*".

**Spectral properties of linearizations.**    Note that the spectral properties of Koopman operators depend on the choice of function space. Though we do not make this precise, by nature of improper learning our analysis implies that if there exists *any* function space on which the (truncation of the) Koopman operator satisfies the above desiderata then spectral filtering will succeed.

For many physical systems $f$ which satisfy some detailed balance condition (e.g. Langevin dynamics/energy minimization, see Section B.4) the Koopman operator can be made self-adjoint, which yields $Q_\star(f, h, \varepsilon, [-1, 1]) = O(1)$ for all $\varepsilon$ and $h$ which are observable. In other words, for many physical systems we truly do expect the existence of a high-dimensional approximating LDS with few marginally stable complex eigenvalues (see Section B.5 for numerics regarding this). Combining this with the previous conclusion, we get the statement that "*Algorithm 1 can learn physical systems provably and efficiently with few parameters*".

**A control-theoretic perspective on learnability.**    High-dimensional LDS's are in a sense universal dynamical computers, since by e.g. Lemma C.5 any reasonable nonlinear system can be approximated to arbitrary accuracy within

this class. As such, it is very natural to define "learnability" of a nonlinear signal in terms of learnability of *some* high-dimensional linear approximation.

Armed with these guiding principles, a central theme in our analysis is the use of control-theoretic tools – Luenberger observer systems and global linearization via Koopman-like methods – not as components to be explicitly designed or learned, but to construct a theoretical benchmark in the form of a high-dimensional LDS predictor that an algorithm should aspire to match. By design, $Q_\star(f, h, \Sigma)$ serves as a fundamental measure of the difficulty of learning a system $f$ from its observations $h$ using an improper algorithm that competes against LDS's with spectrum in $\Sigma$.

This yields a notion of learnability that is tailored to a given improper algorithm. For example, spectral filtering naturally competes against real-diagonalizable LDS's ($\Sigma \subseteq [-1, 1]$, i.e. physical systems with detailed balance) and regression naturally competes against stable LDS's ($\Sigma \subseteq \mathbb{D}_{1-\gamma}$, i.e. systems with an attractive fixed point), and Algorithm 1 demonstrates that we can compose these building blocks to cover a larger spectral region. The ability of these algorithms to learn a nonlinear signal depends sharply on the corresponding $Q_\star$ values of the generating system via our regret bounds, and our results give a way to determine the natural algorithm for a nonlinear system based on its qualitative properties (such as reversibility or strong stability).

**Summary.** To summarize in a few words, the Luenberger program provides a new lens through which to analyze dynamical systems: by characterizing their $Q_\star$ values, one can make principled a priori judgments about their amenability to learning from partial observations with improper methods. Algorithm 1 is a very good general choice for such a method, since it can efficiently and provably learn to predict systems that are either physical or strongly stable. This methodology is universal, since the algorithm can be run without needing to know anything about the system in advance.

# 7 EXPERIMENTS

In this section, we aim to numerically confirm our theoretical results. We investigate the algorithm's performance on a particular nonlinear next-token prediction task. In Appendix B we present some more experiments, such as for asymmetric LDS, the double pendulum, and Langevin dynamics. In addition, in Appendix B we provide some numerical evidence that global linearizations of these nonlinear systems have desirable system eigenvalues and small $Q_\star$.

We run four algorithms for next-token prediction. Two of them correspond to learning a linear predictor over the stream of $y_t$'s via either parameterizing the observer LDS (`LDS`) or spectral filtering as in Algorithm 1 (`SF`). In addition, the classical `eDMD` algorithm for Koopman operator learning is directly parameterizing an LDS over a lifted family of observables, and we introduce `SFeDMD` which learns a spectral filtering predictor over the lifted observables. For details, see Section B.2.

## 7.1 LORENZ SYSTEM

The Lorenz system is a 3-dimensional deterministic nonlinear dynamical system with state $x = [x^{(1)}, x^{(2)}, x^{(3)}] \in \mathbb{R}^3$ evolving according to the coupled differential equations[3]

$$\begin{cases} \dot{x}^{(1)} = \sigma(x^{(2)} - x^{(1)}), \\ \dot{x}^{(2)} = x^{(1)}(\rho - x^{(3)}) - x^{(2)}, \\ \dot{x}^{(3)} = x^{(1)}x^{(2)} - \beta x^{(3)} \end{cases}$$

for fixed constants $\sigma, \rho, \beta$. This was first considered by Edward Lorenz in 1963 as a simplified model for atmospheric dynamics, and since then it has become a standard dynamical system on which to test prediction algorithms. For the choices $\sigma = 10, \rho = 28, \beta = \frac{8}{3}$ (which we make from now on) the dynamics become quite complicated, with chaotic solutions and the existence of a strange attractor of fractal dimension.

For each fixed initial condition $x_0$, the Lorenz system produces a sequence of states $x_1, \ldots, x_T$. We consider two prediction tasks: one in which the algorithm sees full observations $y_t = x_t$, and one in which the algorithm sees partial observations via $y_t = Cx_t$ for $C \in \mathbb{R}^{1 \times 3}$. We run the four algorithms from Section B.2 on both the fully and partially observed versions of the task, with initial conditions and the value of $C$ i.i.d. standard Gaussian. Losses are plotted in Figure 1, averaged over random seeds and smoothed.

---

[3]We simulate this ODE using a simple explicit Euler discretization with $\Delta t = 0.01$ to generate our plots and datasets.

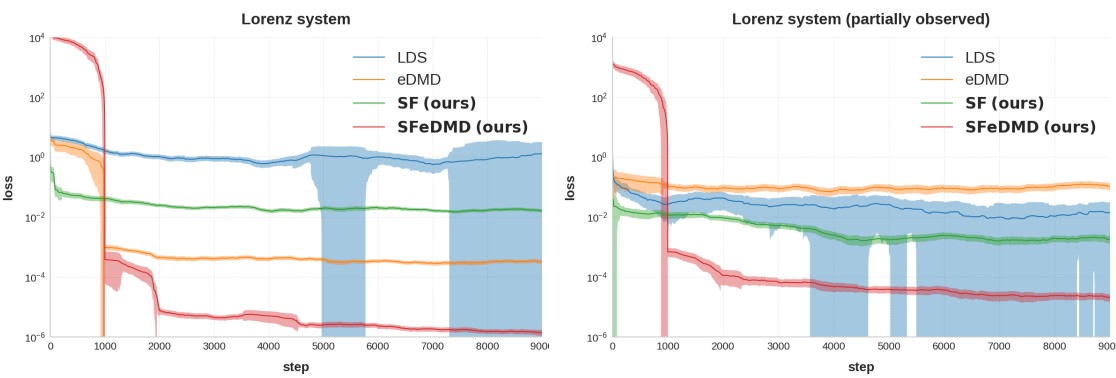

Figure 1: Instantaneous losses $\|\hat{y}_t - y_t\|^2$ plotted on log scale as a function of $t$ for the fully and partially observed Lorenz system, respectively, averaged over 12 random seeds with a smoothing filter of length 1000.

As we see, Algorithm 1 is able to learn successfully in either the partial or full observation setting when filtering over either the original or nonlinearly-augmented observations – this corroborates the conclusion of Theorem 4.1. By contrast, attempting to learn an observer system directly suffers from instability and nonconvexity, making it unfeasible even on this fairly low-dimensional task. Furthermore, we see that spectral filtering over a sequence augmented by nonlinear observables is certainly the best way to go: the `SFeDMD` method is able to outperform `eDMD` with fewer parameters in the fully observed setting, despite the fact that they both make use of the same dictionary of observables. In addition, a filtering-based predictor is able to aggregate information across time in a way that can handle partial observations naturally, whereas `eDMD` cannot[4].

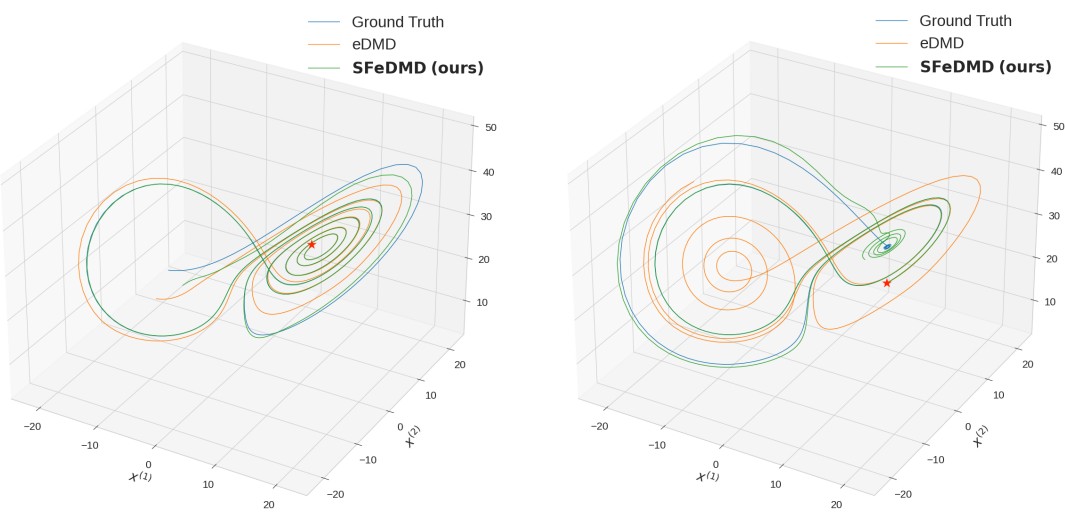

Figure 2: Two sets of autoregressive trajectories of length 512, plotted alongside the ground truth trajectory given by continuing to simulate the Lorenz ODE. The initial positions at which the rollouts start are marked with a red star.

So far, we have investigated the performance of various methods on the task of next-token prediction; after all, this is the loss on which we are training and the regret metric that we use for our theory. While this is by far the most common objective in practice, predictors learned under this loss are often deployed in an autoregressive way[5], such as for text generation in LLMs or rollouts of world models in MPC-type control and planning applications. While our

---

[4]The classical eDMD approach to handle partial observation would be to use time-delayed observables to generate an injective representation of state via Takens' theorem, but (1) anecdotally this is easier said than done since the dynamics of a time-delay lifting are often much more complex and (2) this vastly increases the number of parameters and the complexity of gradient-based optimization.

[5]By autoregression, we mean that there is an initial context $y_1, \ldots, y_L$, which the model uses to produce a prediction $\hat{y}_{L+1}$. In the next iteration, the model uses the context $y_1, \ldots, y_L, \hat{y}_{L+1}$ to predict $\hat{y}_{L+2}$, which in turn gets appended to the context and the

theory does not capture this type of generalization (a predictor which gets very small next-token prediction loss may fail catastrophically in rollout), we find it interesting to inspect the autoregressive behaviors of our learned predictors.

In Figure 2 we plot two autoregressive trajectories of the learned `eDMD` and `SFeDMD` predictors[6] started from contexts unseen during training. On the first one, both methods are able to follow the general trend of the dynamics during the whole rollout, though `SFeDMD`'s predictions remain sharp for longer. On the second trajectory, the errors made by `eDMD` combine with the chaotic nature of this system to produce a qualitatively different rollout, which `SFeDMD` is able to avoid. Of course, the ability of a predictor to generalize from next-token training to autoregressive rollout is heavily dependent on the parameterization, and deep models or different nonlinear liftings likely produce different behaviors. Since autoregressive rollout is not the goal of this paper, we leave a more detailed investigation to future work.

## 8 CONCLUSIONS AND FUTURE WORK

Deviating from the extensive literature on learning in dynamical systems, we have introduced a new approach based on improper learning and spectral filtering. We demonstrated how we can use classical tools from control theory, such as the Koopman operator and the Luenberger program for an observer system, to prove the *existence* of a real-diagonalizable linear dynamical system that is equivalent to the given nonlinear one in terms of observations. We then used the power of spectral filtering over this real-diagonalizable system to learn it provably and efficiently via convex optimization. The new methodology is accompanied by experiments on both linear and nonlinear systems validating the predicted efficiency, robustness, and accuracy, and showing that the regret scaling follows the control-theoretic condition number $Q_\star$.

Unsurprisingly, augmenting the observations nonlinearly improves the performance of an otherwise linear predictor. We suspect that much of the two-part eDMD paradigm (fixing/learning a nonlinear dictionary of observables and learning the lifted dynamics) will benefit from the use of spectral filtering – we propose no changes to the lifting part, but only to learn to predict lifted dynamics via spectral filtering instead of direct LDS parametrization. Deep neural networks with alternating MLP and SSM layers are currently a state of the art for deep learning of dynamical systems, and they may be viewed as jointly learning a nonlinear lifting and (linear) dynamics in the lifted space[7]. The success of the `SFeDMD` algorithm over `eDMD` strongly motivates the use of STU Liu et al. (2024) over direct SSM layers in such arrangements, since we see that spectral filtering is able to make better use of a nonlinear lifting for the task of sequence prediction, with improvement in terms of accuracy, number of parameters, optimization robustness, and no decrease in efficiency[8]. We leave large-scale experimentation on more complex tasks to future work.

On the theoretical side, the most natural next step is to remove the need for a $\frac{1}{\sqrt{T}}$ spectral gap in the observer system (since we don't see its effect much in practice). The work is already done in terms of spectral filtering (we can use the $\rho = 0$ case of Lemma C.4), and the only thing to be done is to construct a high-dimensional LDS which approximates the original nonlinear system for which both $R_C$ and $R_{x_0}$ can be small. We anticipate this to be achievable using the more standard Koopman operator methodology. In addition, one could adapt our analysis to tolerate adversarial disturbances to the nonlinear system by translating them to additive disturbances to the original LDS and then applying our Theorem A.2(c) – with a Koopman eigenlifting, this would require control on the Lipschitz constants of Koopman eigenfunctions, which can become technically involved. We leave these extensions to future work. A more significant extension would be to allow for the presence of open-loop inputs in the nonlinear system – both the Koopman formalism and our discretized Markov chain are not well-adapted to handle non-control-affine systems, and we consider this a valuable direction for future work.

---

process is repeated. In this way, the predictor may be queried with contexts corrupted by its prior mistakes, which allows errors to compound.

[6]The `LDS` and `SF` predictors consistently diverge during rollouts, which may be due to their online training via gradient descent. At each iteration of training the predictor is adapted to the current state and dynamics, and so it may forget about behaviors that it saw earlier in training. For this reason we suspect that training in batch on a dataset of trajectories may fix this.

[7]This is also a current paradigm for robotics control and planning tasks, in which one learns an embedding from e.g. pixel space to latent space together with a "world model" that tracks these latent dynamics.

[8]During inference/rollout, one can use fast online convolution methods such as Agarwal et al. (2024) or LDS distillation methods such as Shah et al. (2025) to achieve the same generation efficiency as directly parameterizing an LDS.

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

# A   LINEAR DYNAMICAL SYSTEMS

Recall that a linear dynamical system (LDS) is parameterized by system matrices $(A, B, C, D)$. For input vectors $u_1, \ldots, u_T \in \mathbb{R}^{d_{\text{in}}}$, the system generates a sequence of output vectors $y_1, \ldots y_T \in \mathbb{R}^{d_{\text{out}}}$ according to equations (2.1), i.e. with dynamics

$$x_{t+1} = Ax_t + Bu_t + w_t, \quad y_t = Cx_t,$$

where $x_0, \ldots, x_T \in \mathbb{R}^{d_h}$ is a sequence of hidden states and $w_t, \xi_t$ are perturbation terms. The problem of *learning* in such systems refers to predicting the next observation $y_t$ given all previous inputs $u_{1:t}$ and past observations $y_{1:t-1}$.

Our proofs for nonlinear systems rely on a reduction to LDS and provable improper learning algorithms via online convex optimization. Linear systems are universal dynamical predictors – as we will see, most dynamical systems have linear approximations, which is what inspires our construction of $Q_\star$ via the complexity of linear approximations. In this section we survey the state of the art in LDS prediction and describe how our results advance it.

## A.1   RELATED WORK

The fundamental problem of learning in linear dynamical systems has been studied for many decades. In complex systems the hidden dimension can be very large, and it is thus crucial to avoid explicit complexity in this parameter. Other sources of difficulty come from the spectral properties of the dynamics matrix $A$ such as its spectral radius, which determines how close it is to marginal stability. Asymmetry of $A$ and the presence of process noise $w$ can further complicate the situation. We briefly survey the extensive and decades-spanning literature on this topic, focusing on the main approaches or techniques.

**System identification.**   System identification refers to the method of recovering $A, B, C$ from the data. This is a hard nonconvex problem that is known to fail if the system is marginally stable, i.e. has eigenvalues that are close to one. Also, recovering the system means time and memory proportional to the hidden dimension, which can be extremely large. Hardt et al. (2018) shows that the matrices $A, B, C$ of an unknown and partially observed stable linear dynamical system can be learned using first-order optimization methods.

In the statistical noise setting, a natural approach for learning is to perform system identification and then use the identified system for prediction. This approach was taken by Simchowitz et al. (2018); Simchowitz & Foster (2020); Sarkar & Rakhlin (2019). The work of Ghai et al. (2020) extends identification-based techniques to adversarial noise and marginally stable systems. The work of Simchowitz et al. (2019) presented a regression procedure for identifying Markov operators under nonstochastic (or even adversarial) noise. This approach has roots in the classical Ho-Kalman system identification procedure HO & Kálmán (1966), for which the first non-asymptotic sample complexity, although for stochastic noise, was furnished in Oymak & Ozay (2019). Nonstochastic control of unknown, unstable systems without any access to prior information, also called black-box control, was addressed in Chen & Hazan (2021).

**Regression a.k.a. convex relaxation.**   The (auto) regression method predicts according to $\hat{y}_t = \sum_{i=0}^m M_i u_{t-i}$. The coefficients $M_i$ can be learned using convex regression, and without dependence on the hidden dimension. The downside of this approach is that if the spectral radius of $A$ is $1 - \delta$, it can be seen that $m \approx \frac{1}{\delta}$ terms are needed. The regression method can be further enhanced with "closed loop" components, that regress on prior observations $y_{t-1:1}$. It can be shown using the Cayley-Hamilton theorem that using this method, $d_h$ components are needed, see e.g. Kozdoba et al. (2019); Hazan et al. (2017).

**Method of moments.**   Hazan et al. (2020) give a method-of-moments system identification procedure as well as the first end-to-end regret result for nonstochastic control starting with unknown system matrices. Extension of these methods to tensors were used in Bakshi et al. (2023a), and in Bakshi et al. (2023b) to learn a mixture of linear dynamical systems.

**Kalman filtering.**   Kalman Filtering Kalman (1960); Anderson & Moore (1991) involves recovering the state $x_t$ from observations. While Kalman filtering is optimal under specific noise conditions, it generally fails in the presence of marginal stability and adversarial noise. In addition, its complexity depends on the hidden dimension of the system.

**Spectral filtering.**   The technique of spectral filtering Hazan et al. (2017) combines the advantages of many previous methods. It is an efficient method whose complexity does **not** depend on the hidden dimension, and works even for marginally stable linear dynamical systems. Various extensions of the basic technique were proposed for learning and control Hazan et al. (2018); Arora et al. (2018); Brahmbhatt et al. (2025), however these do not apply in full generality.

The most significant advancement was the recent result of Marsden & Hazan (2025), who gave an extension to certain asymmetric LDS, those whose imaginary eigenvalues are bounded from above.

The main technical contribution of this paper, from which we derive the guarantees for learning nonlinear dynamics, is a spectral filtering method which circumvents previous difficulties and obtains the best of all worlds – an efficient method whose parametrization does not depend on the hidden dimension, and allows for marginal stability and asymmetry. Furthermore, our method naturally allows handling of adversarial perturbations in the regret minimization sense. Table 1 below summarizes the state of the art in terms of the fundamental problem of learning linear dynamical systems, and how our result advances it.

| Method | Marginally stable | $d_{\text{hidden}}$-free | asymmetric | adversarial noise |
|---|---|---|---|---|
| Sys-ID | $\times$ | $\times$ | $\checkmark$ | $\times$ |
| Regression, open-loop | $\times$ | $\checkmark$ | $\checkmark$ | $\checkmark$ |
| Regression, closed-loop | $\checkmark$ | $\times$ | $\checkmark$ | $\checkmark$ |
| Spectral filtering Hazan et al. (2017) | $\checkmark$ | $\checkmark$ | $\times$ | $\times$ |
| USP Marsden & Hazan (2025) | $\checkmark$ | $\checkmark$ | $\sim \checkmark$ | $\times$ |
| OSF (ours) | $\checkmark$ | $\checkmark$ | $\checkmark$ | $\checkmark$ |

Table 1: Methods for learning LDS.

## A.2 IMPROVED SPECTRAL FILTERING GUARANTEE

As in the nonlinear case, our proof techniques naturally admit a regret-based online guarantee; for simplicity, we will state things in the realizable case now and extend to agnostic results in Appendix G. We define the data generator for the realizable case as follows:

**Assumption A.1** (Linear data generator). *The input/output signal $(u_t, y_t)_{t=1}^T$ is generated by an LDS $(A, B, C, x_0)$ as per equations (2.1) with arbitrary perturbations $w_1, \ldots, w_T$. We assume that:*

*(i) $A$ is diagonalizable with all eigenvalues in the unit disk $\mathbb{D}_1$ and $\kappa_{diag}(A) = 1$[9].*

*(ii) The system is bounded, i.e. $\|B\|_F \leq R_B$, $\|C\|_F \leq R_C$, and $\|x_0\| \leq R_{x_0}$.*

*(iii) For all $t$, the inputs and outputs are bounded by $\|u_t\|, \|y_t\| \leq R$, and $\|w_t\| \leq W$.*

*(iv) The system $(A, C)$ is observable[10].*

Our main technical tool advances the linear case, and is the second step in the argument: a guarantee for improperly learning marginally stable and asymmetric linear dynamical systems with controls and noise. The proof of the following theorem combines eigenvalue placement, Luenberger observer systems, and the complexity $Q_\star$ with the spectral filtering theory for symmetric systems.

**Theorem A.2** (Linear). *Let $y_1, \ldots, y_T$ be the observations of an LDS driven by inputs $u_1, \ldots, u_T$ and disturbances $w_1, \ldots, w_T$ for which Assumption A.1 is satisfied. Fix the number of algorithm parameters $h = \Theta(\log^2 T)$ and $m \geq 1$, and for notation define $\gamma = \frac{\log(mT)}{m}$ and $Q = R_B R_C Q_\star + Q_\star^2$. Set $\mathcal{K} = R_D$ for $D = \Theta((m + h)Q)$ and $\eta_t = \Theta\left(\frac{Q}{R\sqrt{t}\log T}\right)$, and let $\hat{y}_1, \ldots, \hat{y}_T$ be the sequence of predictions that Algorithm 1 produces.*

*(a). In the noiseless setting $w_t \equiv 0$ let $Q_\star = Q_\star(A, C, [0, 1] \cup \mathbb{D}_{1-\gamma})$. Then,*

$$\sum_{t=1}^T \|\hat{y}_t - y_t\| \leq O\left(QRm^2 \log^7(mT) \cdot \sqrt{T} + R_C Q_\star R_{x_0} \cdot (m + \log T)\right).$$

---

[9]For an LDS $(A, B, C)$ with a general diagonalizable $A = HDH^{-1}$, one may consider the equivalent LDS $(D, H^{-1}B, CH)$, in which case the product $R_B R_C$ which appears in our regret bounds scales up by $\kappa_{\text{diag}}(A)$.

[10]This means that the matrix $\mathcal{O} = [C, CA, CA^2, \ldots, CA^{d_{\text{h}}-1}] \in \mathbb{R}^{d_{\text{obs}} d_h \times d_h}$ has rank $d_h$. In the case of 1-dimensional observations, observability is equivalent to invertibility of $\mathcal{O}$, and in general it describes whether the observation space sees all directions of state space. If $A$ has an eigenvalue with multiplicity $> d_{\text{obs}}$, then $(A, C)$ cannot be observable. See Section 10.1 of Hazan & Singh (2025) for more information. The observability assumption can be weakened by ignoring the modes of $A$ corresponding to eigenvalues that don't need to be moved by pole placement – for simplicity, we will assume complete observability.

(b). *In the noiseless setting $w_t \equiv 0$ we may also define $Q_\star = Q_\star \left( A, C, \left[ 0, 1 - \frac{1}{\sqrt{T}} \right] \cup \{1\} \cup \mathbb{D}_{1-\gamma} \right)$, in which case*

$$\sum_{t=1}^{T} \|\hat{y}_t - y_t\| \le O \left( QRm^2 \log_+ (R_C Q_\star R_{x_0}/R) \log^7(mT) \cdot \sqrt{T} \right).$$

(c). *For any $\rho \in (0, \gamma]$, let $Q_\star = Q_\star(A, C, [0, 1 - \rho] \cup \mathbb{D}_{1-\gamma})$. Then, with $R' = R + \frac{W}{\rho}$,*

$$\sum_{t=1}^{T} \|\hat{y}_t - y_t\| \le O \left( Q \cdot R' m^2 \log^7(mT) \cdot \sqrt{T} + \frac{R' \log_+ (R_C Q_\star R_{x_0}/R') + R_C Q_\star \sum_{t=1}^{T-1} \|w_t\|}{\rho} \right).$$

*In all cases, if $N$ denotes the number of eigenvalues outside of the desired region then $Q_\star \le 2^{\widetilde{O}(N)}$.*

The above theorem provides an adaptive guarantee in the form of a data-dependent regret bound for Algorithm 1 run with a universal choice of parameters[11] $m, h$: for example, in result (a) the value of $Q_\star$ measures how easy the signal is to represent via a marginally stable observer system whose eigenvalues are either real or $\widetilde{O}\left(\frac{1}{m}\right)$-stable. If the system originally had a desirable spectral structure then $Q_\star = O(1)$, in which case the regret guarantees are strong. By contrast, if the system has very undesirable spectrum then these guarantees are weaker (since $Q_\star$ may be exponential in the number of undesirable eigenvalues). Furthermore, for the noisy guarantee (c) the predictions made compete with the best choice of $\rho$ in hindsight; if there is significant noise one must take larger $\rho$, whereas if there is almost no noise we can take $\rho$ very small, on the order of $\frac{1}{\sqrt{T}}$. Without knowing any of these considerations, the algorithm does as well as the best possible arrangement. In this way, spectral filtering predicts as well as the best desirable high-dimensional linear observer system for a stream of data, from which Theorem 4.1 follows.

To summarize, our results for linear dynamical systems exhibit the following important properties:

1. **Asymmetric systems.** The spectral filtering guarantee is extended to capture asymmetric systems, without requiring small or clustered phases of complex eigenvalues as in previous work.

2. **No hidden dimension dependence.** There is no explicit[12] dependence on the hidden dimension $d_h$.

3. **Noise robustness.** Previous results in spectral filtering starting from Hazan et al. (2017) exhibit compounding loss from noise (or have dependence on hidden dimension, such as Theorem 2 of Hazan et al. (2018)). This means that in the presence of adversarial noise, the instantaneous prediction error can be as large as $\|\hat{y}_T - y_T\| \approx T\|w\|$. In contrast, Algorithm 1 has $\|\hat{y}_T - y_T\| \lesssim \|w\| + \frac{1}{\sqrt{T}}$. Note that there is a lower bound of $\|\hat{y}_T - y_T\| \gtrsim \|w\|$, see Theorem E.1.

## B   OTHER EXPERIMENTS

In this section, we supplement the experimental results of Section 7. First, we demonstrate that spectral filtering over observations allows for the learning of asymmetric LDS with process noise. Afterwards, we experiment on some more nonlinear sequence prediction tasks.

### B.1   LINEAR SYSTEMS

Recall Algorithm 1, which performs spectral filtering over the observation history $y_t$ as well as the open-loop controls $u_t$. If we run the algorithm with $N^t \equiv 0$ and $m = 1$ then we essentially recover the original vanilla spectral filtering algorithm of Hazan et al. (2017), which has a guarantee for symmetric LDS with no process noise. We will refer to this baseline as SF, and we will denote the full Algorithm 1 by SF+obs. For simplicity, we will only compare these two methods to highlight the contributions of this work; for comparisons between spectral filtering-based algorithms and other LDS learning methods, please see Liu et al. (2024) or Shah et al. (2025).

---

[11]The only way that any information about the data factors into Algorithm 1 is through the optimization, since $Q_\star$ determines the diameter $D$ and learning rates $\eta_t$. If one were to replace OGD with an adaptive/parameter-free optimizer, the algorithm would be fully data-independent.

[12]The hidden dimension does not affect the runtime, number of parameters, or convexity of the optimization in any way. It indirectly affects $Q_\star$ since an LDS with larger $d_h$ has more eigenvalues, and $Q_\star$ scales with the number of eigenvalues that must be moved via pole placement.

Theorem A.2 proves two main benefits of the additional filtering over observations: (1) robustness to adversarial process noise and (2) the ability to learn asymmetric LDS. To demonstrate these advancements, we consider signals $(y_t)_{t=1}^T$ generated by an LDS according to (2.1) in two settings:

(1) $A$ is a Gaussian random $128 \times 128$ matrix, normalized for $\|A\| = 1$, and $w_t = 0.1 \cdot \sin(3\pi t) \cdot [1, \dots, 1] \in \mathbb{R}^{128}$ are correlated sinusoidal disturbances.

(2) $A$ is the $16 \times 16$ cyclical permutation matrix and $w_t = 0$.

In both cases, $d_{in} = d_{out} = 1$, we sample $B, C, x_0, (u_t)_{t=1}^T$ as i.i.d. Gaussians[13], we use the `optax.contrib.cocob` optimizer for simplicity[14], and we run Algorithm 1 with $m = 1$ and $h = 24$. The instantaneous losses are plotted in Figure 3, averaged over random seeds and smoothed.

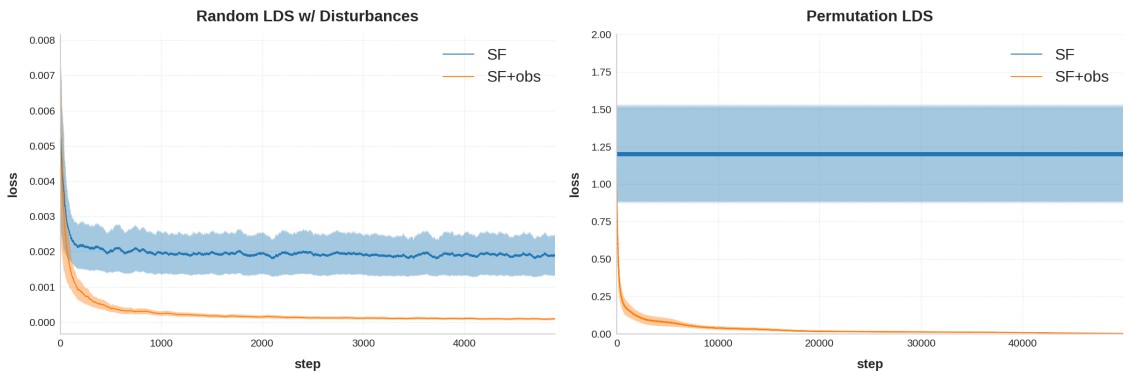

Figure 3: Instantaneous losses $\|\hat{y}_t - y_t\|^2$ plotted as a function of $t$ for experiments (1) and (2), averaged over 12 random seeds with a smoothing filter of length 100 and 1000, respectively.

As expected, filtering over observations improves the tolerance to asymmetry and correlated process noise, and on setup (1) the minimal loss achieved is a fixed deterministic nonzero value determined by the noise structure. On setup (2), we see that vanilla spectral filtering is unable to make any progress for the permutation system (which is the most asymmetric marginally stable LDS); by contrast, the simple addition of filtering over observations enables learning. Furthermore, in (2) the number of steps needed to achieve 0 loss is very large[15], as predicted by the value of $Q_\star$ (see Example F.3). Note that even though the systems in (1) are asymmetric, it is in a much tamer way that results in smaller $Q_\star$ and faster learning. To sum up, the benefits of filtering over observations appear exactly as predicted by Theorem A.2.

## B.2 Nonlinear Experimental Setup

Consider an autonomous nonlinear system, which generates a signal $(y_t)_{t=1}^T$ according to (1.1). The task is next-token prediction, in which we use $y_1, \dots, y_t$ to form predictions $\hat{y}_{t+1}$.

Imagine running Algorithm 1 with $J_t, M_t \equiv 0$, which corresponds to spectral filtering with regression over the observation history. As before, we run this with $k = 24$ and $m = 1$ (yielding $(k + m) \cdot d_y^2$ many parameters), and for optimizer we use either COCOB or ONS with tuned learning rate, depending on computational restrictions. For the following experimental results, we refer to this as SF. As baselines, we consider:

---

[13]For a marginally-stable system like the permutation LDS, if $u_t \sim \mathcal{N}(0, 1)$ then we are expecting $\|y_t\| \asymp \sqrt{t}$. To preserve boundedness of the observations, we decay the size of controls for the permutation experiment.

[14]Since the involved optimizations are convex, these results are robust to optimizer choice. By contrast, learning either of the above problems with another approach such as S4 or Mamba are nonconvex and very dependent on the optimizer used.

[15]Interestingly, for the permutation system there is an initial phase of rapid decrease, followed by a very slow second phase of refinement. Anecdotally, we find that for the first order optimizers we experimented with, the length of the initial phase appears independent of $d_h$ whereas the length of the second phase is governed by $Q_\star$. This indicates that perhaps other optimization algorithms would achieve faster convergence, which we leave to future work.

- Directly learning an observer LDS $(A, B, C, x_0)$ via online gradient descent on the next-token MSE loss, with predictions of the form

$$\hat{y}_{t+1} = \sum_{j=0}^{t} CA^j By_{t-j} + CA^{t+1}x_0.$$

  We refer to this baseline as `LDS`, where we fix $d_h = 64$ to be the hidden dimension of the LDS. This predictor has $d_h^2 + 2d_h \cdot d_{\mathcal{Y}} + d_h$ parameters, and the involved optimization is nonconvex[16].

- Performing an online variant of the eDMD algorithm Williams et al. (2015), which fixes a dictionary $\{f_j\}_{j=1}^n$ of nonlinear observables $f_j : \mathcal{Y} \to \mathbb{R}$, defines the lifted states $z_t = [f_1(y_t), \ldots, f_n(y_t)] \in \mathbb{R}^n$, and attempts to fit a predictor $z_{t+1} = Az_t$, $\hat{y}_t = Cz_t$ for matrices $A \in \mathbb{R}^{n \times n}$, $C \in \mathbb{R}^{d_{\mathcal{Y}} \times n}$. We use the `pykoopman` library Pan et al. (2023) implementation, which fits $A, C$ using least squares linear regression over the history of past observations. For the choice of observables, we let $n = d_{\mathcal{Y}} + 20$, where

$$f_j(y) = \begin{cases} y_j & j = 1, \ldots, d_{\mathcal{Y}} \\ \text{RBF}_j(y) & j = d_{\mathcal{Y}} + 1, \ldots, n \end{cases},$$

  where $\text{RBF}_j$ refers to a radial basis function using thinplate splines with centers fitted to the data[17]. In other words, our nonlinear dictionary concatenates the initial observations with 20 RBF observables. We refer to this baseline as `eDMD`, and fit every $T/10$ steps for efficiency. This method has $n^2 + n \cdot d_{\mathcal{Y}}$ parameters.

- Lastly, we introduce a new method of eDMD combined with spectral filtering, which we call `SFeDMD`. We fix the same set of observables as in the `eDMD` baseline, but instead of fitting $A, C$ directly we apply the predictor from Algorithm 1 (with $k = 24$, $m = 0$) to learn the lifted dynamics. One could learn the parameters $N^t$ with either gradient descent or least squares linear regression: we choose the latter to have a more direct comparison against the `eDMD` baseline. This method has $(k + m) \cdot n \cdot d_{\mathcal{Y}}$ parameters.

To summarize, the algorithms we run consist of `SF` (which directly applies the observation-only version of Algorithm 1 using gradient descent), `LDS` (which attempts to directly learn an observer LDS via gradient descent), `eDMD` (which performs the classical eDMD algorithm via linear regression with a fixed dictionary of nonlinear observables), and `SFeDMD` (which uses the same dictionary of nonlinear observables and learns the dynamics improperly via spectral filtering and linear regression). These four algorithms constitute both proper and improper learning methods on both direct observations and nonlinear liftings. Rather than using online optimization algorithms for all methods, we choose to use regression for both eDMD baselines in order to directly apply the `pykoopman` library and for more direct comparison against eDMD experimental results of other works.

## B.3 Double Pendulum

A nonlinear system of physical interest is the double pendulum (a pendulum with two joints), which also exhibits a chaotic behavior and can have complicated marginally stable dynamics. We use the implementation in the Brax Freeman et al. (2021) library, which is a `jax`-based physics simulator – this double pendulum is attached to a cart that can move horizontally on a rail, and it takes inputs which push the cart to the left or right. A fixed deterministic agent/policy operating in such an environment yields a deterministic nonlinear dynamical system which produces a sequence of states $x_0, \ldots, x_T$; for this environment, we define the state to be (the sines and cosines of) the angles and the angular velocities of both joints, together with the position and velocity of the cart to total $d_{\mathcal{X}} = 8$. We consider a fixed linear feedback controller of the form $u_t = Kx_t$ for a fixed $K \in \mathbb{R}^{1 \times 8}$ with i.i.d. standard Gaussian entries. As before, we can construct the sequence prediction task with either full observations $y_t = x_t$ or partial observations $y_t = Cx_t$ for a fixed $C \in \mathbb{R}^{1 \times 8}$ with i.i.d. standard Gaussian entries. Losses for both settings are plotted in Figure 4, averaged over random seeds and smoothed.

This is a more difficult system than the Lorenz attractor (since a predictor needs to also learn the effects of $K$). Despite this, we see a similar story as in the Lorenz system: with full observations both `eDMD` and `SF` are able to predict well, and `SFeDMD` is able to predict very well. Furthermore, `SF` and `SFeDMD` are able to succeed when given partial observations, which `eDMD` struggles with. The `LDS` baseline is brittle – on some trajectories it achieves the same performance as `SF`, but on others it can completely diverge even at reasonable learning rates.

---

[16]In fact, the `LDS` baseline tends to blow up unless trained with the Adam optimizer with tuned learning rate, which we use for this baseline throughout the nonlinear experiments.

[17]This choice is described in Section 3 of Williams et al. (2015), and is also used in the eDMD tutorials for the `pykoopman` library, available online. Furthermore, this dictionary seemed to be a good choice uniformly over all our tasks.

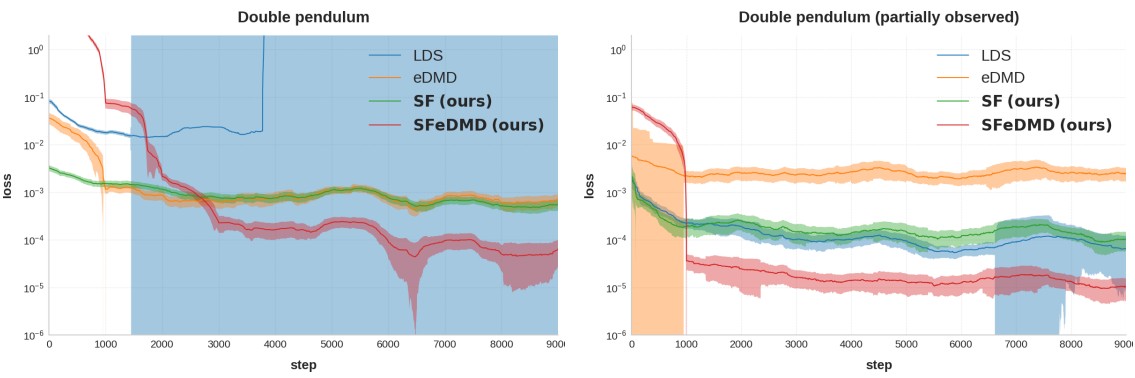

Figure 4: Instantaneous losses $\|\hat{y}_t - y_t\|^2$ plotted on log scale as a function of $t$ for the fully and partially observed double pendulum, respectively, averaged over 12 random seeds with a smoothing filter of length 1000.

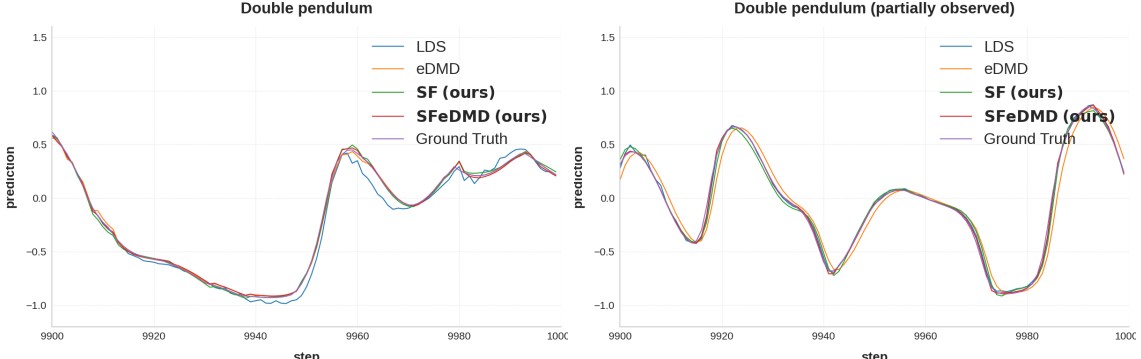

Figure 5: Predictions $\hat{y}_t$ of the four algorithms along with the ground truth $y_t$ plotted as a function of $t$ for the fully and partially observed double pendulum, respectively, on a single random seed. We zoom in on the last 100 steps of the trajectory. For the full observation case, since we cannot plot the full state we choose to plot the sine of the angle between the two arms of the pendulum, which is a representative coordinate for the task of next-state prediction.

To add a bit more clarity to the behaviors of these algorithms, we plot in Figure 5 the actual predictions $\hat{y}_t$ made during training at the end of a typical trajectory, along with the ground truth signal $y_t$. We see in the full observation setting that the pendulum is swinging irregularly in a way that the models can predict with reasonable accuracy (except for `LDS`, which suffers from unstable optimization). In the partial observation setting the algorithms all do quite well, but we do get to see some differences between the algorithms – `eDMD` is unable to be very precise due to the partial observation setting and appears to be predicting $\hat{y}_{t+1} = y_t$, whereas `SFeDMD` produces very precise predictions.

## B.4 LANGEVIN DYNAMICS

As another autonomous nonlinear system, we consider stochastic gradient flow/Gibbs energy minimization/Langevin dynamics. For a clear and in-depth description of this dynamical system from a stochastic differential equation (SDE) perspective, see Section 1 of the textbook Chewi (2025). Given a differentiable loss/energy/potential function $V : \mathbb{R}^{d_x} \to \mathbb{R}$, one can define a dynamical system according to the SDE

$$dX_t = -\nabla V(X_t)dt + \sqrt{2}dW_t, \quad X_0 = x_0,$$

which produces a random sequence of states $X_1, \ldots, X_T$ (we use capital letters here to denote random variables). This is very common in the physical sciences and more recently in machine learning, and can be interpreted through many different perspectives:

- A first perspective is to interpret this as a **gradient descent**: we can imagine a discretized version via[18]

$$X_{t+1} = X_t - \eta \nabla V(X_t) + \sqrt{2\eta} w_t,$$

where $\eta$ is a small step size and $w_t \sim \mathcal{N}(0,1)$ is an independent noise term that can be viewed as a stochastic gradient. In this way, the states correspond to the iterates of a first-order optimization algorithm.

- Under very mild assumptions, the distribution of $X_T$ for very large $T$ will approach the probability distribution $\pi$ on $\mathbb{R}^{d_{\mathcal{X}}}$ with (unnormalized) density $e^{-V}$. A particular trajectory can be viewed as a way to transform $x_0$ into a sample from $\pi \propto e^{-V}$, which yields a **sampling** perspective.

- Langevin diffusions arise naturally in thermodynamics and statistical physics, in which case $V$ is the energy of a particle/configuration – it is known that the equilibrium distribution (at least at temperature 1) will be the stationary distribution $\pi \propto e^{-V}$, which is sometimes also called the Gibbs distribution. The evolution of the physical system will follow these trajectories, yielding a **statistical physics** perspective.

In any case, it is clear that prediction of such systems is a very useful task. The main difficulty is this: it can take very long for the system to equilibrate, and the transient/nonequilibrium dynamics are often the most useful (predicting optimization trajectories becomes useless if we wait until convergence). For some intuition, the Koopman operator[19] may have multiple marginally stable modes, in which case there is slow convergence (i.e. no strong mixing), and in such cases prediction is difficult. Under extra assumptions (strong convexity of $V$, a Poincare inequality for $\pi$, etc.) we would have exponentially-fast convergence mixing, but the general case is most interesting.

A crucial property of such SDEs is that they satisfy a detailed balance condition, and so the Koopman operator is a self-adjoint operator on $L^2(\pi)$ (see Remark 1 of Kostic et al. (2022)). In our language, this implies that **there exists a well-conditioned high-dimensional symmetric LDS approximating the dynamics** – this is very good news for us, since it implies a small $Q_\star$ and so strong performance of Algorithm 1 due to Theorem 4.1.

To make the prediction problem difficult, we consider a potential on $\mathbb{R}^{d_{\mathcal{X}}}$ with exponentially-many asymmetric wells along each coordinate and a nontrivial coupling between coordinates, given by

$$V(x) = \sum_{j=1}^{d_{\mathcal{X}}} \left( 0.05 \cdot x_j^4 - x_j^2 + 0.1 \cdot x_j \right) - 0.2 \cdot \sum_{1 \le i < j \le d_{\mathcal{X}}} x_i^2 x_j^2.$$

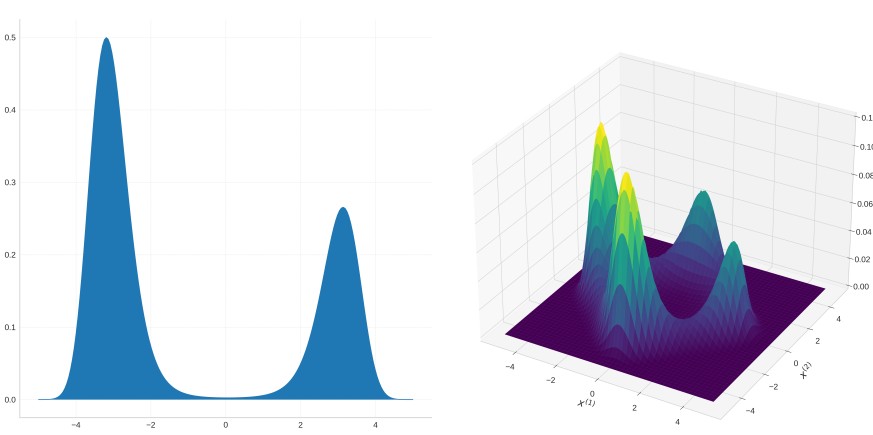

Figure 6: Densities of the stationary distribution $\pi$ corresponding to the chosen potential $V$ for $d_{\mathcal{X}}$ equal to 1 and 2, respectively. We see a number of asymmetric wells that is exponential in the dimension.

For convenience, we plot the density of the stationary distribution $\pi \propto e^{-V}$ in one and two dimensions in Figure 6. In Figure 7 we plot the training losses for the $d_{\mathcal{X}} = 64$ instantiation of the problem, averaged over random seeds and smoothed – we find that the `LDS` method consistently fails in the same way, suggesting some obstacle for learning an

---

[18]This simple integrator with $\eta = 0.01$ is how we will numerically simulate SDEs for our experiments.

[19]Technically one would define the Koopman operator as a family $K_t$ sending $h(x) \to \mathbb{E}[h(X_t) \mid X_0 = x]$, and the derivative of $K_t$ is the generator of the SDE. The main story is the same – approximating the nonlinear dynamics by infinite-dimensional linear ones.

observer system via gradient descent on this task. By contrast, the other methods do well, and in this setting we find that the addition of the nonlinear dictionary offers no help. The minimal loss value reached is consistent with the size of the noise at each step (which is typically of size 0.02), agreeing with the lower bound of Theorem E.1. We anticipate more interesting behaviors for more complicated Langevin dynamics, experimentation on which we leave for future work.

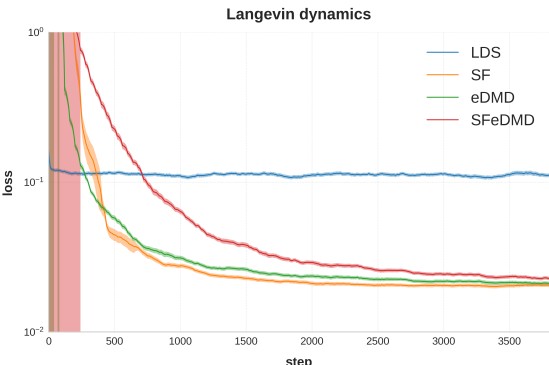

Figure 7: Instantaneous losses $\|\hat{y}_t - y_t\|^2$ plotted on log scale as a function of $t$ for the Langevin dynamics, averaged over 12 random seeds with a smoothing filter of length 200.

### B.5 NUMERICAL TESTS

Thus far, we have demonstrated the behaviors of Algorithm 1 on various nonlinear systems and compared to some simple baselines; we saw that spectral filtering over observations seems to succeed without much issue, and without the very slow optimization behaviors that were present in the permutation system (recall Figure 3). There is the intuition developed in the previous section that physical systems ought to have (approximately) real Koopman spectrum and therefore should be amenable to learning via spectral filtering. However, it would be nice to probe this question numerically, which we will quickly do in this section.

A classical approach to estimating Koopman eigenvalues of a nonlinear system is to run the eDMD algorithm as we have been doing, in which one learns a linear dynamics $A$ on a lifted space. By inspecting the eigenvalues of $A$, we gain understanding of the structure of (a finite-dimensional approximation to) the Koopman operator[20]. We take the eDMD models[21] trained on the full observation versions of our three nonlinear systems and plot the learned eigenvalues in Figure 8. As we can see, on these marginally stable physical systems the Koopman eigenvalues appear as expected: approximately real and near 1.

In addition, this numerical experiment allows us to compute an estimate for the spectral gaps: if we sort $1 = |\lambda_1| > |\lambda_2| \geq \ldots$, the spectral gap is given by $\rho = 1 - |\lambda_2|$. We give the empirical spectral gaps for these systems below:

$$\rho_{\text{Lorenz}} = 0.00168, \quad \rho_{\text{pendulum}} = 0.00477, \quad \rho_{\text{Langevin}} = 0.000694.$$

The mixing time of a system scales as $\frac{1}{\rho}$, and our regret bound of Theorem 4.1 implies that after $\frac{1}{\rho^2}$ steps we get vanishing prediction error. For the above systems, the mixing time is around 1000 steps and the learning horizon is on the order of hundreds of thousands or millions of steps; however, we see from our experiments that Algorithm 1's learning begins immediately. In other words, *spectral filtering appears able to predict transient dynamics*. This indicates that the dependence of our analysis on a $\frac{1}{\sqrt{T}}$ spectral gap is not sharp, and leaves room for theoretical improvement.

## C ANALYSIS

In this section we perform our main analysis. We first detail our construction of the Luenberger program and $Q_\star$ and its implications for learning linear systems, culminating in a proof of Theorem A.2. We then describe a reduction from nonlinear systems to high-dimensional linear systems via discretization, from which Theorem 4.1 follows.

---

[20]Technically, this spectral relationship is only proven under the assumption that the dictionary of nonlinear observables spans an invariant subspace of the Koopman operator. This assumption is required for most eDMD-based methods to work at all, and inspecting the eigenvalues of the eDMD model is commonplace in practice regardless. See Section 5.1 of Brunton et al. (2021).

[21]One could attempt to do this with the observer systems learned directly via gradient descent in the `LDS` method. These appear to be placed generically in the unit disk, though this result is inconsistent and brittle, varying significantly across seeds. By contrast, `eDMD` learns the same system eigenvalues across seeds, which are the ones we report.

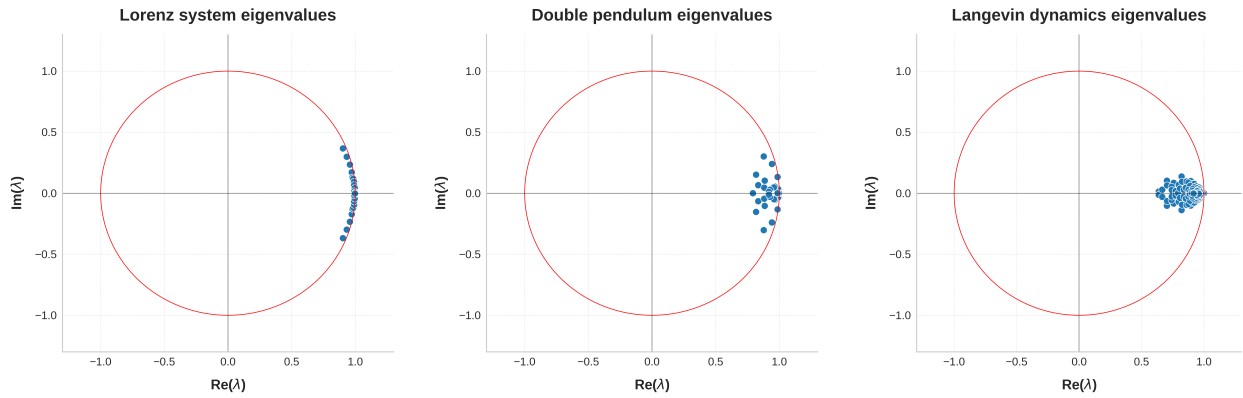

Figure 8: Eigenvalues of the lifted linear dynamics learned by the eDMD algorithm on the Lorenz system, double pendulum, and Langevin dynamics, respectively. The $x$ and $y$ axes display real and complex components, respectively, and we draw the unit circle in red for the reader's convenience.

## C.1 REGRET

The prediction notion we use is borrowed from the setting of online convex optimization Hazan (2016), where the performance of an algorithm is measured by its regret. Regret is a robust performance measure, as it does not assume a specific generative or noise model. It is defined as the difference between the loss incurred by our predictor and that of the best predictor in hindsight belonging to a certain class, i.e.

$$\text{Regret}_T(\mathcal{A}, \Pi) = \sum_{t=1}^{T} \ell(y_t, \hat{y}_t^{\mathcal{A}}) - \min_{\pi \in \Pi} \sum_{t=1}^{T} \ell(y_t, \hat{y}_t^{\pi}).$$

Here $\mathcal{A}$ refers to our predictor, $\hat{y}_t^{\mathcal{A}}$ is its prediction at time $t$, and $\ell$ is a loss function. Further, $\pi$ is a predictor from the class $\Pi$, for example all predictions that correspond to noiseless linear dynamical systems of the form $\hat{y}_{t+1}^{\pi} = \sum_{i=0}^{t} CA^i Bu_{t-i}$. A stronger predictor class is that of all closed-loop linear dynamical predictors that also take into account the initial state and noise, even though they are not observed by an algorithm, i.e.

$$\hat{y}_{t+1}^{\pi} = CA^{t+1}x_0 + \sum_{i=0}^{t-1} CA^i(Bu_{t-i} + w_{t-i}).$$

## C.2 LINEAR DYNAMICAL SYSTEMS

In this subsection, we prove Theorem A.2. Given a linear dynamical system (LDS), we construct another LDS that incorporates past observations as feedback, has a desirable transition matrix, and generates the same trajectories as the original system. This construction relies on the **Luenberger observer** framework Luenberger (1971), which we sketch next.

For a given LDS $(A, B, C, x_0)$, we can always choose a gain $L \in \mathbb{R}^{d_h \times d_{\text{out}}}$ and construct the observer system

$$\tilde{x}_{t+1} = A\tilde{x}_t + Bu_t + L(y_t - \tilde{y}_t), \quad \tilde{y}_t = C\tilde{x}_t, \quad \tilde{x}_0 = x_0.$$

This system has the same dynamics as the original one, but with an additional term that uses the observation error $y_t - \tilde{y}_t$ as linear feedback. Since $y_t = Cx_t$ and $\tilde{y}_t = C\tilde{x}_t$, if we define the state error $\xi_t := x_t - \tilde{x}_t$ then we get

$$\xi_{t+1} = Ax_t + Bu_t + w_t - A\tilde{x}_t - Bu_t - LC(x_t - \tilde{x}_t)$$
$$= (A - LC)\xi_t + w_t.$$

Rolling this out reveals that for all $t$,

$$x_t = \tilde{x}_t + (A - LC)^t(x_0 - \tilde{x}_0) + \sum_{t=1}^{t} (A - LC)^{i-1} w_{t-i}.$$

It is a known fact in control theory that if $(A, C)$ is observable then we can choose $L$ to place the eigenvalues of $A - LC$ as desired, which we describe further in Appendix F. This is a standard construction which is frequently used with

system identification for prediction and control from partial observations Shastri et al. (2023), and the Kalman filter is related to a special case in which the gain $L$ is chosen optimally under Gaussian noise assumptions Kalman (1960). Our approach will be to predict via spectral filtering, which learns improperly and competes in hindsight against all observer systems.

For certain systems $(A, C)$ it is harder to infer hidden state information from observations, which is reflected in terms of the size of $L$ and the complexity of the observer dynamics $A - LC$. Many Luenberger observer-based methods which require $L$ explicitly end up approaching this through heuristics or structural assumptions Peralez & Nadri (2021); Brivadis et al. (2019) or by performing a numerically-difficult optimization over $L$ Pandey et al. (2014). However, since spectral filtering matches the performance of the *best* observer system in hindsight, we will make use of a quantity that captures the complexity of the best observer system:

**Definition C.1** (Luenberger program). *Let $A \in \mathbb{R}^{d_h \times d_h}$ and $C \in \mathbb{R}^{d_{out} \times d_h}$. For each spectral constraint set $\Sigma \subseteq \mathbb{C}$ we consider the optimization problem over $L \in \mathbb{R}^{d_h \times d_{out}}$ given by*

$$\min \quad \kappa_{diag}(A - LC) \quad s.t. \quad A - LC \text{ has all eigenvalues in } \Sigma.$$

*When the constraint set of this optimization is nonempty, denote the minimal value by $Q_\star = Q(A, C, \Sigma) < \infty$ and a minimizing gain by $L_\star$.*

We will be applying Algorithm 1 to improperly learn over the class of observer systems. As we will see in Lemma C.4, spectral filtering does well for $\Sigma \subseteq [0, 1]$ and very well for $\Sigma \subseteq [0, 1 - \rho] \cup \{1\}$[22], and the regression terms are able to handle $\Sigma \subseteq \mathbb{D}_{1-\gamma}$, so we will be mostly interested in $\Sigma = [0, 1 - \rho] \cup \{1\} \cup \mathbb{D}_{1-\gamma}$. For generality, we will state things for arbitrary $\Sigma$ and specify where needed.

In Appendix F we will say more about this optimization and apply known bounds on $Q_\star$, and in particular we show that for observable systems $Q_\star \leq (\frac{N}{\text{vol}(\Sigma)})^{O(N)}$, where $N$ is the number of system eigenvalues outside the desired region $\Sigma$. This can be tight.

**Remark C.2.** *The objective function $Q_\star$ is related to the quantity $S_2$ from Mehrmann & Xu (1998), which describes the stability of eigenvalue placement. For this reason, such minimizations are referred to as "robust pole placement", see also the survey Pandey et al. (2014). It is known that the eigenvalue placement problem is very ill-conditioned, and small perturbations to the data $(A, C)$ yield $Q_\star$-sized changes in $L$ and the placed eigenvalues (see e.g. Theorem 3.1 of Mehrmann & Xu (1996)). If $Q_\star$ is large, then a pipeline based on eigenvalue placement combined with system identification from data can be very brittle.*

In words, the value of $Q_\star$ describes how hard it is (measured in terms of conditioning of the closed-loop dynamics) to approximately recover the state using a desirable observer system[23]. We can describe the observer system construction in terms of $Q_\star$:

**Definition C.3** (Observer system). *Consider the following LDS*

$$x_{t+1} = Ax_t + Bu_t + w_t, \quad y_t = Cx_t.$$

*Fix $\Sigma \subseteq \mathbb{D}_{1-\delta}$ for some $\delta \in [0, 1]$, let $Q_\star = Q(A, C, \Sigma)$ be the value of the corresponding Luenberger program from Definition C.1, and suppose that $Q_\star < \infty$. Then, there exists an observer system which takes as inputs $(u_0, \ldots, u_t)$ and $(y_0, \ldots, y_t)$ and produces the estimated observation $\tilde{y}_{t+1}$ via the LDS*

$$\tilde{x}_{t+1} = \tilde{A}\tilde{x}_t + Bu_t + Ly_t, \quad \tilde{y}_t = C\tilde{x}_t, \quad \tilde{x}_0 = x_0$$

*with the properties:*

*(i) $\|L\| \leq \frac{2Q_\star}{\sigma_{min}(C)}$, where the denominator is the minimal singular value.*

*(ii) $\tilde{A} = HDH^{-1}$ where $D$ is a diagonal matrix with entries in $\Sigma$ and $\kappa_{diag}(\tilde{A}) \equiv \|H\| \|H^{-1}\| = Q_\star$.*

*(iii) $\sum_{t=1}^{T} \|y_t - \tilde{y}_t\| \leq \frac{Q_\star \|C\|}{\delta} \left( \sum_{t=0}^{T-1} \|w_t\| \right)$, and if $w_t \equiv 0$ then $y_t = \tilde{y}_t$ even if $\delta = 0$.*

---

[22]Negative eigenvalues can be handled by spectral filtering easily with a constant amount of extra parameters and computation, see the extension in Theorem 3.1 of Agarwal et al. (2023). For ease of presentation, we will continue to state things for nonnegative real eigenvalues.

[23]Other improper algorithms will require a definition of $Q_\star$ tailored to their structure, see Appendix G. To maintain generality we will often use the phrases "desirable spectral structure" or "undesirable eigenvalues", where the implied spectral constraint set $\Sigma$ will depend on the algorithm.

Let $L_\star = L(A, C, \Sigma) \in \mathbb{R}^{d_h \times d_{\text{out}}}$ be a solution of the Luenberger program. Properties (i) and (ii) follow from the objective function and constraint set of the Luenberger program, along with Lemma F.2. As mentioned previously, if we define the errors $\xi_t := x_t - \tilde{x}_t$ then $\xi_t = \sum_{i=1}^{t} (A - L_\star C)^{i-1} w_{t-i}$ since $\xi_0 = 0$; so, in the noiseless setting we exactly reproduce the states and observations. In the noisy case, since $y_t - \tilde{y}_t = C\xi_t$ and all eigenvalues of $\tilde{A} = A - L_\star C$ have magnitude $\leq 1 - \delta$,

$$\sum_{t=1}^{T} \|\tilde{y}_t - y_t\| \leq \|C\| \, Q_\star \sum_{t=1}^{T} \sum_{i=1}^{t} (1 - \delta)^{i-1} \|w_{t-i}\| \leq \frac{\|C\| \, Q_\star}{\delta} \cdot \left( \sum_{t=0}^{T-1} \|w_t\| \right)$$

The observer system construction shows that there is an LDS over $(u, y)$ with desirable eigenstructure that nearly matches the original one, which may be rolled out without requiring $w$ (without noise, the observer system matches it exactly). We will combine this with spectral filtering's regret guarantee, stated below for the unsquared $\ell_2$ loss (with immediate extension to Lipschitz losses – a similar result can be proven for the squared loss). In addition to the inclusion of autoregressive terms in the algorithm, there is also a sharpening of the analysis that may be of independent interest, see Remark D.2. In the application of the below lemma, it should be imagined that the inputs $u_t$ are actually the stacked $[u_t, y_{t-1}]$ and we are running the algorithm with $P^t, N^t \equiv 0$ in order to compete against an observer LDS.

**Lemma C.4.** *On any sequence $(u_t, y_t)_{t=1}^{T}$ such that $\|u_t\|, \|y_t\| \leq R$, Algorithm 1 run with $P_1^t = 1$ and $P_j^t, N^t \equiv 0$ for $j > 1$, $h = \Theta(\log^2 T)$, $m = \Theta\left(\frac{1}{\gamma} \log\left(\frac{T}{\gamma}\right)\right)$, $\mathcal{K} = B_D$ with $D = \Theta((m + h) R_B R_C \kappa)$, and $\eta_t = \Theta\left(\frac{R_B R_C \kappa}{R\sqrt{t} \log T}\right)$, achieves regret*

$$\sum_{t=1}^{T} \|\hat{y}_t - y_t\| - \min_{\pi \in \Pi} \sum_{t=1}^{T} \|\hat{y}_t^\pi - y_t\| \leq O\left( \frac{\log^7\left(\frac{T}{\gamma}\right)}{\gamma^2} \cdot R_B R_C \kappa R \cdot \sqrt{T} + \frac{R + \log_+(R_C \kappa R_{x_0}/R)}{\rho} \right),$$

*where $\Pi = LDS(\rho, \gamma, \kappa, R_B, R_C, R_{x_0}, R)$ is the class of LDS predictors $\pi$ parameterized by $(A, B, C, x_0)$ where $A$ is diagonalizable with eigenvalues in $[0, 1 - \rho] \cup \{1\} \cup \mathbb{D}_{1-\gamma}$ for $\rho \leq \gamma$, $\kappa_{diag}(A) \leq \kappa$, $\|B\|_F \leq R_B$, $\|C\|_F \leq R_C$, $\|x_0\| \leq R_{x_0}$, and $\|\hat{y}_t^\pi\| \leq R$. If the spectral gap $\rho = 0$, then the second term can be replaced by $R_C \kappa R_{x_0} \cdot \left(\frac{1}{\gamma} + \log T\right)$.*

*Proof.* See Appendix D. □

*Proof of Theorem A.2.* Suppose that the data is generated by a ground truth LDS $(A, B, C, x_0)$ with disturbances $w_t$ such that Assumption A.1 is satisfied. Observability implies that $Q_\star(A, C, \Sigma) < \infty$ by Lemma F.1, where we only move those eigenvalues outside of $\Sigma$. Let $(\tilde{A}, \tilde{B}, C, x_0)$ be the observer system constructed in Definition C.3, which we may treat as an LDS over concatenated inputs $[u_t, y_{t-1}]$ with a block-diagonal input gain $\tilde{B} = \begin{bmatrix} B & 0 \\ 0 & L \end{bmatrix}$. For $d_\mathcal{Y} = 1$ this will have the bounds $\kappa_{\text{diag}}(\tilde{A}) \leq Q_\star$ and $R_{\tilde{B}} = O(R_B + \frac{Q_\star}{R_C})$.

For (c) we use the given $\rho$ and $\gamma = \frac{\log(mT)}{m}$ (so that $\frac{1}{\gamma} \log(\frac{T}{\gamma}) \leq m$) and define $\Sigma = [0, 1 - \rho] \cup \mathbb{D}_{1-\gamma} \subseteq \mathbb{D}_{1-\rho}$. This is in the comparator class $\Pi = \text{LDS}\left(\rho, \gamma, Q_\star, R_B + \frac{Q_\star}{R_C}, R_C, R_{x_0}, R + \frac{W}{\rho}\right)$, and so the error bound of the observer system implies

$$\min_{\pi \in \Pi} \sum_{t=1}^{T} \|\hat{y}_t^\pi - y_t\| \leq \frac{R_C Q_\star}{\rho} \left( \sum_{t=0}^{T-1} \|w_t\| \right).$$

The result follows from directly applying the spectral filtering guarantee of Lemma C.4.

For (b) we take $\Sigma = [0, 1 - \rho] \cup \{1\} \cup \mathbb{D}_{1-\gamma}$ with $\rho = \frac{1}{\sqrt{T}}$. The addition of the $\{1\}$ eigenvalue does not change the comparator class, and so we may apply the result of (c). Result (a) is proved the same way, using the $\rho = 0$ case of Lemma C.4. □

### C.3 NONLINEAR DYNAMICAL SYSTEMS

In this subsection, we will prove Theorem 4.1. Our main tool is a space-discretized Markov chain argument that achieves the same goal as the Koopman operator, which may be of independent interest. The discretization argument is sufficient for sequence prediction and avoids traditional compactness and spectral assumptions, replacing them by simpler boundedness and Lipschitz conditions on the dynamics.

**Lemma C.5.** *Let $y_1, \ldots, y_T$ be generated by a deterministic discrete-time dynamical system $(f, h, x_0)$ satisfying Assumption 3.1. Let $\mathcal{S}$ be an $\varepsilon$-net of the ball of radius $R$ in $\mathcal{X}$. The cardinality of $\mathcal{S}$ satisfies*

$$N = |\mathcal{S}| \leq \left( \frac{2R}{\varepsilon} \right)^{d_{\mathcal{X}}}.$$

*Then, there exists an LDS of hidden dimension $N$ with states $z_t \in \mathbb{R}^N$ and an observation matrix $C' \in \mathbb{R}^{d_{\mathcal{Y}} \times N}$ with Frobenius norm $\|C'\|_F \leq R\sqrt{N d_{\mathcal{Y}}}$, such that the output sequence $y_t' = C' z_t$ satisfies*

$$\sum_{t \leq T} \|y_t' - y_t\| \leq \frac{T^2}{2} \varepsilon.$$

*Proof.* The $\varepsilon$-net condition guarantees that for every $x \in \mathcal{X} \cap B_R$ there exists $s \in \mathcal{S}$ such that $\|x - s\| \leq \varepsilon$. It is standard that such an $\varepsilon$-net can be chosen with $|\mathcal{S}| = N$, and we enumerate the elements as $s^{(1)}, \ldots, s^{(N)}$.

Define the mapping $\pi : \mathcal{X} \to \mathcal{S}$ as the nearest neighbor projection, i.e. $\pi(x) = \arg\min_{s \in \mathcal{S}} \|x - s\|$. We consider the deterministic transition operator $T : \mathcal{S} \to \mathcal{S}$ as $T(s) = \pi(f(s))$, yielding a Markov chain with states $s_t$ in a finite state space. If we let $z_t \in \mathbb{R}^N$ be the indicator vectors corresponding to this state and the evolution operator $A' \in \mathbb{R}^{N \times N}$ with $A'_{ij} = 1$ if $T(s^{(j)}) = s^{(i)}$ and 0 else, then we get a linear evolution

$$z_{t+1} = A' z_t, \quad y_t' = C z_t$$

with $C \in \mathbb{R}^{d_{\mathcal{Y}} \times N}$ sending the one-hot vector $z$ corresponding to $s^{(j)}$ to $h(s^{(j)})$. Since $\|y_t\| \leq R$, the matrix elements of $C'$ can be taken to be bounded by $R$.

With these definitions in place, let $x_0, \ldots, x_T$ be a trajectory of the original nonlinear system with observations $y_0, \ldots, y_T$. We let $s_0 = \pi(x_0)$ and $s_{t+1} = T(s_t)$ be the rollout of the finite state space Markov chain, and define $\xi_t = \|s_t - x_t\|$ the error against the true states. Then,

$$
\begin{aligned}
\xi_{t+1} &= \|s_{t+1} - x_{t+1}\| \\
&= \|\pi(f(s_t)) - f(x_t)\| \\
&\leq \|f(s_t) - f(x_t)\| + \|f(s_t) - \pi(f(s_t))\| \\
&\leq \xi_t + \varepsilon,
\end{aligned}
$$

where we used 1-Lipschitzness of $f$ and the $\varepsilon$-net condition in the last line. Since $\xi_0 \leq \varepsilon$ again by the $\varepsilon$-net construction, we find that $\xi_t \leq t\varepsilon$ and so $\|y_t' - y_t\| \leq t\varepsilon$ by Lipschitzness of $h$. $\square$

This essentially says that a bounded and marginally stable nonlinear dynamical system can be approximated via a $T^{O(d_{\mathcal{X}})}$-dimensional LDS over a horizon of length $T$. An alternate approach to approximating nonlinear systems via high-dimensional linear ones is to use a spectral truncation of the Koopman operator, as outlined in e.g. Brunton et al. (2021). However, proving convergence of the finite-dimensional spectral truncations of Koopman operators can require strong assumptions or compactification tricks such as different input and output spaces Powell et al. (2023), without which it becomes difficult to describe convergence of continuous and residual spectra. Furthermore, the mechanism of pole placement via observer systems becomes obscured in such settings.

We adopt the improper learning perspective that the only thing that matters is approximately correct observations $y_t' \approx y_t$, and that convergence of spectra is not strictly necessary for sequence prediction. The discretization and Markov chain outlined above achieves this, and the proof is clean and simple with fewer assumptions[24].

**Remark C.6.** *This Markov chain is closely related to the Perron-Frobenius operator $P$, the adjoint to the Koopman operator. Informally, $P$ is a linear operator on spaces of measures on the state space $\mathcal{X}$ which sends a measure $\mu$ to its pushforward under the dynamics, i.e. $P\mu = f_{\#}\mu$. One could imagine an infinite-dimensional LDS of the form*

$$\mu_{t+1} = P\mu_t, \quad y_t = \int_{\mathcal{X}} h(x) d\mu_t(x), \quad \mu_0 = \delta_{x_0}$$

*which generates the same observations as the original system, where both the dynamics and observations are linear with respect to the evolving measure. The Markov chain in Lemma C.5 is constructing exactly this LDS, but on a discretized state space to ensure finite-dimensionality (i.e. probability measures are represented via probability vectors). This discretization argument circumvents any need for proofs of convergence of spectral truncations of the Koopman operator or its adjoint $P$.*

---

[24]Note, however, that any construction made via truncated Koopman operators will also fall into the comparator class of high-dimensional LDS, against which we have a regret guarantee from Theorem A.2. If there were a better high-dimensional LDS that approximates the sequence $y_1, \ldots, y_T$, then Algorithm 1 would match it without needing to know anything about it.

Importantly, the observation that the Markov chain constructed in Lemma C.5 is (dual to) the Koopman operator implies that the discretized LDS retains the spectral properties of the Koopman operator. Technically, since we use the Euclidean norm for the analysis of this discretized LDS we inherit the spectral properties of the Koopman operator on $L^2(\mathcal{X}, \pi)$ function spaces, where the choice of measure $\pi$ is equivalent to the choice of discretization grid. A very useful result of our improper learning formulation of the problem is that our analysis adapts in hindsight to the best choice of discretization grid, i.e. the Koopman operator on a function space that gives it the most desirable spectral properties. In this way, we sidestep the suboptimalities and difficulties of any explicit choices of function space.

In any case, we are left with a high-dimensional LDS which approximates the original system, and we will conclude by applying Theorem A.2. First, we define the relevant quantities which we use in the statement and proof of Theorem 4.1.

**Definition C.7** (Nonlinear Luenberger program). *Let $(f, h, x_0)$ be a nonlinear dynamical system as per (1.1) with $\|x_t\|, \|y_t\| \le R$ and $f, h$ being 1-Lipschitz. For any discretization $\varepsilon > 0$ and spectral constraint set $\Sigma \subseteq \mathbb{C}$, we consider the optimization program over $L \in \mathbb{R}^{\left(\frac{2R}{\varepsilon}\right)^{d_{\mathcal{X}}} \times d_{\mathcal{Y}}}$ given by*

$$\min \quad \kappa_{diag}(A' - LC') \quad s.t. \quad A' - LC' \text{ has all eigenvalues in } \Sigma,$$

*where $A'$ and $C'$ are the Markov chain transition and observation matrices constructed in Lemma C.5. When the constraint set of the optimization is nonempty, with an overloading of notation we denote the minimal value by $Q_\star = Q_\star(f, h, \varepsilon, \Sigma) = Q_\star(A', C', \Sigma) < \infty$ and a minimizing gain by $L_\star$.*

*Proof of Theorem 4.1.* Let $(f, h, x_0)$ be a nonlinear dynamical system satisfying Assumption 3.1, and let

$$z_{t+1} = A' z_t, \quad y_t' = C' z_t$$

be the high-dimensional LDS constructed in Lemma C.5 at some scale $\varepsilon$, where $z_0$ is the indicator vector of $\pi(x_0)$. With $\varepsilon = \frac{1}{T^{3/2}}$ this satisfies Assumption A.1 with $R_{x_0} = 1$ (since $z_0$ is a one hot probability vector), $W = 0$, the same bound $R$ on the signal, $R_B = 0$ (since there are no inputs), and $R_C \le R\sqrt{N} \le R \cdot \left(2RT^{3/2}\right)^{d_{\mathcal{X}}/2}$. Letting $\hat{y}_1, \ldots, \hat{y}_T$ be the predictions made by Algorithm 1, by Theorem A.2(b) (or more precisely by a regret phrasing, see Theorem G.2(b)),

$$\sum_{t=1}^{T} \|\hat{y}_t - y_t\| \le \sum_{t=1}^{T} \|y_t' - y_t\| + O\left(Q_\star^2 \log(Q_\star) \cdot Rm^2 \log\left((2RT^{3/2})^{d_{\mathcal{X}}}\right) \log^7(mT) \cdot \sqrt{T}\right)$$

The bound in (i) follows from the discretization error bound of Lemma C.5. (ii) is proved in the same way by using the VAW forecaster's logarithmic dependence on optimization diameter, see Theorem G.6. $\square$

## D  PROOF OF LEMMA C.4

In this section we provide a proof of Lemma C.4, which follows the proof techniques from Hazan et al. (2017).

*Proof.* We measure regret against $\text{LDS}(\rho, \gamma, \kappa, R_B, R_C, R_{x_0})$, which we recall to be the class of LDS predictors with spectrum $\sigma(A) \subseteq [0, 1 - \rho] \cup \{1\} \cup \mathbb{D}_{1-\gamma}$, spectral condition number $\kappa_{\text{diag}}(A) \le \kappa$, and the bounds $\|B\|_F \le R_B$, $\|C\|_F \le R_C$, and $\|x_0\| \le R_{x_0}$. For notation, we let $\Theta = (A, B, C, x_0)$ denote the LDS predictor and $y_t^\Theta$ its predictions, and $A = HDH^{-1}$ with $D = \text{diag}(\alpha_1, \ldots, \alpha_{d_h})$ the eigenvalues of $A$.

We first prove an approximation result, i.e. that there exist parameters $J, M, P, N$ that match the predictions of any given LDS. As in Section 2.2 of Hazan et al. (2017), our predictor $\hat{y}_{t+1}$ will add a prediction of the derivative of the impulse response $y_{t+1}^\Theta - y_t^\Theta$ to $y_t$. The derivative is

$$y_t^\Theta - y_{t-1}^\Theta = CBu_{t-1} + \sum_{i=1}^{t} C(A^i - A^{i-1})Bu_{t-i-1} + C(A^t - A^{t-1})x_0$$

$$= CBu_{t-1} + \sum_{i=1}^{t} CH(D^i - D^{i-1})H^{-1}Bu_{t-i-1} + C(A^t - A^{t-1})x_0.$$

We will order the eigenvalues so that there exists $N \in [d_h]$ such that $\alpha_j \in [0,1]$ for all $j \leq N$ and $|\alpha_j| \leq 1 - \gamma$ for all $j > N$. Defining $\mu_\alpha = (\alpha - 1)[1, \alpha, \ldots, \alpha^{T-2}] \in \mathbb{R}^{T-1}$, we have

$$y_t^\Theta - y_{t-1}^\Theta = CBu_{t-1} + \sum_{i=1}^{t} \sum_{j=1}^{d_h} (\alpha_j^i - \alpha_j^{i-1}) CHe_j e_j^\top H^{-1} Bu_{t-i-1} + C(A^t - A^{t-1})x_0$$

$$= CBu_{t-1} + \sum_{j=1}^{d_h} CHe_j e_j^\top H^{-1} B \left( \sum_{i=1}^{t} (\alpha_j - 1)\alpha_j^{i-1} u_{t-i-1} \right) + C(A^t - A^{t-1})x_0$$

$$= CBu_{t-1} + \sum_{j=1}^{d_h} CHe_j e_j^\top H^{-1} B\mu_{\alpha_j}^\top u_{t-2:t-T} + C(A^t - A^{t-1})x_0,$$

where $u_{t-2:t-T} \in \mathbb{R}^{(T-1) \times d_{\mathrm{in}}}$ and so $\mu_{\alpha_j}^\top u_{t-2:t-T} \in \mathbb{R}^{d_{\mathrm{in}}}$. We split the sum over hidden dimension into two terms: $j \leq N$ will be handled via spectral filtering and $j > N$ via regression. Letting $\{\sigma_i, \phi_i\}_{i=1}^{T-1}$ be the eigenvalues and eigenvectors of the Hankel matrix from Algorithm 1 (see Section 3 of Hazan et al. (2017)), we have

$$\sum_{j=1}^{N} CHe_j e_j^\top H^{-1} \mu_{\alpha_j}^\top u_{t-2:t-T} = \sum_{j=1}^{N} CHe_j e_j^\top H^{-1} B\mu_{\alpha_j}^\top \left( \sum_{i=1}^{T-1} \phi_i \phi_i^\top \right) u_{t-2:t-T}$$

$$= \sum_{i=1}^{T-1} \sum_{j=1}^{N} CHe_j e_j^\top H^{-1} B(\mu_{\alpha_j}^\top \phi_i)(\phi_i^\top u_{t-2:t-T})$$

Define the optimal parameters $J_1 = CB$, $J_i = CH\left(\sum_{j=N+1}^{d_h}(\alpha_j^i - \alpha_j^{i-1})e_j e_j^\top H^{-1} B\right)$ for $i > 1$, $M_i = \sigma_i^{-1/4} CH\left(\sum_{j=1}^{N} \mu_{\alpha_j}^\top \phi_i e_j e_j^\top\right) H^{-1} B$ for $i \in [T-1]$, and $P_1 = 1$, $P_i = 0$ for $i > 1$, and $N \equiv 0$. Then, we have the approximation error[25]

$$y_t^\Theta - \hat{y}_t(J, M, P, N) = \sum_{i=m+1}^{t} J_i u_{t-i} + \sum_{i=h+1}^{T-1} \sigma_i^{1/4} M_i \langle \phi_i, u_{t-2:t-T} \rangle + C(A^t - A^{t-1})x_0.$$

We bound

$$\|J_i\| \leq \left( \sup_{\alpha \in \mathbb{D}_{1-\gamma}} |\alpha^i - \alpha^{i-1}| \right) \cdot \sum_{j=N+1}^{d_h} \|CHe_j\| \cdot \|e_j^\top H^{-1} B\|$$

$$\leq 2(1-\gamma)^{i-1} \cdot \|CH\|_F \cdot \|H^{-1}B\|_F$$

$$\leq 2(1-\gamma)^{i-1} R_B R_C \kappa,$$

where in the second line we applied Cauchy-Schwartz and in the third we used $\|S_1 S_2\|_F \leq \min(\|S_1\|\|S_2\|_F, \|S_1\|_F\|S_2\|)$. Similarly,

$$\|M_i\| \leq \sigma_i^{-1/4} \cdot \left( \sup_{\alpha \in [0,1]} |\mu_\alpha^\top \phi_i| \right) \cdot \sum_{j=1}^{N} \|CHe_j\| \cdot \|e_j^\top H^{-1} B\|$$

$$\leq 6^{1/4} R_B R_C \kappa,$$

where the second line uses Lemma E.4 of Hazan et al. (2017). Since $\|u_{t-i}\| \leq R$ and $|\langle \phi_i, u_{t-2} : t-T \rangle| \leq R\sqrt{T}$ by Cauchy-Schwartz, it remains to bound $\|C(A^t - A^{t-1})x_0\|$. If $A$ has all eigenvalues 1 then this term is zero, and so suppose that the largest eigenvalue of $A$ not equal to 1 has magnitude $1 - \rho$. Then,

$$\left\| C(A^t - A^{t-1})x_0 \right\| \leq \sum_{j=1}^{d_h} |\alpha_j^t - \alpha_j^{t-1}| \cdot \|CHe_j\| \cdot \|e_j^\top H^{-1} x_0\|.$$

---

[25]Technically, we are considering the class of LDS predictors which add the derivative of the impulse response to the ground truth $y_{t-1}$. Otherwise, we would suffer an additional $y_{t-1} - y_{t-1}^\Theta$ approximation error.

For $j$ such that $\alpha_j = 1$ the summand is 0, and for all other $j$ we know $|\alpha_j^t - \alpha_j^{t-1}| \leq 2 \cdot (1-\rho)^{t-1}$. Furthermore, again by Cauchy-Schwartz we have $\sum_{j=1}^{d_h} \|CHe_j\| \cdot \|e_j^\top H^{-1}x_0\| \leq \|CH\|_F \cdot \|H^{-1}x_0\| \leq R_C\kappa R_{x_0}$. Combining these ingredients,

$$\left\|y_t^\Theta - \hat{y}_t(J,M,P,N)\right\| \leq 2R_B R_C \kappa R \cdot \sum_{i=m+1}^{T} (1-\gamma)^{i-1} + 6^{1/4}R_B R_C \kappa R\sqrt{T} \cdot \sum_{i=h+1}^{T-1} \sigma_i^{1/4} + 2R_C\kappa R_{x_0} \cdot (1-\rho)^{t-1}$$

We note that $(1-\gamma)^n \leq e^{-\gamma n}$ and that $\sigma_i^{1/4} \leq Ke^{-i/(2\log T)}$ for an absolute constant $K < 10^6$ by Lemma E.2 of Hazan et al. (2017). Since $\sum_{i>n} e^{-ai} \leq \frac{e^{-an}}{a}$, we get the approximation result

$$\left\|y_t^\Theta - \hat{y}_t(J,M,P,N)\right\| \lesssim R_B R_C \kappa R \cdot \frac{e^{-\gamma(m-1)}}{\gamma} + R_B R_C \kappa R \log(T)\sqrt{T} \cdot e^{-\frac{h}{2\log T}} + R_C\kappa R_{x_0} \cdot e^{-\rho(t-1)}$$

However, we also know that $\left\|y_t^\Theta - \hat{y}_t(J,M,P,N)\right\| \leq 2R$, and so if we let $\tau = \max\left(\frac{\log(R_C\kappa R_{x_0}/2R)}{\rho}, 1\right)$ then $R_C\kappa R_{x_0}e^{-\rho\tau} \leq 2R$, and therefore

$$\left\|y_t^\Theta - \hat{y}_t(J,M,P,N)\right\| \lesssim R_B R_C \kappa R \cdot \frac{e^{-\gamma(m-1)}}{\gamma} + R_B R_C \kappa R \log(T)\sqrt{T} \cdot e^{-\frac{h}{2\log T}} + R \cdot \begin{cases} 1 & t \leq \tau \\ e^{-\rho(t-\tau)} & t > \tau \end{cases}$$

Summing over all $t$ and using $\sum_{t>\tau} e^{-\rho(t-\tau)} \leq \frac{1}{\rho}$ gives

$$\sum_{t=1}^{T} \left\|y_t^\Theta - \hat{y}_t(J,M,P,N)\right\| \lesssim R_B R_C \kappa R T \cdot \frac{e^{-\gamma(m-1)}}{\gamma} + R_B R_C \kappa R \log(T)T^{3/2} \cdot e^{-\frac{h}{2\log T}}$$
$$+ \frac{R\log_+(2R_C\kappa R_{x_0}/R)}{\rho} .$$

Taking $m = \Theta\left(\frac{1}{\gamma}\log\left(\frac{T}{\gamma}\right)\right)$ and $h = \Theta\left(\log^2 T\right)$, we get for any LDS parameterized by $\Theta$ there exist parameters $J_\Theta, M_\Theta, P_\Theta, N_\Theta$ for which

$$\sum_{t=1}^{T} \left\|y_t^\Theta - \hat{y}_t(J,M,P,N)\right\| \leq O\left(R_B R_C \kappa R \log(T)\sqrt{T} + \frac{R\log_+(R_C\kappa R_{x_0}/R)}{\rho}\right).$$

If we don't want to assume a spectral gap (i.e. $\rho = 0$), then we can instead use that $\sup_{\alpha\in[0,1]} |\alpha^t - \alpha^{t-1}| \leq \frac{1}{t-1}$ and $\sup_{\alpha\in\mathbb{D}_{1-\gamma}} |\alpha^t - \alpha^{t-1}| \leq 2(1-\gamma)^{t-1}$ to get

$$\sum_{t=1}^{T} \left\|C(A^t - A^{t-1})x_0\right\| \leq \|C\|\kappa \cdot \left(\frac{1}{\gamma} + \log T\right).$$

Now, we know that the loss is a convex function of $J, M, P, N$, which means that we can perform optimization via OGD. By our earlier bounds, the diameter of the constraint set is $D = O((m+h)R_B R_C \kappa)$. The gradient of the unsquared $\ell_2$ loss w.r.t. the prediction error is of size 1, and the gradient of the prediction w.r.t. $J$ and $P$ are bounded by $mR$ and $R$, respectively. The gradient w.r.t. $M_i$ is bounded by $\sigma_i^{1/4} \cdot |\langle\phi_j, u_{t-2:t-T}\rangle| \lesssim R\log T$, where we applied Corollary E.6 of Hazan et al. (2017) and Holder's inequality. In total, we get the gradient bound $G \leq O(mR + hR\log T) \leq O((m+h)R\log T)$. By the OGD regret guarantee (Theorem 3.1 of Hazan (2016)), if $\hat{y}_t$ are the iterates of Algorithm 1 then

$$\sum_{t=1}^{T} \|\hat{y}_t - y_t\| - \min_\Theta \sum_{t=1}^{T} \|\hat{y}_t(J_\Theta, M_\Theta, P_\Theta, N_\Theta) - y_t\| \leq O\left((m+h)^2 R_B R_C \kappa R \log(T)\sqrt{T}\right).$$

$\square$

**Remark D.1.** *The above proof shows that, if one increases $m$ and $h$ by $\log(R_B R_C \kappa R)$, then the approximation error can be made independent of those constants. By these proof techniques, the only place where one must incur $\Theta(\kappa)$ regret is in the diameter of the constraint set. This dependence can be improved to $\Theta(\log\kappa)$ for the square loss with more sophisiticated online learning algorithms, such as the Vovk-Azoury-Warmuth (VAW) forecaster Azoury & Warmuth (2001), and in certain cases Online Newton Step Hazan (2016). This is implemented in Theorem G.6.*

**Remark D.2.** *This proof has two main differences from that of Theorem 1 of Hazan et al. (2017). Firstly, we prove things for an algorithm which combines regression terms in order to learn systems with spectrum in $[0,1] \cup \mathbb{D}_{1-\gamma}$; spectral filtering handles the $[0,1]$ part and regression with $\widetilde{O}\left(\frac{1}{\gamma}\right)$ parameters handles the $\mathbb{D}_{1-\gamma}$ part. The second (and more theoretically interesting) difference is that rather than handling the vanishing initial state using the envelope $\sup_{\alpha \in [0,1]} |\alpha^t - \alpha^{t-1}| \leq \frac{1}{t}$ to get linear decay, we make use of a spectral gap condition to get exponential decay of the initial state effect.*

# E    LOWER BOUNDS

In this section we give lower bounds for learning sequences generated by linear and nonlinear dynamical systems, which complement our upper bounds and show them to be tight in certain respects. This does not mean they cannot be improved: to the contrary, this investigation motivates further study of the leading order constants and parameters of our bounds.

## E.1    LDS WITH NOISE

We have shown that spectral filtering with observation feedback is able to learn asymmetric marginally stable LDS's under adversarial disturbances. Our regret guarantee scales with two quantities: the optimal observer complexity $Q_\star$ and the disturbance sizes $\sum_{t=0}^{T-1} \|w_t\|$. In this section, we investigate the the leading order terms of this result via lower bounds for sequence prediction in terms of these quantities. In particular, we prove that any algorithm which is not able to see $w_t$ when predicting $\hat{y}_{t+1}$ suffers a linear cost in terms of the disturbance sizes and ambient dimension up to constant factors:

**Theorem E.1.** *Let $\mathcal{A}$ be any algorithm that predicts $\hat{y}_{t+1}$ using $u_1, \ldots, u_t$, $y_1, \ldots, y_t$, and $w_1, \ldots, w_{t-1}$. Then, there exists a problem instance $(A, B, C, x_0)$ and sequences $u_1, \ldots, u_T$ and $w_1, \ldots, w_T$ satisfying Assumption A.1 for which*

$$\sum_{t=1}^{T} \|\hat{y}_t^{\mathcal{A}} - y_t\| \geq \Omega\left(d \sum_{t=0}^{T-1} \|Cw_t\|\right)$$

*Proof.* Consider the following linear dynamical system. We let $A$ be the permutation matrix over $d$ elements, $B$ is zero, and $C$ is the first standard basis vector, as given by

$$A_d^{\mathrm{perm}} = \begin{bmatrix} 0 & 0 & \cdots & 0 & 1 \\ 1 & 0 & \cdots & 0 & 0 \\ 0 & 1 & \cdots & 0 & 0 \\ \vdots & \vdots & \ddots & \vdots & \vdots \\ 0 & 0 & \cdots & 0 & 0 \\ 0 & 0 & \cdots & 1 & 0 \end{bmatrix} \ , \ B = 0 \ , \ C = \begin{bmatrix} 1 \\ 0 \\ \vdots \\ 0 \end{bmatrix} .$$

Notice that the observation at time $t$ is given by

$$y_t = w_{1:t}(t \mod d),$$

that is, the sum of all noises at a rotating coordinate according to the permutation. Suppose $t \mod d = 1$ for simplicity.

Consider any prediction algorithm $\mathcal{A}$. We create a noise sequence as follows: in the $d$ iterations before the iteration $t$ such that $t \mod d = 1$, the noises $w_\tau(1)$ are going to be either $-y_{t-d} - 1$ or $-y_{t-d} + 1$, chosen uniformly at random, and equal for these $d$ iterations. The following hold:

- Regardless of the algorithm $\mathcal{A}$ prediction, it's loss is on expectation can predict w.l.o.g. $y_{t-d} \in \{-d, d\}$, since the observation belongs to this set. We have that $\|\hat{y}_t^{\mathcal{A}} - y_t\|$ is zero w.p. $\frac{1}{2}$ and $2d$ w.p. $\frac{1}{2}$.

- The expected loss of the algorithm is thus, over $T$ iterations,

$$\sum_{t=1}^{T} \|\hat{y}_t^{\mathcal{A}} - y_t\| \geq dT = d \sum_t \|Cw_t\|$$

- The optimal predictor in hindsight that has access to $w_{t-1}$, which has the correct sign of the upcoming prediction, and thus has loss of zero.

$\square$

## E.2 COMPUTATIONAL LOWER BOUNDS

The preceding lower bound applies to linear dynamical systems with adversarial (or even stochastic) noise. We next give a computational lower bound sketch, based on cryptographic hardness assumptions, for deterministic nonlinear dynamical systems.

**Theorem E.2.** *Let* $\mathrm{PRG} : \{0,1\}^d \to \{0,1\}^T$ *be a cryptographically secure pseudorandom generator with stretch* $T = \mathrm{poly}(d)$. *Consider the deterministic dynamical system*

$$\mathcal{X} = \{0,1\}^d \times \{0,1,\ldots,T-1\}, \quad F(s,t) = (s, (t+1) \bmod T), \quad g(s,t) = 2\,\mathrm{PRG}(s)_t - 1 \in \{-1,+1\},$$

*and observations* $y_t = g(x_t)$ *from hidden initial state* $(s,0)$.

*Then, for every probabilistic polynomial-time predictor* $\widehat{f}$, *every polynomial* $p(\cdot)$, *and all sufficiently large* $d$,

$$\forall t \in \{0,\ldots,T-1\}: \qquad \mathbb{E}\left[\left|\widehat{f}(y_0,\ldots,y_t) - y_{t+1}\right|\right] \geq 1 - \tfrac{1}{p(d)}.$$

*That is, the expected* $\ell_1$ *loss per step is* $1 - o(1)$, *matching the loss of a random* $\pm 1$ *guess, for all* $t < T$.

*Moreover, there exists an algorithm running in* $O(2^d \cdot \mathrm{poly}(T))$ *time that, given* $(y_0,\ldots,y_{T-1})$, *recovers the seed* $s$ *by exhaustive search and then predicts all future outputs exactly.*

*Proof sketch.* Suppose for contradiction that there exists a PPT predictor $\widehat{f}$, an index $t < T$, and a polynomial $q(\cdot)$ such that

$$\mathbb{E}\left[\left|\widehat{f}(y_0,\ldots,y_t) - y_{t+1}\right|\right] \leq 1 - \tfrac{2}{q(d)}.$$

Since $y_{t+1} \in \{-1,+1\}$, this means $\widehat{f}$ predicts the sign of $y_{t+1}$ with probability at least $\tfrac{1}{2} + \tfrac{1}{q(d)}$. We now construct a distinguisher $\mathcal{D}$ for PRG security: Given a string $z \in \{-1,+1\}^T$, feed $(z_0,\ldots,z_t)$ to $\widehat{f}$, output "PRG" if the sign of $\widehat{f}$ matches $z_{t+1}$, and "uniform" otherwise. If $z$ is $\mathrm{PRG}(s)$ for random $s$, this succeeds with probability $\tfrac{1}{2} + \tfrac{1}{q(d)}$; if $z$ is uniform, the success probability is exactly $\tfrac{1}{2}$. This yields a polynomial-time distinguisher with advantage $1/q(d)$, contradicting PRG security.

The exhaustive search recovery algorithm enumerates all $2^d$ seeds $s'$, computes $\mathrm{PRG}(s')$, and finds the unique one matching the observed $(y_0,\ldots,y_{T-1})$. This takes $O(2^d \cdot \mathrm{poly}(T))$ time and recovers the seed exactly, enabling perfect prediction thereafter. $\square$

# F LUENBERGER PROGRAM

In this section, we investigate the Luenberger program of Definition C.1, restated here for convenience: for given $A \in \mathbb{R}^{d_h \times d_h}$, $C \in \mathbb{R}^{d_{\mathrm{obs}} \times d_h}$, and $\Sigma \subseteq \mathbb{D}_1$, we consider the optimization

$$\min_{L \in \mathbb{R}^{d_h \times d_{\mathrm{obs}}}} \kappa_{\mathrm{diag}}(A - LC) \quad \text{s.t.} \quad A - LC \text{ has all eigenvalues in } \Sigma, \tag{F.1}$$

and we denote the minimal value by $Q_\star = Q(A,C,\Sigma)$ and a minimizing matrix, when it exists, by $L_\star$. First, we summarize some known formulas for the important quantities:

**Lemma F.1.** *Suppose that* $(A,C)$ *is observable. Then, for any monic, degree-$d_h$ polynomial $p$ with distinct roots, there is a unique $L \in \mathbb{R}^{d_h \times d_{\mathrm{obs}}}$ such that $A - LC$ is diagonalizable with characteristic polynomial $p$. In particular, if $(A,C)$ is observable then $Q_\star(A,C,\Sigma) < \infty$ for all $\Sigma$ containing more than $d_h$ points which is symmetric about the real line.*

*Proof.* This is a standard result in eigenvalue placement, and for $d_{\mathrm{obs}} = 1$ there is an explicit formula known as Ackermann's formula (see e.g. Corollary 2.6 of Mehrmann & Xu (1996)). $\square$

The above shows that observability implies $Q_\star < \infty$. Also, the bijection between $p$ and $L$ informs us that we may instead consider the optimization (F.1) as being parameterized by the desired poles. As a crude but general upper bound:

**Lemma F.2.** *Suppose that* $(A,C)$ *is observable and* $\dim(\Sigma) \geq 1$, *let* $\mathrm{vol}(\Sigma)$ *denote the appropriate notion of volume, and let* $N$ *be the number of eigenvalues of* $A$ *not in* $\Sigma$. *Then, we have the bounds*

$$Q_\star(A,C,\Sigma) \leq \left(1 + \frac{2N^8}{\mathrm{vol}(\Sigma)}\right)^{2N-2} = \left(\frac{N}{\mathrm{vol}(\Sigma)}\right)^{O(N)},$$

$$\|L_\star\| \leq \frac{2Q_\star}{\sigma_{min}(C)}.$$

*Proof.* Let $\hat{A} = A - LC$ for notation. Firstly, since eigenvalue placement respects blocks of $A$ we can w.l.o.g. assume that $A$ is $N \times N$. By Theorem 1 of Jiang & Lam (1997), we have

$$\kappa_{\text{diag}}(\hat{A}) \leq \left(1 + \frac{\alpha}{\delta}\right)^{2N-2}$$

for $\delta$ the minimum eigengap of $\hat{A}$ and $\alpha \leq N^{3/4} \cdot \mu_2(\hat{A})$, where we use the $\mu_j$ notation of Elsner & Paardekooper (1987). By (C0), (C3), and (C5) of Elsner & Paardekooper (1987) we know $\mu_2^2 \leq 4\sqrt{N}\|\hat{A}\|^2$ and therefore $\alpha \leq 2N\|\hat{A}\|$. If we take the rows of $H$ to be unit norm (i.e. normalizing the eigenvectors) then by Theorem 2 of Gheorghiu (2003) we have that

$$\|\hat{A}\| \leq 1 + N^{9/4}\rho(A)\gamma(A) \leq 1 + N^{23/4} \leq N^6,$$

where we also used $\ell_1, \ell_2$ norm inequalities. Combining everything (and observing that we can choose to place the eigenvalues with an eigengap of $\frac{\text{vol}(\Sigma)}{N}$) gives the bound on $\kappa_{\text{diag}}(\hat{A})$. Since it is independent of $p$, this implies the bound on $Q_\star$. The bound on $\|L\|$ in terms of the condition number of $H$ is equation (22) of Kautsky et al. (1985) and the fact that $Q_\star \geq 1$. □

The above bound is very general, but is exponential in the hidden dimension. Next, we look into a representative example of an asymmetric system to demonstrate the finer properties of this optimization:

**Example F.3** (Permutation system). *Let $A$ be a cyclical $n \times n$ permutation matrix, i.e.*

$$A = \begin{bmatrix} 0 & 1 & 0 & \cdots & 0 \\ 0 & 0 & 1 & \cdots & 0 \\ \vdots & \vdots & \ddots & \ddots & \vdots \\ 0 & 0 & \cdots & 0 & 1 \\ 1 & 0 & \cdots & 0 & 0 \end{bmatrix},$$

*$C = e_1$ be the projection on the first coordinate, and $p$ be a monic degree-$n$ polynomial (with coefficients $p_j$ and roots $\lambda_j$). Then, the gain $L \in \mathbb{R}^{n \times 1}$ and eigenvector matrix $H$ satisfy*

$$\|L\|^2 = \sum_{j=0}^{n} |p_j|^2, \quad \kappa(H) = \kappa\left(\text{diag}\left(\frac{1}{1 - \vec{\lambda}^n}\right) \cdot V(\vec{\lambda})\right)$$

*for $V(\vec{\lambda})$ the Vandermonde matrix with knots $\lambda_j$. In particular, $\|L\|$ can be made to be small but $\kappa(H)$ is always exponential if $\Sigma$ does not include the roots of unity, i.e.*

$$Q_\star = \Omega(2^n).$$

*Proof.* We will use the explicit formulas for $H$ and $L$ from Corollaries 2.5 and 2.6 of Mehrmann & Xu (1996). We know that $c^\top A^k e_j = \langle c, A^k e_j \rangle = \langle c, e_{j+k \mod n} \rangle = \delta_{j+k \mod n=1}$, and so

$$CA = e_2, \quad \ldots, \quad CA^{n-1} = e_n$$

This means that the observability matrix is

$$\mathcal{O} = \begin{bmatrix} C \\ CA \\ \vdots \\ CA^{n-1} \end{bmatrix} = \begin{bmatrix} e_1 \\ e_2 \\ \vdots \\ e_n \end{bmatrix} = \text{Id}$$

In particular, since we know that $A$ is unitarily diagonalized as $A = FDF^{-1}$ for $F$ the DFT matrix and $D = \text{diag}(1, \omega, \omega^2, \ldots, \omega^{n-1})$ with $\omega = e^{2\pi i/n}$, we find

$$L = p(A) \cdot e_n = F \text{diag}(1, p(\omega), p(\omega^2), \ldots, p(\omega^{n-1}))F^{-1} \cdot e_n$$

Writing this in the standard basis, we find that the $j^{th}$ coordinate of $L$ for $1 \leq j \leq n$ is

$$\sum_{k=0}^{n-1} \frac{\omega^{jk} \cdot p(\omega^k)}{n}$$

and so

$$\|L\|^2 = \frac{1}{n^2} \sum_{j=1}^{n} \sum_{k_1=0}^{n-1} \sum_{k_2=0}^{n-1} \omega^{j(k_1+k_2)} \cdot p(\omega^{k_1}) \cdot p(\omega^{k_2})$$

$$= \frac{1}{n^2} \sum_{k_1=0}^{n-1} \sum_{k_2=0}^{n-1} p(\omega^{k_1}) \cdot p(\omega^{k_2}) \cdot \sum_{j=1}^{n} \omega^{j(k_1+k_2)}$$

We note that $\sum_{j=1}^{n} \omega^{q \cdot j}$ is equal to 0 unless $q$ is a multiple of $n$, in which case it equals $n$. Therefore, we are left with

$$\|L\|^2 = \frac{1}{n} \sum_{k_1+k_2=n} p(\omega^{k_1}) \cdot p(\omega^{k_2})$$

$$= \frac{1}{n} \sum_{k=0}^{n-1} p(\omega^k) \cdot p(\omega^{n-k})$$

$$= \frac{1}{n} \sum_{k=0}^{n-1} |p(\omega^k)|^2,$$

where we used that $p(\omega^{-k}) = \overline{p(\omega^k)}$ for a polynomial $p$ with real coefficients. Lastly, noting that

$$p(\omega^k) = \sum_{j=0}^{n} p_j \omega^{kj}$$

we see that the sequence $p(\omega^k)$ is none other than the DFT of the sequence $p_j$ of coefficients of $p$. By Parseval's identity, we get the stated result about $L$.

For $H$, we once again write $A = FDF^{-1}$, and so that $j^{th}$ eigenvector $h_j$ of $A - LC$ equals

$$h_j = (A - \lambda_j \,\mathrm{Id})^{-1} C = F \,\mathrm{diag}\left( \frac{1}{1-\lambda_j}, \frac{1}{\omega - \lambda_j}, \ldots, \frac{1}{\omega^{n-1} - \lambda_j} \right) F^{-1} e_1$$

Noting that $F^{-1} e_1 = \frac{1}{\sqrt{n}}[1, \ldots, 1]$, we see that if $V'$ is the Cauchy matrix with entries

$$V'_{ij} = \frac{1}{\omega^{j-1} - \lambda_i} \quad (1 \le i, j \le n)$$

then $H = \frac{1}{\sqrt{n}} FV$. Therefore, the condition number of $H$ equals that of $V'$. By equation (7) of Pan (2013), we find that $V'$ has the same condition number as $(D^n - \mathrm{Id})^{-1} V$, where $V$ is the Vandermonde matrix with knots $\lambda_j$ and $D$ is the diagonal matrix with entries $\lambda_j$. We have arrived at the fact that

$$Q_\star = \min_{\vec{\lambda} \subseteq \Sigma} \kappa \left( \mathrm{diag}\left( \frac{1}{1 - \vec{\lambda}^n} \right) \cdot V(\vec{\lambda}) \right).$$

The only Vandermonde matrices with subexponential condition number are those with knots close to the roots of unity Pan (2015), in which case the row rescalings blow up. As such, it can be seen that for any admissible choice of $\lambda_j$'s, the condition number of this row-rescaled Vandermonde matrix will be exponential in the dimension. $\qquad\square$

## G  OTHER GUARANTEES

The presented framework is quite general: using the Markov chain discretization together with observer systems, we can construct comparator LDS's with any specified spectral structure (with a cost measured by $Q_\star$), and we combine this with a regret guarantee against the chosen class of comparator LDS's.

Before deriving other forms of loss and regret bounds, we stop to note that since we prove things for the unsquared $\ell_2$ loss, everything extends immediately to all Lipschitz loss sequences (since we use OCO, the losses may be time-varying). Furthermore, since we assume boundedness of the outputs, we can also use the square loss with an $O(R)$ Lipschitz constant.

To start, our proof technique yields for free the agnostic versions of our main results. It is nice to view our results in this way, as it defines exactly which types of computation are able to be efficiently implemented via spectral filtering – on arbitrary data, if there exists a computer of the classes defined below which fits the data well with small $Q_\star$, then spectral filtering will succeed.

**Theorem G.1** (Nonlinear, agnostic). *Let $y_1, \ldots, y_T$ be an arbitrary sequence bounded by $R$. Fix the number of algorithm parameters $h = \Theta(\log^2 T)$ and $m \geq 1$, and for notation define $\gamma = \frac{\log(mT)}{m}$.*

*Let $\Pi_{NL}(q)$ denote the class of nonlinear dynamical systems $(f, h, x_0)$ such that $\|x_t\| \leq R$ and $Q_\star\left(f, h, \frac{1}{T^{3/2}}, \left[0, 1 - \frac{1}{\sqrt{T}}\right] \cup \{1\} \cup \mathbb{D}_{1-\gamma}\right) \leq q$. Then, running Algorithm 1 with $J^t, M^t \equiv 0$, $\mathcal{K} = R_D$ for $D = \Theta((m + h)q^2)$, and $\eta_t = \Theta\left(\frac{q^2}{R\sqrt{t}\log T}\right)$ produces a sequence of predictions $\hat{y}_1, \ldots, \hat{y}_T$ for which*

$$\sum_{t=1}^{T} \|\hat{y}_t - y_t\| - \min_{\pi \in \Pi_{NL}(q)} \sum_{t=1}^{T} \|\hat{y}_t^\pi - y_t\| \leq O\left(q^2 \log(q) \cdot d_{\mathcal{X}} R \log(RT) m^2 \log^7(mT) \cdot \sqrt{T}\right).$$

*Proof.* The proof proceeds in the same way as that of Theorem 4.1: we use Lemma C.5 to construct a high-dimensional LDS, which we have a regret guarantee against via Theorem G.2. □

**Theorem G.2** (Linear, agnostic). *Let $u_1, \ldots, u_T$ and $y_1, \ldots, y_T$ be arbitrary sequences bounded by $R$. Fix the number of algorithm parameters $h = \Theta(\log^2 T)$ and $m \geq 1$, and for notation define $\gamma = \frac{\log(mT)}{m}$ and $Q = R_B R_C q + q^2$. Set $\mathcal{K} = R_D$ for $D = \Theta((m + h)Q)$ and $\eta_t = \Theta\left(\frac{Q}{R\sqrt{t}\log T}\right)$, and let $\hat{y}_1, \ldots, \hat{y}_T$ be the sequence of predictions that Algorithm 1 produces.*

*Fix the constants $R_B, R_C, R_{x_0}, W$, and let $\Pi_L(w, q, \Sigma)$ denote the class of LDS predictors (with arbitrary disturbances bounded by $\sum_{t=1}^{T-1} \|w_t\| \leq w$) which satisfy Assumption A.1 with the specified constants and for which $Q_\star(A, C, \Sigma) \leq q$. Then, we have the following three regret guarantees:*

*(a). With $w = 0$,*

$$\sum_{t=1}^{T} \|\hat{y}_t - y_t\| - \min_{\pi \in \Pi_L(0, q, [0,1] \cup \mathbb{D}_{1-\gamma})} \sum_{t=1}^{T} \|\hat{y}_t^\pi - y_t\| \leq O\left(QRm^2 \log^7(mT) \cdot \sqrt{T} + R_C q R_{x_0} \cdot (m + \log T)\right).$$

*(b). With $w = 0$,*

$$\sum_{t=1}^{T} \|\hat{y}_t - y_t\| - \min_{\pi \in \Pi_L\left(0, q, \left[0, 1 - \frac{1}{\sqrt{T}}\right] \cup \{1\} \cup \mathbb{D}_{1-\gamma}\right)} \sum_{t=1}^{T} \|\hat{y}_t^\pi - y_t\| \leq O\left(QRm^2 \log_+(R_C q R_{x_0}/R) \log^7(mT) \cdot \sqrt{T}\right).$$

*(c). For any $\rho \in (0, \gamma]$, with $R' = R + \frac{W}{\rho}$,*

$$\sum_{t=1}^{T} \|\hat{y}_t - y_t\| - \min_{\pi \in \Pi_L(w, q, [0, 1-\rho] \cup \mathbb{D}_{1-\gamma})} \sum_{t=1}^{T} \|\hat{y}_t^\pi - y_t\| \leq O\left(Q \cdot R' m^2 \log^7(mT) \cdot \sqrt{T}\right)$$
$$+ O\left(\frac{R' \log_+(R_C q R_{x_0}/R') + R_C q w}{\rho}\right).$$

*Proof.* By the observer system construction, the approximation result that constitutes the first part of the proof of Lemma C.4 extends immediately to this agnostic case. Then, since OGD's guarantee is given in the regret formulation of online convex optimization, we get the stated results. □

The above statements describe the performance of Algorithm 1, which is a combination of spectral filtering and regression. However, our techniques yield a sharp analysis for other types of algorithms. For simplicity, we consider two additional algorithms: (1) regression by itself and (2) spectral filtering combined with the Chebyshev sequence preconditioning of Marsden & Hazan (2025). We will state the corresponding results for the nonlinear case with minimal proofs – the reader can fill in the details, as well as any statements for the linear case.

**Theorem G.3.** *Let $y_1, \ldots, y_T$ be an arbitrary sequence bounded by $R$. Let $\Pi_{NL}(q, \gamma)$ be the class of nonlinear dynamical systems $(f, h, x_0)$ such that $\|x_t\| \leq R$ and $Q_\star(f, h, \frac{1}{T^{3/2}}, \mathbb{D}_{1-\gamma}) \leq q$. Then, running Algorithm 2 with $m \geq \frac{1}{\gamma} \log\left(\frac{T}{\gamma}\right)$, $\mathcal{K} = R_D$ for $D = \Theta(mq^2)$, and $\eta_t = \Theta\left(\frac{q^2}{R\sqrt{t}\log T}\right)$ produces a sequence of predictions $\hat{y}_1, \ldots, \hat{y}_T$ for which*

$$\sum_{t=1}^{T} \|\hat{y}_t - y_t\| - \min_{\pi \in \Pi_{NL}(q, \gamma)} \sum_{t=1}^{T} \|\hat{y}_t^\pi - y_t\| \leq O\left(q^2 \log(q) \cdot d_{\mathcal{X}} R \log(RT) m^2 \log^7(mT) \cdot \sqrt{T}\right).$$

---

**Algorithm 2** Regression

---

1: **Input:** Horizon $T$, number of autoregressive components $m$, step sizes $\eta_t$, convex constraint set $\mathcal{K}$, prediction bound $R$.
2: Initialize $P_j^0 \in \mathbb{R}^{d_{out} \times d_{out}}$ for $j \in \{1, \ldots, m\}$.
3: **for** $t = 0, \ldots, T-1$ **do**
4:     Compute

$$\hat{y}_t(\Theta^t) = \sum_{j=1}^m P_j^t y_{t-j},$$

    and if necessary project $\hat{y}_t(P^t)$ to the ball of radius $R$ in $\mathcal{Y}$.
5:     Compute loss $\ell_t(P^t) = \|\hat{y}_t(P^t) - y_t\|$.
6:     Update $P^{t+1} \leftarrow \Pi_{\mathcal{K}}\left[P^t - \eta_t \nabla_P \ell_t(P^t)\right]$.          # can be replaced with other optimizer
7: **end for**

---

*Proof sketch.* As before, we use Lemma C.5 to perform a global linearization via discretization. We then use eigenvalue placement to place all the eigenvalues of an observer system into $[0, 1 - \gamma]$, and the dynamics matrix of this observer system will have spectral condition number $q$. The regression part of the bound from Lemma C.4 together with OGD's regret guarantee lets us conclude. $\qquad\square$

**Remark G.4.** *Consider a system with $N$ marginally stable modes, such as the permutation system of size $N$ from Example F.3. For such a system, $Q_\star$ is exponential in $N$ when we take the spectral constraint set $\Sigma = \mathbb{D}_{1-\gamma}$. The regret bound tells us that we require $T$ to also be exponential in $N$, which means we must have $m$ linear in $N$. This recovers what we would expect from the Cayley-Hamilton theorem, which implies that any LDS with hidden dimension $d_h$ should be learnable with $d_h$ autoregressive terms. This also exemplifies a main benefit that spectral filtering offers: it allows for $Q_\star$ to be much smaller when there are real and marginally stable eigenvalues, which in turn allows for learnability using fewer parameters.*

---

**Algorithm 3** Observation Spectral Filtering + Chebyshev

---

1: **Input:** Horizon $T$, number of filters $h$, step sizes $\eta_t$, convex constraint set $\mathcal{K}$, prediction bound $R$.
2: Compute $\{(\sigma_j, \phi_j)\}_{j=1}^k$ the top eigenpairs of the $(T-1)$-dimensional Hankel matrix of Hazan et al. (2017).
3: Initialize $N_i^0 \in \mathbb{R}^{d_{out} \times d_{out}}$ for $i \in \{1, \ldots, h\}$.
4: **for** $t = 0, \ldots, T-1$ **do**
5:     Compute

$$\hat{y}_t(N^t) = -\sum_{j=1}^{\log T} c_j y_{t-j} + \sum_{i=1}^h \sigma_i^{1/4} N_i^t \langle \phi_j, y_{t-1:t-T} \rangle,$$

    and if necessary project $\hat{y}_t(N^t)$ to the ball of radius $R$ in $\mathcal{Y}$.
6:     Compute loss $\ell_t(N^t) = \|\hat{y}_t(N^t) - y_t\|$.
7:     Update $N^{t+1} \leftarrow \Pi_{\mathcal{K}}\left[N^t - \eta_t \nabla_N \ell_t(N^t)\right]$.          # can be replaced with other optimizer
8: **end for**

---

**Theorem G.5.** *Let $y_1, \ldots, y_T$ be an arbitrary sequence bounded by $R$. Let $\Pi_{\mathrm{NL}}(q)$ be the class of nonlinear dynamical systems $(f, h, x_0)$ such that $\|x_t\| \le R$ and $Q_\star\left(f, h, \frac{1}{T^{3/2}}, \Sigma\right) \le q$ for*

$$\Sigma := \left\{\lambda \in \mathbb{D}_1 : \quad |\arg(\lambda)| \le \frac{1}{64 \log^2(T)}\right\}.$$

*Then, running Algorithm 3 with $h = \Theta(\log^2(T))$, $\mathcal{K} = R_D$ for $D = \Theta(q^2)$, and $\eta_t = \Theta\left(\frac{q^2}{R\sqrt{t}\log T}\right)$ produces a sequence of predictions $\hat{y}_1, \ldots, \hat{y}_T$ for which*

$$\sum_{t=1}^T \|\hat{y}_t - y_t\| - \min_{\pi \in \Pi_{NL}(q)} \sum_{t=1}^T \|\hat{y}_t^\pi - y_t\| \le \widetilde{O}\left(q^2 \cdot d_{\mathcal{X}} R \cdot T^{10/13}\right).$$

*Proof sketch.* Form the observer system with a choice of $L$ such that all eigenvalues of $A - LC$ have complex component less than $\delta = \frac{1}{64 \log^2(T)}$. Apply Theorem C.2 of Marsden & Hazan (2025), where the same trick as in Lemma C.4 may be used to improve the dependence on $R_C$. $\qquad\square$

Lastly, we recall from the proof of Lemma C.4 that the approximation error can be made independent of $Q_\star$ with a cost of $\log Q_\star$ extra parameters, though the diameter of the constraint set will still depenend linearly on $Q_\star$. More sophisticated online algorithms can have logarithmic dependence on the diameter, yielding the following result. As before, we state this only for the nonlinear realizable case, and the linear case (as well as the agnostic cases) are proved along the same lines.

**Theorem G.6.** *Let $y_1, \ldots, y_T$ be the observations of a nonlinear dynamical system satisfying Assumption 3.1, and let $\gamma \in (0,1)$. Let $Q_\star = Q_\star\left(f, h, \frac{1}{T^{3/2}}, \left[0, 1 - \frac{1}{\sqrt{T}}\right] \cup \{1\} \cup \mathbb{D}_{1-\gamma}\right)$ as specified in Definition C.7. Fix the number of algorithm parameters $h = \Theta\left(\log^2(Q_\star T)\right)$ and $m = \Theta\left(\frac{1}{\gamma} \log\left(\frac{Q_\star T}{\gamma}\right)\right)$, and adjust Algorithm 1 (with $J^t, M^t \equiv 0$) to run unconstrained with the Vovk-Azoury-Warmuth forecaster for the square loss. Then, the predictions $\hat{y}_1, \ldots, \hat{y}_T$ satisfy*

$$\sum_{t=1}^{T} \|\hat{y}_t - y_t\|^2 \leq O\left(\log^2(Q_\star) \cdot d_\mathcal{X} R^2 \log(RT) \log(T) \cdot \sqrt{T}\right).$$

*Proof sketch.* With this choice of parameters, after constructing the discretized LDS according to Lemma C.5, the approximation bound from the proof of Lemma C.4 says that there exists a choice of parameters for which the summed squared losses are bounded by $O\left(\log^2(Q_\star) \cdot d_\mathcal{X} R^2 \log(RT) \sqrt{T}\right)$. The regret bound from equation (3.12) of Azoury & Warmuth (2001) adds $O\left(R^2 \log((m+h)Q_\star T)\right)$ from the optimization. $\qquad\square$

The above result tells roughly the same story as our main Theorem 4.1: as long as only a small portion of the Koopman eigenvalues are undesirable, spectral filtering has a strong regret guarantee. This statement of the result better trades dependence on $Q_\star$ in the regret bound for an increase in the number of parameters.

# H    MULTI-DIMENSIONAL OBSERVATIONS

In this section, we discuss what changes when allowing higher $d_\mathcal{Y}$. To begin, in such settings we can have $(A, C)$ being observable even if $A$ has an eigenvalue with multiplicity $d_\mathcal{Y}$, and so eigenvalue placement will allow us to move undesirable eigenvalues with multiplicity. In addition, while Ackermann's formula as stated no longer holds, the bound

$$\|L_\star\| \leq \frac{2Q_\star}{\sigma_{\min}(C)}$$

of Lemma F.2 continues to hold. The regret guarantee of Lemma C.4 depends on $\|L_\star\|_F \cdot \|C\|_F$ in a sharp way, and we use that $\|L_\star\|_F = \|L_\star\|$ and $\|C\|_F = \|C\| = \sigma_{\min}(C)$ in the 1-dimensional observation case in order to simplify several expressions. While in most places our bounds will simply introduce a $\sqrt{d_\mathcal{Y}}$ factor gotten from converting between operator and Frobenius norms, we will also see a $\frac{\|C\|}{\sigma_{\min}(C)}$ factor in the regret bounds of Theorems 4.1 and A.2.

To summarize, increasing the observation dimension (1) strengthens eigenvalue placement, (2) adds $\sqrt{d_\mathcal{Y}}$ factors to many of the norms, and (3) creates a difference between the max and min singular values of $C$, which will show up in the $\|L_\star\|_F \cdot \|C\|_F$ terms in the regret bounds. It is difficult to get control on the conditioning of $C$ in our Markov chain construction, but morally everything stated remains true.

# I    THEORETICAL SUBTLETIES

In this section, we discuss a few nuances in our theory. Some appear paradoxical at first glance, but with more care they yield useful information and intuition about these proof techniques.

**Strong stability.**    To begin, since we can construct the observer system with whichever poles we want, one could ask "why not just make $A - LC$ to be $\gamma$-stable for $\gamma = \frac{1}{2}$ and use $\widetilde{\Theta}\left(\frac{1}{\gamma}\right)$ regression terms?". The answer, as explored a bit in Remark G.4, is that the number of autoregressive terms needed also scales with the spectral condition number

of the resulting observer system, i.e. one would actually need $\widetilde{\Theta}\left(\frac{\log Q_\star}{\gamma}\right)$ many regression terms. There is a price to pay for more extreme eigenvalue placement given by $Q_\star$, and the cost of a fully stable observer system is seen by the number of regression terms needed to learn it. To drive this point home, if we consider a symmetric system with $N$ many marginally stable eigenvalues, regression would require $N$ parameters, whereas spectral filtering can get the job done independently of $N$.

**Use of observations.** Next, since we use pole placement for two purposes (to change the system eigenstructure and to construct an LDS over observations against which to apply spectral filtering), one could ask "what value of $L$ should be taken if the system already has desirable spectral structure?". For example, if we start with a symmetric autonomous LDS $x_{t+1} = Ax_t$ and $y_t = Cx_t$, there is no immediate input-output sequence-to-sequence mapping against which spectral filtering should hope to compete against. The classical way to enforce real-diagonalizability and marginal stability of $A - LC$ in this case would be to choose $L = 0$, but this does not fix the problem that there is no natural $[1, \alpha, \alpha^2, \ldots, \alpha^T]$ structure to an input-output mapping. The solution is to recognize that spectral filtering competes against *all* observer systems – for the choice of $L = 0$ we pay the full price of the initial state, and if $A$ is marginally stable we cannot hope for any decay. However, one can choose $L$ nonzero but such that $A - LC$ remains desirable (for example, if $A$ has an eigenvalue strictly less than 1 we can move it and only it, whereas if $A$ has all eigenvalues 1 then we can take $L$ proportional to $C^\top$). In this way, we construct an observer LDS with a nontrivial input-output mapping which retains the desirable spectral structure, and spectral filtering can efficiently learn to represent this.

## J  NONLINEAR SYSTEMS WITH STOCHASTIC NOISE

We have discussed deterministic systems of the form $x_{t+1} = f(x_t)$ with $x_0$ fixed. This yields a Koopman operator sending $h(x) \mapsto h(f(x))$ and and a Markov chain which tracks the deterministic transitions. Both of these notions can be extended seamlessly to the case of stochastic systems of the form

$$x_{t+1} = f(x_t) + w_t, \quad y_t = h(x_t), \quad x_0 \sim \mu_{x_0}$$

for an initial condition drawn from $\mu_{x_0}$, where $w_t \sim \mu_w$ are i.i.d. stochastic noises. For example, applying the Koopman operator $n$ times sends the function $h(x) \mapsto \mathbb{E}[f(x_n) \mid x_0 = x]$, and the Markov chain simply gets non-integer entries that record the stochastic transition probabilities. As long as we assume that $\|x_t\|, \|y_t\| \leq R$ almost surely, the discretization argument still holds exactly as before, where now $y_t' = C'z_t$ approximates $\mathbb{E}[y_t]$. Note that we need not assume anything about the distributions $\mu_w, \mu_{x_0}$ themselves – boundedness of states and observations is enough. The above reasoning is sufficient to prove:

**Corollary J.1.** *Let $x_{t+1} = f(x_t) + w_t$ be a stochastic dynamical system with observation function $h$ producing a sequence $y_1, \ldots, y_T$ of observations, where $w_t \sim \mu_w$ i.i.d. and $x_0 \sim \mu_{x_0}$. Assume that for all $t$, $\|x_t\|, \|y_t\| \leq R$ almost surely, $f, h$ are 1-Lipschitz, and the discretized linear lifting $(A', C')$ from Lemma C.5 with stochastic transitions is an observable LDS.*

*Fix the number of algorithm parameters $h = \Theta(\log^2 T)$ and $m \geq 1$, and for notation define $\gamma = \frac{\log(mT)}{m}$. Let $Q_\star = Q_\star\left(A', C', \left[0, 1 - \frac{1}{\sqrt{T}}\right] \cup \{1\} \cup \mathbb{D}_{1-\gamma}\right)$ as specified in Definition C.1. Then, running Algorithm 1 with $J^t, M^t \equiv 0$, $\mathcal{K} = R_D$ for $D = \Theta((m + h)Q_\star^2)$, and $\eta_t = \Theta\left(\frac{Q_\star^2}{R\sqrt{t}\log T}\right)$ produces a sequence of predictions $\hat{y}_1, \ldots, \hat{y}_T$ for which*

$$\mathbb{E}\sum_{t=1}^T \|\hat{y}_t - \mathbb{E}y_t\| \leq O\left(Q_\star^2 \log(Q_\star) \cdot d_\mathcal{X} R \log(RT) m^2 \log^7(mT) \cdot \sqrt{T}\right).$$

*Proof.* The proof goes the same way as that of Theorem 4.1. We form the discretized Markov chain $A', C'$, where now the transition matrix is given by the transition probabilities from one grid point to another after quantization. More precisely, we have

$$A_{ij}' := \mathbb{P}_{w \sim \mu_w}\left(\pi\left(f(s^{(j)}) + w\right) = s^{(i)}\right).$$

The LDS is then

$$z_{t+1} = A'z_t, \quad y_t' = C'z_t, \quad z_0 = \pi_\# \mu_{x_0},$$

where $\pi$ is the projection to the grid (i.e. $z_0$ is the probability vector for the location of the initial state). Since each $z_t$ is the law of the grid state of the discretized system, we see that $y_t' = \mathbb{E}[(h \circ \underbrace{T \circ \ldots \circ T}_{t \text{ times}} \circ \pi)(x_0)]$ in the language of

Lemma C.5. Applying the same Lipschitz argument allows us to bound the difference between this discretized rollout and the true trajectory almost surely along the whole path, which implies the same bound on the size of the difference between their expectations. From here on we have again reduced to competing against the predictions of a deterministic LDS. Note that by the Lipschitzness argument, since $f(x_t) + w_t$ is a.s. bounded by $R$ by assumption, we will have $f(s_t) + w_t$ bounded by $R + t\varepsilon \leq R + \frac{1}{\sqrt{T}}$ always. This means that under the assumptions we imposed, we can use an $\varepsilon$-net of the ball of size $O(R)$ regardless of how large $w_t$ is.

The final subtlety is that our predictions $\hat{y}_{t+1}$ are in fact random, since we filter over $y_1, \ldots, y_t$ instead of the deterministic $y'_t$ which is needed to compete against the Markov chain LDS. This introduces an additional error of size $\|L\| \cdot \|y_t - y'_t\|$, which gets upper bounded by $Q_\star \cdot t\varepsilon$ almost surely. This gets absorbed into our other terms. $\square$

