# OpenReview forum: "Universal Learning of Nonlinear Dynamics"
_ICLR.cc/2026/Conference — Submitted to ICLR 2026_

### Official Review · Reviewer_xUFb · 2025-10-27

**Soundness:** 2
**Presentation:** 1
**Contribution:** 2
**Rating:** 4
**Confidence:** 3

**Summary:**

Based on  spectral filtering, this paper proposes an algorithm to improperly learn  a marginally stable unknown nonlinear dynamical
system. The algorithm can predict future outputs   from past observations with  vanishing prediction error for any nonlinear dynamical system that has finitely many marginally stable modes. Experimental results on next-token prediction numerically demonstrate its effectiveness over several methods.

**Strengths:**

1: The paper generalizes the original spectral filtering algorithm designed for  linear dynamical systems with the symmetric transition matrix  to both asymmetric dynamics as well as incorporating noise correction.

2: The authors introduce a simple construction to approximate any bounded, Lipschitz nonlinear system with a high-dimensional LDS and further provide a novel analytical framework to  derive learning guarantees in nonlinear settings using linear methods.

**Weaknesses:**

1: The authors overlook a substantial body of related work on learning nonlinear dynamical systems of the form (1.1). For example, many recent studies have investigated the use of Gaussian processes to approximate both the transition and observation functions to learn such systems [1-3].

2: For the identification of linear dynamical systems mentioned in Appendix A, many methods can be used to learn system matrices, such as subspace state-space system identification; however, these methods are not discussed by the authors.

3: The paper is difficult to follow due to a lack of explanation of technical details. For instance, while the proposed algorithm is based on spectral filtering, the authors do not provide a detailed description of this technique in the paper.

[1] Fan X, Bonilla E V, O’Kane T, Sisson S A. Free-form variational inference for Gaussian process state-space models, International Conference on Machine Learning. 2023 : 9603-9622.

[2] Frigola R, Chen Y, Rasmussen C E. Variational Gaussian process state-space models. Advances in neural information processing systems, 27, 2014.

[3] Turner R, Deisenroth M, Rasmussen C. State-space inference and learning with Gaussian processes, Proceedings of the thirteenth international conference on artificial intelligence and statistics. 2010 : 868 – 875.

**Questions:**

1: In line 149, could the authors explain how to fold the terms $D$ and $\xi_t$ into the input of the previous iteration?

2: How to derive the Algorithm 1?

3: In Algorithm 1, how to determine $h,m$  and $\eta_t$. Moreover, it would be helpful to clarify whether these parameters influence the algorithm’s performance. In addition, what is $\Pi_\mathcal{k}$ in line 208.

4: Could the authors elaborate on how their proposed algorithm extends or improves spectral filtering approaches, and clarify the specific distinctions from prior methods.

5: In line 196, the authors define that $\Theta^t=(J^t,M^t,P^t,N^t)$. Hence, what does $\Theta(\cdot)$ mean in line 222. The notation is somewhat confusing.

6: In experiments, the baseline methods used for comparison are relatively simple. The authors are encouraged to include additional learning methods for nonlinear dynamical systems mentioned in Weakness for a more comprehensive evaluation.

7: The proposed algorithm seems to perform prediction only on the collected output samples, rather than truly forecasting future outputs beyond the observed data.

---

> ### Author Response · Authors · 2025-11-24
> **Response to Reviewer SGuc**
>
> We thank the reviewer for the thoughtful and constructive feedback. Below we address each point in turn.
>
> **Responses to the Weaknesses**
>
> *W1. The authors overlook a substantial body of related work on learning nonlinear dynamical systems of the form (1.1)...*
>
> *A.* The GP-based literature on learning nonlinear dynamical systems (e.g., variational GPSSMs and free-form GP inference) focuses on *properly* learning the latent transition and observation functions by placing a GP prior on both. All such methods inherently require explicit reconstruction or approximation of the hidden state sequence, using the observation sequence.
>
> In contrast, the central contribution of our work is to deviate from this line of research by introducing an *improper* learning approach. We do not attempt to recover the true latent states or the true transition function; instead, we construct a high-dimensional linear surrogate that only needs to predict outputs, and we analyze an online spectral-filtering algorithm that operates entirely in the observable space. This closely models the way in which existing solutions for the next token prediction problem work, in which the next observation is predicted using a parameterized function of previous observations, which is updated at every step using gradient descent.
>
> We thank the reviewer for pointing out this line of work, and we will include a discussion of these papers in our related-work section.
>
> *W2. For the identification of linear dynamical systems mentioned in Appendix A, many methods can be used to learn system matrices...*
>
> *A.* It is true that there are many system identification algorithms for LDS, including subspace/Krylov-type methods. That being said, such methods can suffer from hidden-state dimension, marginal stability, and adversarial noise as stated in Table 1, which makes them unsuitable for our application. In general, subspace methods tend to be good when the system is separable into low-dimensional parts with simple interactions between them, which is often not the case for linearizations of nonlinear systems. We thank the reviewer for bringing this to our attention.
>
> *W3. The paper is difficult to follow due to a lack of explanation of technical details. For instance,...*
>
> *A.* We appreciate this comment. Spectral filtering was introduced by Hazan et al. (2017), and has since been discussed in multiple papers and as the main topic of Chapter 11 of the textbook “Introduction to Online Control” by Hazan and Singh. Given the space constraints and the fact that it is an established technique with pedagogical expositions (blog posts and a textbook chapter) freely available, we consider it sufficient to cite the relevant resources.
>
> **Responses to the Questions**
>
> *Q1. In line 149, could the authors explain how to fold the terms  and  into the input of the previous iteration?*
>
> *A.* Suppose that $x_t$ denotes the original hidden states (of dimension $N$). One can consider augmenting the hidden state to $z_t$ (of dimension $N+d_y$): the new $A$ matrix will zero out these coordinates, the new $B$ matrix will write to these coordinates the contribution of $Du_t$, the new process noise $w_t$ will have the measurement noises $\xi_t$ on these coordinates, and the new $C$ will sum the old output with the value of these coordinates.
>
> In short, process noise is strictly stronger than measurement noise. We can always imagine measurement noise as “process noise on a coordinate of the hidden state which is forgotten every step”.
>
> *Q2. How to derive the Algorithm 1?*
>
> *A.* Algorithm 1 is more or less the naive, first-attempt way to apply spectral filtering to a nonlinear LDS (it is the same as Algorithm 1 of Hazan et al. (2017), see also Algorithm 7 of Chapter 11 of “Introduction to Online Control” by Hazan-Singh). The derivation of Algorithm 1 is given in full detail in Appendices C and D. Appendix C constructs the lifted LDS used to approximate the nonlinear predictor, and Appendix D provides the spectral filtering machinery and regret analysis.

---

> > ### Author Response · Authors · 2025-11-24
> > **Additional Responses**
> >
> > *Q3. In Algorithm 1, how to determine...*
> >
> > *A.* We thank the reviewer for this question. The parameter $h$ is the number of filters used: in theory it should be $log^2(T)$ for $T$ the prediction time horizon, but in practice it can usually be set to $16$ or $24$ or so (this is more a hyperparameter one chooses based on computational restrictions, and can be viewed like the “width” of the spectral filtering layer). We have not observed any benefit to going beyond this. The parameter $m$ is the number of regression terms: our theory handles any choice of $m$ that the user wants, and again it is a hyperparameter. If the reviewer wants to think of Algorithm 1 as a single layer of a sequence-to-sequence model, then $h$ and $m$ are simply the hyperparameters for the size of that layer. The parameters $\eta_t$ are the learning rate – while there is a choice of learning rate which is used to apply the regret guarantee of online gradient descent, in practice we recommend a hyperparameter sweep (and most likely practitioners will use Adam with their favorite decay schedule, a parameter-free method, etc.). Since any optimizer with convex optimization guarantees can be used with Algorithm 1 and each will have its own way to choose learning rate, there is unfortunately not much more we can say without specifying an optimizer.
> >
> > Regarding the second question, $\Pi_K$ denotes the projection operator onto the convex set $K$. We shall state this for clarity in the notation section (Section 3.1).
> >
> >
> > *Q4. Could the authors elaborate on how their proposed algorithm extends or improves spectral filtering approaches, and clarify the specific distinctions from prior methods.*
> >
> > *A.* Prior spectral-filtering methods (Hazan et al., 2017) handle only linear dynamical systems. Our contribution is to show that nonlinear marginally stable systems satisfying Assumption 3.1 can be approximated by a lifted LDS in a high-dimensional space. This allows spectral filtering to learn the outputs of nonlinear systems by competing against a carefully constructed linear surrogate. The extension is nontrivial because it requires new structural assumptions, approximation lemmas, and stability arguments. We emphasize that our contribution is a novel analysis of spectral filtering: Algorithm 1 is the usual spectral filtering algorithm, but prior to this work there were no guarantees for this algorithm on nonlinear systems (or asymmetric systems or adversarial noise).
> >
> > *Q5. In line 196, the authors define that...*
> >
> > *A.* In line 196, $\Theta$ denotes the collection of algorithmic parameters, whereas in line 222 it denotes the standard big-$\Theta(\cdot)$ asymptotic notation. We will rename the parameter tuple to $\Gamma$ to avoid confusion.
> >
> > *Q6. In experiments, the baseline methods used for comparison are relatively simple...*
> >
> > *A.* We agree that nonlinear dynamical systems can be approached through a wide range of powerful modeling techniques, including GP-based state-space models and other latent-variable approaches. However, as discussed in our rebuttal, these methods operate in a fundamentally different regime from ours: they perform proper learning of the latent transition function and require explicit inference over the latent state trajectory, making them applicable only when latent-state reconstruction is feasible and computationally affordable. In contrast, our work introduces an improper learning framework that operates entirely in the observable space and does not attempt to reconstruct latent states or estimate the true nonlinear dynamics. As such, the appropriate baselines for our method are those that share this output-only, improper-learning perspective. For this reason, we compared against algorithms that are methodologically aligned with our approach.

---

> > > ### Author Response · Authors · 2025-11-24
> > > **Additional Responses**
> > >
> > > *Q7. The proposed algorithm seems to perform prediction only on the collected output samples, rather than truly forecasting future outputs beyond the observed data.*
> > >
> > > *A.* Our setting focuses on the next-token prediction task, where the learner predicts $\hat{y}_{t+1}$​ using ground-truth observations $(y_1,…,y_t)$. This is a natural abstraction of a fundamental problem in modern machine learning, with applications including time series modeling (stock prices, weather, etc.), sequential decision making, training of LLMs and generative approaches to inverse problems, and so on. We will clarify in the revision that our theory focuses on this online prediction task.
> > >
> > > We strongly agree with the reviewer that theoretical guarantees for autoregressive rollout are crucial. There are such results for system identification-based methods, but it is often difficult to get a handle on the “train for next token prediction, test on rollout” behavior of improper predictors. One can sometimes even come up with adversarial examples causing such predictors to fail. However, in practice we see that improper predictors can do very well: consider the transformer in an LLM (it is an improper predictor trained on next-token prediction and used in rollouts) as opposed to a state-space model (which is a proper learner of LDS). There is certainly a mystery that needs to be clarified by theoretical investigation of rollout behavior of improper predictors. The current theory is not set up to describe this application at all, and we consider it a very valuable direction for future work.
> > >
> > > For completeness, we note that Figure 2 includes fully autoregressive experiments on the Lorenz attractor, wherein predictions $\hat{y}_{t+1}$​ are generated using previous predicted values $(\hat{y}_1,…,\hat{y}_t)$, and our method appears to perform well in this setting too (assuming an expressive enough set of observables). Our current theory does not attempt to explain this.

---

> > > > ### Comment · Reviewer_xUFb · 2025-11-27
> > > >
> > > > Many thanks for the authors' response and I will keep my score.
> > > >
> > > > 1. In fact, GP-based methods  use GP to approximate nonlinear functions, rather than to learn their explicit expressions. Hence,  it is inappropriate to claim that they are proper learning approaches. In addition,  a central goal of GP-based methods is also to perform prediction. Hence,  comparing the proposed algorithm with GP-based methods is necessary to demonstrate its effectiveness.
> > > >
> > > > 2. Could you provide a mathematical description of the answer to Q1?

---

> > > > > ### Author Response · Authors · 2025-11-28
> > > > > **response to reviewer**
> > > > >
> > > > > 2. We provide the mathematical description to Q1 below.
> > > > >
> > > > > Consider the original linear dynamical system
> > > > > \begin{align}
> > > > >     x_{t+1} &= A x_t + B u_t + w_t, \\
> > > > >     y_t^{\mathrm{orig}} &= C x_t + D u_t + \xi_t,
> > > > > \end{align}
> > > > > where $x_t \in \mathbb{R}^n$, $u_t \in \mathbb{R}^m$, $y_t^{\mathrm{orig}} \in \mathbb{R}^{d_y}$,
> > > > > $w_t \in \mathbb{R}^n$, and $\xi_t \in \mathbb{R}^{d_y}$.
> > > > >
> > > > > Our goal is to construct an equivalent LDS whose output satisfies
> > > > > \[
> > > > >     y_t = y_{t-1}^{\mathrm{orig}} = C x_{t-1} + D u_{t-1} + \xi_{t-1}.
> > > > > \]
> > > > >
> > > > > \paragraph{Augmented State.}
> > > > > Define the augmented hidden state
> > > > > \[
> > > > >     z_t :=
> > > > >     \begin{bmatrix}
> > > > >         x_{t-1} \\
> > > > >         u_{t-1} \\
> > > > >         \xi_{t-1}
> > > > >     \end{bmatrix}
> > > > >     \in \mathbb{R}^{n + m + d_y}.
> > > > > \]
> > > > >
> > > > > \paragraph{Output Matrix.}
> > > > > We choose
> > > > > \[
> > > > >     C' := \begin{bmatrix} C & D & I_{d_y} \end{bmatrix},
> > > > > \]
> > > > > so that
> > > > > \[
> > > > >     y_t = C' z_t = C x_{t-1} + D u_{t-1} + \xi_{t-1}
> > > > >     = y_{t-1}^{\mathrm{orig}}.
> > > > > \]
> > > > >
> > > > > \paragraph{Dynamics.}
> > > > > Using the identity
> > > > > \[
> > > > >     x_t = A x_{t-1} + B u_{t-1} + w_{t-1},
> > > > > \]
> > > > > we want $z_{t+1}$ to equal
> > > > > \[
> > > > >     z_{t+1} =
> > > > >     \begin{bmatrix}
> > > > >         x_t \\
> > > > >         u_t \\
> > > > >         \xi_t
> > > > >     \end{bmatrix}.
> > > > > \]
> > > > > This is achieved by choosing
> > > > > \[
> > > > >     A' :=
> > > > >     \begin{bmatrix}
> > > > >         A & B & 0 \\
> > > > >         0 & 0 & 0 \\
> > > > >         0 & 0 & 0
> > > > >     \end{bmatrix},
> > > > >     \qquad
> > > > >     B' :=
> > > > >     \begin{bmatrix}
> > > > >         0 \\
> > > > >         I_m \\
> > > > >         0
> > > > >     \end{bmatrix},
> > > > > \]
> > > > > and process noise
> > > > > \[
> > > > >     w'_t :=
> > > > >     \begin{bmatrix}
> > > > >         w_{t-1} \\
> > > > >         0 \\
> > > > >         \xi_t
> > > > >     \end{bmatrix}.
> > > > > \]
> > > > >
> > > > > Then the augmented system satisfies
> > > > > \[
> > > > >     z_{t+1} = A' z_t + B' u_t + w'_t,
> > > > > \]
> > > > > and for all $t \ge 1$,
> > > > > \[
> > > > >     y_t = C' z_t = y_{t-1}^{\mathrm{orig}}.
> > > > > \]
> > > > >
> > > > > \paragraph{Conclusion.}
> > > > > Thus, by augmenting the hidden state with $(x_{t-1}, u_{t-1}, \xi_{t-1})$ and using the matrices above,
> > > > > we obtain an LDS of the form
> > > > > \[
> > > > >     z_{t+1} = A' z_t + B' u_t + w'_t,
> > > > >     \qquad
> > > > >     y_t = C' z_t,
> > > > > \]
> > > > > whose output equals the original system's output with a one-step time shift.

---

> > > > > > ### Author Response · Authors · 2025-11-28
> > > > > > **Response to the Reviewer**
> > > > > >
> > > > > > We examined the GP-based references the reviewer cites. All three papers explicitly formulate a state-space model in which a Gaussian process prior is placed on the latent transition function. Although this transition function is not learned in closed form, the GP posterior defines a full dynamical model whose hidden state must be recursively propagated in order to generate predictions. In this sense, these approaches are “proper learners” of a dynamical system: they learn a transition map governing the evolution of a latent state and use that learned map to perform multi-step prediction.
> > > > > >
> > > > > > It is in this precise sense that we refer to GP-based approaches as “proper learning” methods. They differ fundamentally from our method, which bypasses reconstruction of a latent transition function and instead produces predictions directly in the observable space via spectral filtering. Under this terminology, a “proper learner” is any prediction method that must explicitly evolve a hidden state using a learned or estimated transition function. GP-based LDS models fall squarely into this category, just as a neural-network parametrization of the transition map would: both methods approximate a nonlinear function (rather than recovering an explicit symbolic expression), but both still require forward simulation of the latent state to make predictions.
> > > > > >
> > > > > > Finally, while GP-based methods are powerful, their sample and computational complexity scale with the effective dimension of the latent state, and there exist high-dimensional LDS examples for which they provably fail to learn accurate dynamical models. This is precisely why a comparison between our improper learning approach and proper GP-based learners reflects a meaningful conceptual distinction, rather than a semantic one.

---

### Official Review · Reviewer_SGuc · 2025-10-27

**Soundness:** 2
**Presentation:** 2
**Contribution:** 2
**Rating:** 4
**Confidence:** 3

**Summary:**

This paper presents a novel method to learn an LDS observer for marginally stable nonlinear systems with finitely many modes. The proposed approach leverages Spectral filtering theory for marginally stable LDS, yielding an efficient algorithm based on online convex optimization.

**Strengths:**

1. Learning an one-step output predictor in the online learning setup is an interesting topic.

2. The spectral filtering method yields a convex approximation of the long-horizon behavior of a marginally stable LDS with relatively small set of parameters. This feature offers an efficient online least square regression for problems that require long-horizon trajectory information.

**Weaknesses:**

1. **Overstated Claims**

The title and several statements (e.g., the first sentence of the abstract) substantially overstate the scope of the work relative to the specific technical problem it addresses. The paper is **not** about learning a full dynamical system. Instead, it treats **state estimation + one-step prediction** as a black-box problem and learns an adaptive online (time-varying) least-squares regression (LSR) model from the input–output trajectory. This is distinct from the conventional notion of *learning nonlinear dynamics*, which entails the ability to perform multi-step prediction once an accurate state estimate is obtained from past data. In the proposed approach, the model depends on the true system output $y_{t+1}$ to predict $\hat{y}_{t+2}$. If the true $y$ is replaced with the observer output $\hat{y}$, then the prediction error would increase dramatically.

2. **Overly strong assumption on the nonlinear system**

The paper relies on a strong assumption that the underlying data-generating process (the nonlinear system) is stable. However, the chaotic examples in Section 7 do not satisfy this assumption; yet their trajectories can still be estimated well from output data. In nonlinear observer theory, assuming strong stability conditions such as $\mathrm{Lip}(f)\leq 1$ is often overly restrictive. The control literature provides alternative frameworks for relaxing this assumption by using the notion of differential observability, roughly speaking,
$$\sum_{t=0}^{T}| y_{t}^{a}-y_{t}^b | \leq \gamma |x_0^a-x_0^b|$$ for some $\gamma>0$ and $h$ is Lipschitz-bounded.

For such systems, **contracting** (incrementally exponentially stable) observers can be designed [1]. Moreover, a recent work [2] shows that contracting systems can be embedded into stable LDS with the same dimension (not infinite-dimensional). These works help explain why the proposed method may appear to work for chaotic systems, even though they are not marginally stable in the sense assumed by the paper.

3. **Restrictive assumption: finitely many modes**

Nonlinear systems often exhibit *continuous modes*, rather than finitely many discrete modes. Hence, the assumption of finitely many marginally stable modes is quite restrictive and limits the generality of the theoretical claims.

4. **Potential training instability**

The proposed algorithm may suffer from training stability issues. For instance, if the underlying dynamics operate in a specific mode, the learned LDS may experience mode collapse. When the operational mode changes abruptly due to external inputs or disturbances, the training process could become unstable. The authors briefly acknowledge some sort of instability in a footnote. To address this issue, it may need the notion of **persistent excitation** (PE) from the control literature, which is often used to show the stability of online LSR for adaptive control. Roughly speaking, the external input or internal dynamics excites a wide range of modes.

5. **lack of experiments on noisy cases**

In Table 1, the authors claim that OSF can handle adversarial noise. However, the illustrative examples are noise-free, and the theoretical framework itself assumes the absence of measurement noise.

6. **Inaccurate statements on system identification**

The claims about classical system identification (Sys-ID) in Table 1 are inaccurate. System identification has been studied extensively for nearly six decades, and the related work discussed in the paper represents only a small subset of the existing literature, particularly from the ML community. As far as I know, there are many Sys-ID approaches which can handle at least two or three of those requirements listed in Table 1.

[1] W. Lohmiller and J.J.E. Slotine. On Contraction Analysis for Nonlinear Systems. Automatica, 1998.

[2] B. Yi and I. Manchester. On the equivalence of contraction and Koopman approaches for nonlinear stability and control. IEEE-TAC, 2023.

**Questions:**

1. Below Eq. (2.1), the authors wrote "We assume that $D,\xi_t=0$ as these terms can be folded into the input of the previous iteration". I do not understand it. How can one get rid of measurement noise without loss of generality?

---

> ### Author Response · Authors · 2025-11-19
> **Response to Reviewer SGuc**
>
> We thank the reviewer for the thoughtful and constructive feedback. Below we address each point in turn.
>
> **Responses to the Weaknesses**
>
> *W1. Overstated Claims*
>
> *A.* Our problem setting focuses specifically on next-token prediction: given outputs ${y_1,…,y_t}$, the goal is to predict $\hat{y}_{t+1​}$. This corresponds to observing ground-truth data rather than autoregressive predictions. We will state this explicitly in the abstract to emphasize our focus on next-token prediction. Please note that this is an extremely common form of learning in modern tasks, from LLMs to generative models for inverse problems to online control. It is true that this is weaker than a notion of learning that enables autoregressive rollout – this distinction is pointed out in Section 7.1, where we run the algorithm in a multi-step prediction setting (see Figure 2) despite our theory not attempting to cover this case. We do consider this as a very valuable next line of theoretical inquiry, and we thank the reviewer for their emphasis on strengthening the notion of learning in nonlinear systems.
>
> *W2. Overly strong assumption on the nonlinear system*
>
> *A.* We agree that $Lip(f) \leq 1$ is restrictive, although it still includes meaningful classes of marginally stable systems (e.g., permutation systems). We would like to note that the $Lip(f) \leq 1$ notion of stability is only used in our construction of the approximating LDS – any other notion that allows a similar statement as Lemma C.5 will pair nicely with the observer system + spectral filtering combination to produce end-to-end guarantees. In this sense, $Lip(f) \leq 1$ is less of a fundamental restriction of our proof techniques and more so a setting we chose that balances simplicity/generality in order to function as a good environment for our first results in this direction. The door is certainly open to generalize with more precise/subtle/dynamical notions of marginal-stability, and we are particularly excited by the reviewer’s suggested pathwise notion of stability and corresponding contraction arguments. We thank the reviewer for the suggested references and will explore weakening this assumption in future work.
>
> *W3. Restrictive assumption: finitely many modes*
>
> *A.* When $||x_t||$ and $||y_t||$ are uniformly bounded, Lemma C.5 allows us to construct a very high-dimensional linear system approximating the nonlinear one. Thus, Algorithm 1 (OSF) still applies even with continuously many modes. We do not state “finitely many modes” in any theorems/proofs as an assumption, but we use it informally as a concise and intuitive special case that can be stated in a couple words. For precise theoretical claims, the argument of approximation via high-dimensional LDS yields the following statement: if the Koopman operator’s spectrum is contained in the spectral constraint set Σ (which is the union of [-1, 1] and a disk of radius $1 - \gamma$) then $Q_*$ equals $1$, regardless of whether the spectrum is pure point/continuous/residual. If the part of the spectrum which lies outside $\Sigma$ is a discrete set of $N$ elements, then $Q_*$ is exponential in $N$. These notions do not miss the class of nonlinear systems with continuous spectrum, though it is perhaps conveyed in language which is different from the classical Koopman theory.
>
> As a clarifying example consider the doubling map on the circle: it is marginally-stable with continuous spectrum (no nontrivial eigenfunctions on, say, $L^2$). One can construct a linear approximation using discretization + Markov chain (see Ulam’s method for a phrasing of this method which is familiar to Koopman theorists), which will yield LDS approximations with one eigenvalue at 1 (the invariant density) and all other eigenvalues in the disk of radius $1/2$. As such, our method (with $2$ regression coefficients, i.e. $\gamma=1/2$) learns this system effortlessly with a complexity of $Q_* = 1$.
>
> *W4. Potential training instability*
>
> *A.* We would like to emphasize that for our nonlinear guarantees, we do not consider any inputs or excitations. That being said, if there are excitations/mode switches, when learning in an online manner the predictor will adapt and the instantaneous losses that get incurred will be averaged over time. Furthermore, since any reasonable competitor will also suffer from excitations, the regret guarantee still holds. We find the notion of persistent excitation very interesting and appreciate the reviewer’s suggestion in this direction. It is definitely worth investigating in future work, as it would allow extension beyond our task of online next-token prediction.

---

> > ### Author Response · Authors · 2025-11-19
> > **Additional responses**
> >
> > *W5. lack of experiments on noisy cases*
> >
> > *A.* Table 1 is specific to the learning of linear dynamical systems. We demonstrate both theoretically (see Theorem A.2(c)) and empirically (see Figure 3(1)) that in the LDS setting OSF can indeed handle adversarial noise, which we feel substantiates this claim. Appendix J also looks into how i.i.d. stochastic noise may be incorporated in the nonlinear setting. Adversarial disturbances in the nonlinear setting are indeed not captured by our current theory, and we consider it an important direction for future work (though we aren’t very sure what is the right way to approach it yet). Furthermore, the measurement noise can be assumed 0 without loss of generality (it can be absorbed in the process noise, see the response to Question 1).
> >
> > *W6. Inaccurate statements on system identification*
> >
> > *A.* We agree with the reviewer that SysID is a decades-old approach to the prediction of linear dynamical systems, and that most of our related work for LDS learning is from the ML community (though many ideas behind them have roots in classic control theory). That being said, any SysID method will certainly have complexity scaling with the hidden dimension (the system matrix is that big). We are not aware of any SysID method that can handle partially-observed LDS with fully adversarial noise – to our knowledge, the state of the art is the work of Simchowitz-Boczar-Recht (2019) which requires noise within a filtration.
> >
> > **Responses to the Questions**
> >
> > *Q1. Below Eq. (2.1), the authors wrote "We assume that...*
> >
> > *A.* Suppose that $x_t$ denotes the original hidden states (of dimension $N$). One can consider augmenting the hidden state to $z_t$ (of dimension $N+d_y$): the new $A$ matrix will zero out these coordinates, the new $B$ matrix will write to these coordinates the contribution of $Du_t$, the new process noise $w_t$ will have the measurement noises $\xi_t$ on these coordinates, and the new $C$ will sum the old output with the value of these coordinates.
> >
> > In short, process noise is strictly stronger than measurement noise. We can always imagine measurement noise as “process noise on a coordinate of the hidden state which is forgotten every step”.

---

> > > ### Comment · Reviewer_SGuc · 2025-11-27
> > >
> > > I do not agree with the authors' argument on W5 and Q1. That is, measurement noise can be removed without loss of generality.
> > >
> > > Consider an extreme case where the measurement noise is so large that the history of $y_t$ contains almost no useful information about the underlying dynamics. Under such conditions, how can one reliably predict the next output?
> > >
> > > For example, imagine driving a car: we typically anticipate the behavior of the vehicle ahead based on visual observations—such as its speed or brake lights. But if a dense fog (i.e., severe noise) obscures our view, how can we make an accurate prediction of what will happen next?

---

> > > > ### Author Response · Authors · 2025-11-28
> > > > **response to reviewer misunderstanding on W5**
> > > >
> > > > This is a small and easy detail, which we answer below. We ask the reviewer to focus on the main claims of the paper:  very general and important problem, totally new approach to it, with the first provably efficient algorithm that is also shown experimentally to work well.
> > > >
> > > > The details:
> > > > We did not claim that measurement noise can be “removed” without loss of generality in the sense of making the system easier or more predictable. Rather, we show that any LDS with a measurement–noise term can be rewritten as an equivalent higher–dimensional LDS with no explicit measurement-noise term, by absorbing the observation noise into an augmented state and process-noise vector. This transformation preserves the input–output behavior (up to a one-step reindexing) but does not reduce the difficulty of prediction when the measurement noise is large.
> > > >
> > > > In particular, in the reviewer’s example of driving through dense fog, our transformed LDS would still encode the fog-induced uncertainty inside its (augmented) process noise; it would not yield a more informative or accurate predictor. The point of the construction is purely algebraic: it allows us to analyze an equivalent noise model that has no direct feedthrough or measurement-noise term, without altering the fundamental challenge posed by high observation noise.
> > > >
> > > > For completeness, we provide the precise construction of this equivalent LDS below.
> > > >
> > > >
> > > > Consider the original linear dynamical system
> > > > \begin{align}
> > > >     x_{t+1} &= A x_t + B u_t + w_t, \\
> > > >     y_t^{\mathrm{orig}} &= C x_t + D u_t + \xi_t,
> > > > \end{align}
> > > > where $x_t \in \mathbb{R}^n$, $u_t \in \mathbb{R}^m$, $y_t^{\mathrm{orig}} \in \mathbb{R}^{d_y}$,
> > > > $w_t \in \mathbb{R}^n$, and $\xi_t \in \mathbb{R}^{d_y}$.
> > > >
> > > > Our goal is to construct an equivalent LDS whose output satisfies
> > > > \[
> > > >     y_t = y_{t-1}^{\mathrm{orig}} = C x_{t-1} + D u_{t-1} + \xi_{t-1}.
> > > > \]
> > > >
> > > > \paragraph{Augmented State.}
> > > > Define the augmented hidden state
> > > > \[
> > > >     z_t :=
> > > >     \begin{bmatrix}
> > > >         x_{t-1} \\
> > > >         u_{t-1} \\
> > > >         \xi_{t-1}
> > > >     \end{bmatrix}
> > > >     \in \mathbb{R}^{n + m + d_y}.
> > > > \]
> > > >
> > > > \paragraph{Output Matrix.}
> > > > We choose
> > > > \[
> > > >     C' := \begin{bmatrix} C & D & I_{d_y} \end{bmatrix},
> > > > \]
> > > > so that
> > > > \[
> > > >     y_t = C' z_t = C x_{t-1} + D u_{t-1} + \xi_{t-1}
> > > >     = y_{t-1}^{\mathrm{orig}}.
> > > > \]
> > > >
> > > > \paragraph{Dynamics.}
> > > > Using the identity
> > > > \[
> > > >     x_t = A x_{t-1} + B u_{t-1} + w_{t-1},
> > > > \]
> > > > we want $z_{t+1}$ to equal
> > > > \[
> > > >     z_{t+1} =
> > > >     \begin{bmatrix}
> > > >         x_t \\
> > > >         u_t \\
> > > >         \xi_t
> > > >     \end{bmatrix}.
> > > > \]
> > > > This is achieved by choosing
> > > > \[
> > > >     A' :=
> > > >     \begin{bmatrix}
> > > >         A & B & 0 \\
> > > >         0 & 0 & 0 \\
> > > >         0 & 0 & 0
> > > >     \end{bmatrix},
> > > >     \qquad
> > > >     B' :=
> > > >     \begin{bmatrix}
> > > >         0 \\
> > > >         I_m \\
> > > >         0
> > > >     \end{bmatrix},
> > > > \]
> > > > and process noise
> > > > \[
> > > >     w'_t :=
> > > >     \begin{bmatrix}
> > > >         w_{t-1} \\
> > > >         0 \\
> > > >         \xi_t
> > > >     \end{bmatrix}.
> > > > \]
> > > >
> > > > Then the augmented system satisfies
> > > > \[
> > > >     z_{t+1} = A' z_t + B' u_t + w'_t,
> > > > \]
> > > > and for all $t \ge 1$,
> > > > \[
> > > >     y_t = C' z_t = y_{t-1}^{\mathrm{orig}}.
> > > > \]
> > > >
> > > > \paragraph{Conclusion.}
> > > > Thus, by augmenting the hidden state with $(x_{t-1}, u_{t-1}, \xi_{t-1})$ and using the matrices above,
> > > > we obtain an LDS of the form
> > > > \[
> > > >     z_{t+1} = A' z_t + B' u_t + w'_t,
> > > >     \qquad
> > > >     y_t = C' z_t,
> > > > \]
> > > > whose output equals the original system's output with a one-step time shift.

---

> > ### Comment · Reviewer_SGuc · 2025-11-27
> >
> > I disagree with the authors’ argument regarding W1. Although the next-token prediction setup in this paper appears superficially similar to that used in LLMs—suggesting that the learned nonlinear dynamics might be universal—the two settings differ fundamentally. The problem studied in this paper requires continuous observations from the underlying dynamical system, whereas an LLM operates more like a simulation given an initial condition.
> >
> > A simple example helps illustrate my point. Consider listening to a speaker. One type of prediction involves continuously listening to the ongoing speech and predicting the speaker’s next word. Another type of prediction assumes that, given only a few initial words indicating the topic—and perhaps some samples of the speaker’s past speeches—one can predict the entire script, which might be hidden in the speaker's pocket or mind. The former corresponds to learning an observer, which is the focus of this work. The latter corresponds to system identification or learning a system model.
> >
> > An observer captures some information about the underlying nonlinear system but differs from the nonlinear model. Another technique example is that nonlinear system could be chaotic but its observer is a very stable dynamics, as illustrated in this paper.
> >
> > In summary, the title of the paper makes an overstated claim.

---

> > > ### Author Response · Authors · 2025-11-28
> > > **response to reviewer**
> > >
> > > The reviewers misunderstands the connection we draw to next-token prediction in LLMs. Modern LLMs are trained purely via next-token prediction and are later used autoregressively for generation. Our work analyzes this same foundational learning problem: predicting the next observation given the past. Our theoretical guarantees establish diminishing regret for this next-token prediction task when the data are generated by a nonlinear dynamical system.
> > > Regarding the reviewer’s concern about “universality,” we use the term in a technical sense. Much of the prior theory on predicting nonlinear dynamical systems provides guarantees only locally, e.g., under assumptions of local linearity, Lipschitz continuity, or accuracy within a small neighborhood of a trajectory. In contrast, our spectral predictor achieves universal no-regret learning: its performance is compared against the best predictor in hindsight over the entire sequence, without relying on any local linear approximation or stability neighborhood. This is the precise sense in which the method is “universal.”
> > > Finally, while our theory focuses on next-observation prediction (i.e., learning an observer), our experiments demonstrate that models trained in this fashion also perform well when used autoregressively, mirroring the behavior of LLMs trained with next-token prediction. Our claim does not equate observers with full system-identification models; rather, it emphasizes that next-token prediction with universal no-regret guarantees is a principled analogue of the core training objective used in large-scale autoregressive models.

---

### Official Review · Reviewer_zHeB · 2025-11-01

**Soundness:** 3
**Presentation:** 2
**Contribution:** 3
**Rating:** 4
**Confidence:** 2

**Summary:**

This paper studies the problem of learning a marginally stable unknown nonlinear dynamical
system. The authors show that, under suitable assumptions, such systems can be approximated by a high-dimension linear dynamicl systems. Then, the authors design a predictor, using techniques from spectral filtering, with vanishing regret for nonlinear dynamical systems with finitely many marginally stable modes. Along the way, the authors identify a complexity measure of nonlinear dynamical systems that governs learnability. Finally, the authors complement their theoretical bounds with experiments on the Lorenz systems.

**Strengths:**

- This paper is really well-written. I actually really enjoyed the prose, and the authors do a great job describing the significance of their results and providing the reader with intuition.
- The problem studied is relevant and of interest to the broader ML community

**Weaknesses:**

Unfortunately, there are several issues that need to be addressed. First, the paper does not conform to the ICLR style guides as the margins are too small. More importantly, I do not think the main text is very understandable, as key quantities/results are left undefined. For example:
- A "Luenberger program" is repeatedly discussed, but never formally defined in the main text
- The system's "robust observability constant $Q_{\star}$ is repeatedly referenced, but never defined in the main text
- The key ideas of spectral filtering are not introduced
- Key assumptions are not fully defined (i.e. see Assumption 3.1 references a Lemma in the Appendix)
- Key algorithmic pieces are not fully specified (see Line 2 of Algorithm 1)
- Key definitions are missing from the main text (i.e. see $Q_{\star}$ in Theorem 4.1).
- Proper descriptions of the plots are missing. What are the shaded areas in Figure 1?
- SFeDMD is claimed to be an algorithm by the authors, but not properly defined anywhere in the main text.

In my opinion, the main text of a paper should be as self-contained as possible, which I do not believe is the case with this paper. I do not doubt the contributions of this paper. However, I feel that this paper needs a significant rewrite of its main text in order fit the usual standards of a top-tier ML conference paper.

**Questions:**

See Weaknesses.

---

> ### Author Response · Authors · 2025-11-19
> **Response to Reviewer zHeB**
>
> We thank the reviewer for the thoughtful and constructive feedback. Below we address each point in turn.
>
> **Responses to the Weaknesses**
>
> *W1. A "Luenberger program" is repeatedly discussed, but never formally defined in the main text.*
>
> *A.* The full definition appears in Appendix C (Definition C.1).
>
> *W2. The system's "robust observability constant" $Q_\*$  is repeatedly referenced, but never defined in the main text.*
>
> *A.* This is currently defined in Appendix C (Definition C.7).
>
> *W3. The key ideas of spectral filtering are not introduced.*
>
> *A.* Spectral filtering was introduced by Hazan et al. (2017), and has since been discussed in multiple papers and as the main topic of Chapter 11 of the textbook “Introduction to Online Control” by Hazan and Singh. Given the space constraints and the fact that it is an established technique with pedagogical expositions (blog posts and a textbook chapter) freely available, we consider it sufficient to cite the relevant resources.
>
> *W4. Key assumptions are not fully defined (i.e. see Assumption 3.1 references a Lemma in the Appendix)*
>
> *A.* We have since been able to remove Assumption 3.1(iii) entirely, which will be reflected in the revised manuscript. In general we agree that key parts of the argument are only informally introduced in the main paper with full definitions/proofs deferred to the appendices (such as the careful definition of $Q_*$). Space constraints make this necessary, and we have tried to be as efficient as possible with our presentation.
>
> *W5. Proper descriptions of the plots are missing. What are the shaded areas in Figure 1?*
>
> *A.* The shaded regions correspond to ±1 standard deviation. We will add this information to the figure captions.
>
> *W6. SFeDMD is claimed to be an algorithm by the authors, but not properly defined anywhere in the main text.*
>
> *A.* SFeDMD is defined in Appendix B.2; we placed it there due to space constraints. See also the paragraph in lines 354-357.

---

### Official Review · Reviewer_YzHr · 2025-11-01

**Soundness:** 3
**Presentation:** 2
**Contribution:** 3
**Rating:** 6
**Confidence:** 4

**Summary:**

This paper studies the problem of learning output predictor for nonlinear systems. The authors propose algorithms based on the idea of approximating a nonlinear dynamical system using high-dimensional linear dynamical system (LDS). Under certain assumptions, the authors show that their algorithm $\sqrt{T}$-regret for general nonlinear systems and the regret bounds become sharper when specializing to linear systems. The authors also show that a quantity ``$Q^{\star}$” plays a key role in their regret analysis and results, which corresponds to the LDSs used to approximate the nonlinear system.

**Strengths:**

1.	The authors provide comprehensive theoretical analysis for the proposed algorithm.
2.	The algorithm design and particularly the regret analysis seem to contain significant novelty.
3.	Existing works typically consider learning the output of linear systems. Extending it to nonlinear systems is a meaningful step.

**Weaknesses:**

1.	The assumptions made in the paper are pretty strong (e.g., Assumption 3.1), which require further justifications. In particular, the bounded state and output assumption can make the claim that the regret results hold for marginally-stable system vacuous. (Please see the questions 1-3 below)
2.	The proposed algorithm (Algorithm 1) may not be constructive, e.g., it is not fully clear how the input parameters can be (please see the question 3 below).
3.	The regret bounds provided in Theorem 4.1 depend exponentially on certain problem parameters.

**Questions:**

1.	The authors assume that $x_t$ and $y_t$ are upper bounded by $R$. While the authors claim that their regret results (including that for nonlinear system and linear system) hold for marginally-stable systems, the regret bounds scale as the upper bound $R$ and $R$ can in turn scale polynomially as $T$ when the underlying system becomes marginally stable. Thus, for marginally-stable systems, the overall regret upper bounds are no longer sublinear in $T$.
2.	How to justify Assumption 3.1(iii)? Since Assumption 3.1(iii) is required for the $(A^{\prime},C^{\prime})$, given the original nonlinear system, how to check whether this assumption holds? In particular, from Lemma C.5 in the Appendix, it is not fully clear how $A^{\prime}$ and $C^{\prime}$ may be explicitly constructed from the original nonlinear system.
3.	While the algorithm proposed in this paper works for learning the output of asymmetric LDSs, the $A$ matrix needs to be diagonalizable and normal, which is still not the general LDS. This point may need to be made explicit in the abstract of the paper.
4.	In the statement of Theorem 4.1, the input parameters $D$ and step size $\eta_t$ need to be chosen based on the quantity $Q^{\star}$. Is there a constructive/explicit way to obtain the value of $Q^{\star}$. In Definition C.7, the value of $Q^{\star}$ is given by an optimization problem which does not seem to yield an explicit solution. Additionally, what is $\gamma$ in the statement of Theorem 4.1, is it also an input parameter to Algorithm 1? The authors provide upper bounds on $Q^{\star}$. Will using the upper bounds enough for setting the aforementioned input parameters?
5.	In Line 301, the authors mention a class of marginally stable bounded nonlinear predictors, what is this class and could the authors provide a formal definition for it? Also, the authors mention at some places in the paper (e.g., in the abstract) the terminology marginally-stable nonlinear system, what is the formal definition for this?
6.	Please correct the double quotation marks in the paper.

---

> ### Author Response · Authors · 2025-11-19
> **Response to Reviewer YzHr**
>
> We thank the reviewer for the thoughtful and constructive feedback. Below we address each point in turn.
>
> **Response to the Weaknesses**
>
> *W1. The assumptions made in the paper are pretty strong...*
>
> *A.* Please see our response to Question 1 below, where we clarify when the boundedness assumption is non-vacuous for marginally stable systems, and how this assumption interacts with the regret guarantees. Our current regret bounds indeed do not capture every marginally-stable system, but we hope to convince the reviewers that (1) many interesting and quite nonlinear systems are captured and (2) the proof techniques (global linearization + improper LDS learning via spectral filtering + eigenvalue placement) are sufficiently flexible to extend to more general systems. In particular, the assumptions of bounded outputs and 1-Lipschitz dynamics are only used in the linearization of Lemma C.5, and alternative linearizations (via Koopman operators, for example) will likely be able to capture other systems.
>
> *W2. The proposed algorithm (Algorithm 1) may not be constructive...*
>
> *A.* Our theoretical results prove the existence of parameters $(h, m, \eta_t)$ that achieve sublinear regret, and that the usual optimization algorithms will find solutions which match these. While we do not provide a closed-form expression for these parameters in the main text, a tighter analysis indeed yields explicit expressions; we omitted those derivations for clarity of presentation. We clarify this further in our answer to Question 4.
>
> *W3. The regret bounds provided in Theorem 4.1 depend exponentially on certain problem parameters.*
>
> *A.* When Algorithm 1 is instantiated with the Vovk–Azoury–Warmuth forecaster, the regret scales as $\log⁡(Q_∗)$. If $Q_∗$​ itself depends doubly exponentially on some underlying system parameters, then the resulting regret will inherit exponential dependence. However, our key point is that whenever $Q_∗$​​ is exponential (rather than doubly exponential), the regret bound remains sublinear. In this sense, a small $Q_∗$​​ is a sufficient condition for learnability. We will clarify this interpretation in the discussion of Theorem 4.1.
>
> It is impossible to prove a guarantee which is always subexponential in all problem parameters, see the lower bound in Theorem E.2. The class of nonlinear dynamical systems that satisfies our assumptions includes computationally hard problems (such as inverting pseudo-random generators), which makes a problem-dependent regret bound a requirement. The best one can hope for at this level of generality is an algorithm whose regret bound indicates it does well on efficiently-solvable problems but suffers exponentially on difficult ones.
>
>
> **Responses to the Questions**
>
> *Q1. The authors assume that $x_t$  and $y_t$ are upper bounded by $R$...*
>
> *A.* We agree that in the worst case, $R=\Theta(T)$ for marginally stable systems. However, some important families of marginally stable systems (e.g., permutation systems) satisfy a uniform constant bound. More generally, whenever $R=\Theta(T^k)$ for any $k<1/4$, our regret bounds remain sublinear. We would also like to note that the assumption of $\|y_t\| < R$ is an artifact of our technique for approximation by LDS. The spectral filtering technique uses a weaker Lipschitzness-in-time assumption (see the constant $L_y$ in the main theorem of “Learning Linear Dynamical Systems via Spectral Filtering” by Hazan et al, 2017), and the interested reader can construct global linearizations in this regime in the spirit of our Lemma C.5. Combining these two ingredients would allow similar results with this assumption relaxed.
>
> *Q2. How to justify Assumption 3.1(iii)...*
>
> *A.* We have since found a way to remove Assumption 3.1(iii) entirely, which we will edit in the revised manuscript (it was without loss of generality since the unobservable subspace by definition does not affect the output sequence, and so it may be projected out).
>
> *Q3. While the algorithm proposed in this paper works for learning the output of asymmetric LDSs, the  matrix needs to be diagonalizable and normal...*
>
> *A.* There is no requirement of normality – spectral filtering (and most spectral LDS algorithms) handles any diagonalizable transition matrix with regret scaling linearly in the condition number of the diagonalizing transform. Note also that the set of matrices diagonalizable over the complex numbers is dense, so we may perturb any given LDS slightly to make it diagonalizable without changing the regret.

---

> > ### Author Response · Authors · 2025-11-19
> > **Additional responses**
> >
> > *Q4. In the statement of Theorem 4.1, the input parameters...*
> >
> > *A.* As the reviewer points out, computing $Q_*$ exactly is difficult since it requires a spectral understanding of a (good) linearization of the system. Definition C.7, which formulates it as a minimization, is the precise way to convey the fact that via improper learning the algorithm competes against any choice of $L$ which places the eigenvalues of $A’-LC’$ correctly. In this sense $Q_*$ is an intrinsic measurement of “how difficult it is to predict a given system using an LDS of arbitrarily-high dimension and certain spectral properties”, and it determines when an LDS-based algorithm will be successful. Compare with other intrinsic notions of complexity of a class of predictors (e.g. Rademacher, Littlestone, …), which are useful constructions because (1) they dictate learnability and (2) they can often be bounded using structural information, even though they are difficult to compute. The upper bound on $Q_*$ provided in Lemma F.2 applies to every system (such as a computationally-hard PRNG system), and in application one should look at their class of systems and reason about how their $Q_*$ looks (similarly to Rademacher, etc.).
> >
> > That being said, it is important that running the algorithm requires no knowledge of $Q_*$; ideally the practitioner runs some algorithm without knowing anything about the system, and when it is possible to do well then the algorithm “magically” does well. Our Algorithm 1 only requires knowledge of $Q_*$ for the parameters $D$ and $\eta$ (the diameter and learning rate of the optimizer, respectively), as opposed to requiring knowledge of the system to choose a particular parameterization of the predictor. Our view is that if a practitioner can run the exact same algorithm/code on pretty much any system they are given and only need to change/tune the learning rate, then it is a successful algorithm. In practice, we recommend hyperparameter sweeps to choose the learning rate, and practitioners will likely use Adam with their favorite learning rate schedule or a parameter-free optimizer over vanilla gradient descent anyway.
> >
> > The parameter $\gamma \in (0,1)$ can be thought of as a computational budget to devote to regression terms, which influences which stable complex eigenvalues of the lifted system are included in $\Sigma$. Smaller $\gamma$ leads to a smaller $\Sigma$ and larger $Q_*$. Users may choose to have more or fewer regression terms depending on computational considerations (such as behaviors when pre-compiled in Jax or torch, for example).  All our results hold for any $\gamma \in (0, 1)$.
> >
> > *Q5. In Line 301, the authors mention a class of marginally stable bounded nonlinear predictors...*
> >
> > *A.* We use this term to refer to nonlinear dynamical systems satisfying Assumption 3.1. For precise statements about learning against such classes, please see the theorems in Appendix G.
> >
> > *Q6. Please correct the double quotation marks in the paper.*
> >
> > *A.* We thank the reviewer for pointing this out and will correct the formatting.

---

### Meta-Review · Area_Chair_VV6B · 2025-12-28

**Summary:**

The submission proposes an online prediction method for nonlinear dynamical systems based on spectral filtering, arguing that a broad class of marginally stable nonlinear systems can be approximated by a high-dimensional linear dynamical surrogate and then learned with no-regret guarantees. Reviewers generally recognized meaningful novelty in extending spectral filtering machinery (including handling asymmetric dynamics and noise-correction ideas in the LDS component) and appreciated the ambition of connecting nonlinear prediction to online convex optimization with a system-dependent learnability notion. At the same time, the review set converges on the view that the current paper overstates what is learned (in particular relative to “learning nonlinear dynamics”), relies on strong and sometimes potentially vacuous assumptions, and is difficult to assess from the main text due to key definitions and algorithmic details being deferred to the appendix. Given that three of four reviewers are below the acceptance threshold and the rebuttal did not clearly resolve the central scope/clarity issues, I recommend Reject.

**Reviewer Concerns:**

A primary concern is scope and framing: multiple reviewers argue that the title/abstract and parts of the narrative suggest learning a nonlinear dynamical system in a broad sense, whereas the technical setting is next-step prediction with access to ground-truth observations at each step (observer-style prediction), which is materially weaker than system identification or autonomous multi-step rollout guarantees; the rebuttal acknowledges this distinction and proposes clarifying edits, but at least one reviewer remained dissatisfied and explicitly reiterated that the title/LLM analogy is misleading. A second concern is the strength and interpretability of assumptions (e.g., bounded state/output, Lipschitz-type stability, marginal stability phrasing, and how “modes” are meant formally), including the possibility that boundedness constants may scale with horizon in marginally stable regimes, undermining the practical meaning of “sublinear regret” without additional structure; the rebuttal provides partial discussion and proposes removing one assumption, but does not fully dispel the perception that the theory applies to a narrower or more delicate regime than the paper’s language suggests. Third, reviewers flagged constructiveness and usability of the theory, notably that key parameters and the complexity quantity Q are not practically computable, and that some bounds can have unfavorable (e.g., exponential) dependence; the rebuttal argues that the algorithm need not “know” Q in practice and that hyperparameter sweeps are acceptable, but this does not fully address concerns about how to operationalize the theoretical guarantees. Fourth, presentation quality is a major issue: several reviewers found the main text not self-contained, with essential objects (e.g., “Luenberger program,” Q, assumptions, and algorithmic specification) left undefined or only defined in appendices, along with additional issues such as formatting/style-guide deviations and missing plot-caption details; while the rebuttal promises revisions, the current version remains hard to evaluate as a conference paper. Finally, reviewers requested stronger empirical validation, including noisy settings consistent with the claimed robustness and more competitive nonlinear baselines (e.g., GP state-space models), and also questioned certain broad statements about system identification; the rebuttal adds clarifications and related-work intent but does not supply new experimental evidence during the review.

**Reviewer Scores:**

Reviewer YzHr scored 6 pre-rebuttal (marginally above threshold) and, based on the tone of the exchange and the rebuttal addressing several technical questions (including removing an assumption and clarifying diagonalizability requirements), is most plausibly unchanged around 6 post-rebuttal rather than moving decisively upward. Reviewer zHeB scored 4 pre-rebuttal (below threshold) with emphasis on missing main-text definitions and overall understandability; although the rebuttal explains where definitions live and promises rewriting, the core critique targets the current submission quality, so a reasonable estimate is remaining ~4 (at most 5 if credit is given to planned revisions). Reviewer SGuc scored 4 pre-rebuttal and subsequently reiterated disagreements on central framing points (and on how the measurement-noise discussion was presented), suggesting no improvement and possibly a slight decrease, so ~3–4 post-rebuttal is the most defensible estimate. Reviewer xUFb scored 4 pre-rebuttal and explicitly stated they would keep their score, implying 4 post-rebuttal. Overall, the score distribution after rebuttal is therefore best estimated as one borderline accept and three below-threshold reviews, which supports the Reject recommendation.

---

### Decision · Program_Chairs · 2026-01-26

Reject